# CONCEPTUAL SCAN: LEARNING WITH AND ABOUT RULES

## ABSTRACT

The ability to learn from a mix of rules and examples and to reflect on the learned abstractions is an important aspect of human intelligence. At the same time, there is a lack of benchmarks that systematically test for this ability, which makes it hard to evaluate the degree to which it is present in state-of-the-art ML architectures. We introduce a novel task format for such benchmarks by using an example structure that allows us to explicitly provide and ask about rules that are relevant for the given task. We present a simple dataset illustrating this format, and we use it to analyze the performance of a variety of T5-based ML models. We identify three challenge areas in this setup: maintaining consistency between learned rules and their application, scaling to larger rule sets, and compositional generalization.

## 1 INTRODUCTION

Machine learning algorithms are typically designed to be able to learn functions from examples. This is a very general paradigm, but it does not explicitly capture some aspects of human learning. Humans, in contrast, are able to learn both by being shown examples of the task to accomplish and by being told rules or instructions about this task. They can even provide relevant rules to others once they have learned the task from examples.

As a realistic illustration of this ability, consider the task of a personal assistant who, among other things, is expected to make movie recommendations based on the age and interests of a user. Even for a task such as this that would currently be considered a standard use case for an example-based recommender system, as humans, we do not learn how to perform this task exclusively by observing examples of movie recommendations. Instead, we can accomplish this task much more efficiently by also taking into account relevant knowledge in the form of rules that have been communicated to us explicitly, i.e., by "learning with rules". For recommending a movie to a girl called Anna, we may, among others, use the rules (and facts, which we consider a special case of rules) illustrated on the left side of Figure 1.

In addition to the ultimate goal of providing movie recommendations (e.g., "What movie could Anna watch?"), we would also expect a human to be able to answer the intermediate questions shown on

| Knowledge | Questions |
|---|---|
| • R rated movies are not appropriate for kids who are less than 17 years old. | • What movie could Anna watch? ⇒ Mission Impossible 6 |
| • The age of a person is the time that has passed since the person was born. | • How old is Anna? ⇒ 14 years |
| • The current date is September 28, 2022. | • Is Jerry Maguire appropriate for 14-year-olds? ⇒ No |
| • Mission Impossible 1 – 6 are PG-13 rated action movies starring Tom Cruise. | • Will Anna like Mission Impossible 6? ⇒ Probably |
| • Jerry Maguire is an R rated comedy starring Tom Cruise. | • Who is Anna's best friend? ⇒ I don't know |
| • Anna was born in January of 2008 and Tom Cruise is her favorite actor. | |
| • Anna saw Mission Impossible 1 -- 5 and liked all of them. | |

Figure 1: Personal assistants answer questions using knowledge consisting of rules and facts. Note that the last knowledge bullet point above can also be construed as an example of the underlying "movie recommendation" task, while the other bullet points represent other relevant knowledge. The first bullet point is a traditional "rule" that states conditional knowledge that can apply to many different movies. The second is a concept definition, which can be equivalently construed as a rule relating two different pieces of information about a person. The other bullet points are facts stated at varying levels of granularity.

the right side of Figure 1 – i.e., to "learn about rules" that are relevant to the ultimate task – and we would expect these questions to be answered *consistently* w.r.t. the provided knowledge and ultimate recommendation. These questions allow us to introspect the understanding of the assistant, e.g., to debug why a movie recommendation was not as expected.

Similar interactions between learning from examples and learning with and about rules can also be observed in simpler synthetic tasks. Consider, for instance, the SCAN task of Lake & Baroni (2017), which our work builds on. This task requires the learner to translate natural language commands (such as "jump left twice") to corresponding actions (such as "LTURN JUMP LTURN JUMP"). The learner is presented with a certain subset of some thousands of (command, action sequence) pairs during training and is then expected to translate unseen commands.

This focus on learning purely from examples, while typical of most traditional ML tasks, differs from the way one would "teach" a human the task, and indeed how the authors of the SCAN paper "teach" the task to their readers. On the one hand, while humans are also adept at guessing rules from examples, rather than depending on thousands of examples, they can often grasp the relevant rule from just a handful of examples (Lake et al., 2015), as we as readers may find ourselves doing when seeing the handful of illustrative examples of SCAN provided by the authors in the paper figures. More fundamentally, however, rather than expecting readers to learn the translation function purely from examples, the authors provide this function in a much

$\llbracket \text{walk} \rrbracket = \text{WALK}$
$\llbracket x_1 \text{ left} \rrbracket = \text{LTURN} \llbracket x_1 \rrbracket$
$\llbracket x_1 \text{ twice} \rrbracket = \llbracket x_1 \rrbracket \llbracket x_1 \rrbracket$
...

Figure 2: Examples of SCAN interpretation rules from Lake & Baroni (2017).

more direct and efficient fashion using a set of interpretation rules like those in Figure 2. The *explicit* nature of the provided rules has the additional advantage that it allows us to *deduce* the translation of a given command by applying the translation rules rather than having to always speculatively *induce* the translation by generalizing from a set of examples.

In this paper, we introduce *conceptual learning tasks (CLTs)*, which are a type of learning task that is specifically designed to evaluate the combination of such inductive and deductive learning, and we make the following contributions:

- We define the notion of a CLT (Section 2).
- We present a first simple instance of a CLT called *Conceptual SCAN (cSCAN)*, which is a synthetically-constructed conceptual learning variation of SCAN (Section 3).
- We formalize metrics to measure a learner's performance on cSCAN, including a novel measurement of consistency between learned rules and their application (Section 4).
- We analyze the performance of baseline ML architectures on cSCAN and identify three challenge areas: consistency, rule set size, and compositional generalization (Section 6).
- We provide the code used in generating cSCAN, constructing compositional generalization splits, and calculating consistency from experiment results.[1]

## 2 CONCEPTUAL LEARNING TASKS (CLTs)

### 2.1 DESIRED PROPERTIES

Motivated by the use case from the introduction and our goal of evaluating learning with and about rules, we interest ourselves in tasks with the following properties, which we formalize in the following section.

1. **Context.** The learner answers requests based on some explicit knowledge (context), which consists of "examples" that directly determine the replies of certain requests and "rules" that provide indirect information to do so. Part of this knowledge varies across contexts (e.g., transient knowledge about Anna's preferences or country-specific rules about movie ratings).

2. **Request.** In addition to requests corresponding to the ultimate goal, (e.g., "What movie could Anna watch?"), we can ask the learner whether an intermediate rule holds given the context (e.g., "Is Jerry Maguire appropriate for 14-year-olds?"). This allows us to test whether the learner "understands" the rules by checking for consistency between rules that the learner claims to be true (or false) and their application to answering the ultimate requests.

---

[1]To be released on GitHub upon paper acceptance. For an overview, see Appendices F and G.

3. **Output.** The learner needs to indicate in the output whether a reply follows deductively – and thus monotonically – from the context (e.g., "How old is Anna? $\Rightarrow$ 14") or whether it requires generalizing the context inductively (i.e, "Will Anna like Mission Impossible 6? $\Rightarrow$ Probably"), which would imply that the reply could change if more information were to become available (i.e., that the reply is defeasible). Also, the learner needs to identify if a request cannot be answered based on the given context (e.g., "Who is Anna's best friend? $\Rightarrow$ I don't know").

4. **Compositionality of rules.** The rules have a compositional structure, which means that it is possible to determine the meaning of an unknown rule from its syntax if we are given the meaning of a sufficient subset of rules. (E.g., we as humans understand the meaning of the rules shown in Figure 1 because of the compositional syntax and semantics of natural language. Even if we have never seen these exact sentences before, we know what they mean based on our exposure to other sentences built from the same natural language building blocks.)

## 2.2 STRUCTURAL DEFINITION

As a way of providing a concrete but generic task format that satisfies these properties, we define a *conceptual learning task (CLT)* as a supervised learning task $\mathcal{T}$ with the following structure. (See Appendix A for a discussion of the design choices.)

- The **task** $\mathcal{T} = \{e_1, ..., e_N\}$ is a finite set of examples $e_k \in \mathcal{E}$, where $\mathcal{E} = 2^{\mathcal{Q} \times \mathcal{R}} \times \mathcal{Q} \times \mathcal{R} \times \mathcal{U}$ denotes the set of possible examples.

- Each **example** $e_k = \langle C_k, q_k, r_k, u_k \rangle$ is a quadruple consisting of a context $C_k \in 2^{\mathcal{Q} \times \mathcal{R}}$, a request $q_k \in \mathcal{Q}$, a reply $r_k \in \mathcal{R}$, and a qualifier $u_k \in \mathcal{U}$. Of these, the context and request together form the input of the ML system, while the reply and qualifier together form the output. (See Appendix O for details on the exact format in which these are input and output by our T5 baseline systems.)

- The set of possible **requests** $\mathcal{Q}$ can be partitioned into rule requests $\mathcal{Q}_R \subseteq \mathcal{Q}$ (i.e., requests that ask whether a certain rule holds) and non-rule requests $\mathcal{Q}_N \subseteq \mathcal{Q}$.

- The set of possible **replies** $\mathcal{R}$ must contain dedicated elements representing true (**1**), false (**0**), and unknown (**?**), which are the only valid replies for rule requests $q \in \mathcal{Q}_R$.

- The set of **qualifiers** $\mathcal{U} = \{\mathbf{M}, \mathbf{D}\}$ indicates whether the reply follows monotonically from the context (**M**) or whether it is defeasible (**D**).

- Each **context** $C \in 2^{\mathcal{Q} \times \mathcal{R}}$ consists of a set of **context examples** $e_i = \langle q_i, r_i \rangle$, which represent either examples of an underlying task or other relevant knowledge expressed in "example" form. Note that unlike the top-level examples, these context examples do not themselves contain a context or qualifier, as for the purposes of this paper, we take all context examples to be unconditional and monotonic. (See Appendix C for a possible generalization of this definition.)

Note that even though the context is strictly a set of examples, it may still contain any rule $q \in \mathcal{Q}_R$ by means of including the example $\langle q, \mathbf{1} \rangle$, which asserts the rule $q$ to be true.

As a more complete illustration, Figure 3 shows a few examples from the cSCAN task, which is a CLT that we introduce in Section 3, while Appendix B shows the motivating example represented as a CLT.

## 2.3 CONSISTENCY REQUIREMENTS

In addition to satisfying the structural definition, for the purposes of this paper, we require CLTs to be logically consistent, which means that each example and the example set as a whole must be non-contradictory. For instance, if a CLT contains a monotonic example $\langle C, q, r, \mathbf{M} \rangle$, it may not contain an example $\langle C', q, r', \mathbf{M} \rangle$ where $C \subseteq C'$ and $r \neq r'$ as this would violate monotonicity. (Or using the example from the introduction, this would mean, for instance, that if for a given context the task contains "How old is Anna? $\Rightarrow$ 14" and "Is Jerry Maguire appropriate for 14-year-olds? $\Rightarrow$ No", it should not contain "Is Jerry Maguire appropriate for Anna $\Rightarrow$ Yes".) While this requirement could be relaxed in practice, consistency of the task is helpful, as it enables us to precisely measure the consistency of the learner.

For the task to be logically consistent, it requires at minimum that the monotonic examples adhere to the axioms of propositional logic. (See Appendix D for a formalization of the consistency requirements.)

Note that even while requiring a CLT to be logically consistent, we still allow for the common phenomenon in which rules are stated in a form that is only "usually" true. Exceptions to a rule are allowed so long as the rule is assigned the qualifier "defeasible" (**D**).

## 2.4 CLASSIFICATION OF EXAMPLES

The structure of CLTs allows us to classify examples according to the following dimensions.

- Request $q \in \mathcal{Q}$: Rule request ($q \in \mathcal{Q}_R$) vs. non-rule request ($q \in \mathcal{Q}_N$)
- Reply $r \in \mathcal{R}$: Known vs. unknown (**?**), and, for rule requests, true (**1**) vs. false (**0**)
- Qualifier $u \in \mathcal{U}$: Monotonic (**M**) vs. defeasible (**D**)

Each class of example should be reasonably well represented in a conceptual learning dataset.

## 3 CONCEPTUAL SCAN (CSCAN)

One benefit of abstracting out key properties of interest from our motivating use case into the definition of a CLT is that we can now seek to study the capabilities of current ML methods on this family of tasks by starting with a very simple CLT instance, which illustrates the key dynamics of a CLT in as pure as possible a form. In particular, it gives us a basis by which we can construct a simple synthetic task that enables exploration of ML system performance on CLTs, while carefully controlling the task complexity. In this section, we present one such "simplest possible" CLT, named Conceptual SCAN (cSCAN). cSCAN is a conceptual learning adaptation of the SCAN task (Lake & Baroni, 2017), which was itself originally presented as a kind of "simplest possible" compositional generalization task. We construct cSCAN according to a recipe illustrating one possible way of deriving a CLT from a base task.

### 3.1 BASE TASK: SCAN

SCAN is a task where natural language commands (such as "jump left twice") are translated into sequences of actions (such as "LTURN JUMP LTURN JUMP"). SCAN was designed to evaluate compositional generalization using non-standard train-test splits, which is one of the themes we explore in this paper. In addition, SCAN is an attractive choice of base task for a first conceptual learning benchmark because it can be generated automatically from a simple set of rules: a phrase-structure grammar for generating the valid input commands and a set of interpretation rules that specifies how to compute the action sequence for each command (see Figure 2 and Appendix E.1).

### 3.2 CONSTRUCTING CSCAN

In contrast to SCAN, which tests whether a learner can learn one specific translation from commands to actions, the goal of cSCAN is to test whether the learner can learn to perform a family of SCAN-like tasks using knowledge that consists of an arbitrary mix of rules and examples.

Note that moving from a single task to a family of related tasks is essential for cSCAN because it forces the learner to take into account the knowledge provided by the context rather than just memorize behavior that is constant across all examples. In our original motivating example, this corresponds to the fact that we do not want a learner to make movie recommendations in a single, fixed scenario, but rather based on knowledge that may differ from person to person and evolve over time.

cSCAN is constructed using the following recipe.

**Step 1: Accommodate the base task.** A CLT subsumes the base task. This means that any input of SCAN is a valid request $q \in Q$, and any output of SCAN is a valid reply $r \in R$.

**Step 2: Make some of the relevant knowledge available as explicit rules.** We choose which part of the relevant knowledge we want to be able to talk about explicitly and vary across examples. For cSCAN, we choose to talk only about interpretation rules like those shown in Figure 2. This means that any interpretation rule is a valid request $r \in \mathcal{R}$, which allows us to assert such a rule in the context (by adding the example $\langle r, \mathbf{1} \rangle$ to the context) and teach / ask the learner whether any such rule holds. CLTs require that explicit rules have a

$C_1$ = { ⟨$⟦walk⟧$ = WALK, 1⟩, ⟨$⟦run⟧$ = RUN, 1⟩, ⟨$⟦$turn left$⟧$ = LTURN, 1⟩, ⟨$⟦x_l$ left$⟧$ = LTURN $⟦x_l⟧$, 1⟩, ⟨$⟦x_l$ thrice$⟧$ = $⟦x_l⟧$ $⟦x_l⟧$ $⟦x_l⟧$, 1⟩, ⟨$⟦x_l$ and $x_2⟧$ = $⟦x_l⟧$ $⟦x_2⟧$, 1⟩, ⟨walk twice, WALK WALK⟩, ⟨run left twice, LTURN RUN LTURN RUN, ⟨walk twice and run, WALK WALK RUN⟩ }

| | | |
|---|---|---|
| $e_1$ = ⟨$C_1$, walk left and run, LTURN WALK RUN, **M**⟩ | $e_3$ = ⟨$C_1$, run twice, RUN RUN, **D**⟩ | $e_5$ = ⟨$C_1$, run twice and walk, RUN RUN WALK, **D**⟩ |
| $e_2$ = ⟨$C_1$, $⟦x_l$ left and run$⟧$ = $⟦$LTURN $x_l⟧$ RUN, 1, **M**⟩ | $e_4$ = ⟨$C_1$, $⟦x_l$ twice$⟧$ = $⟦x_l⟧$ $⟦x_l⟧$, 1, **D**⟩ | $e_6$ = ⟨$C_1$, $⟦x$ twice and $y⟧$ = $⟦x⟧$ $⟦y⟧$ $⟦y⟧$, **0**, **D**⟩ |

$C_2$ = { ⟨$⟦walk⟧$ = RUN, 1⟩, ⟨$⟦run⟧$ = JUMP, 1⟩, ⟨$⟦jump⟧$ = LOOK, 1⟩, ⟨$⟦$turn left$⟧$ = RTURN, 1⟩, ⟨$⟦x_l$ left$⟧$ = $⟦x_l⟧$ RTURN, 1⟩, ⟨$⟦x_l$ and $x_3⟧$ = $⟦x_l⟧$ $⟦x_l⟧$ $⟦x_3⟧$, 1⟩, ⟨walk twice, RUN RUN RUN⟩, ⟨run left twice, JUMP RTURN JUMP RTURN JUMP RTURN⟩, ⟨jump twice and walk, RUN LOOK LOOK RUN⟩, ⟨walk thrice, RUN RUN RUN RUN⟩, ⟨run thrice, JUMP⟩, ⟨turn left thrice, RTURN RTURN⟩}

| | |
|---|---|
| $e_7$ = ⟨$C_2$, jump left twice and walk, RUN LOOK RTURN LOOK RTURN RUN, **D**⟩ | $e_9$ = ⟨$C_2$, jump thrice, ?, **D**⟩ |
| $e_8$ = ⟨$C_2$, $⟦x_l$ left twice and $x_2⟧$ = $⟦x_2⟧$ $⟦x_l⟧$ RTURN $⟦x_l⟧$ RTURN $⟦x_2⟧$, 1, **D**⟩ | $e_{10}$ = ⟨$C_2$, $⟦x_l$ thrice$⟧$ = $⟦x_l⟧$ $⟦x_l⟧$, **0**, **M**⟩ |

Figure 3: Hypothetical cSCAN examples based on two different contexts that contradict each other.

compositional structure (Section 2.1). This is the case for cSCAN because the meaning of the interpretation rules can be determined using a compositional grounding function that we provide in Appendix E.

**Step 3: Generate or curate examples.** For cSCAN, we generate examples automatically using a Python program, which we summarize here and describe in detail in Appendix F. As a first step of generating each example, we create a context by first picking a coherent set of interpretation rules like those shown in Figure 2 (or in Figure 4 from the appendix) and then choosing which of those rules to (a) provide explicitly in the context, (b) illustrate implicitly through context examples, (c) illustrate insufficiently or not at all, or (d) contradict in one or more cases. Once we have fixed a context, we choose a request, reply and qualifier that, together with the context, satisfy the consistency criteria stated in Section 2.3 and agree with the task's assumed inductive bias for distinguishing between a rule being "defeasibly true" vs. "unknown" (see Appendix E.4).

We make sure that the different example classes are evenly represented (see the statistics in Section 3.3) and that, for simplicity, contexts contain only unconditional, monotonic examples that do not conflict with one another. We provide a detailed specification of cSCAN in Appendix E.

### 3.3 THE CSCAN DATASET

**Examples.** Figure 3 shows some hypothetical cSCAN examples in human-friendly notation. (We chose these examples for simplicity and conciseness. For a sample of actual cSCAN examples, see Appendix I. For examples in the exact format in which they are presented to the baseline systems, see Appendix O.)

These examples are based on two contexts ($C_1$ and $C_2$) that are incompatible (e.g., they define the meaning of $⟦walk⟧$ differently) and correspond to different SCAN variations. Each context contains some explicit rules (e.g., $⟦x_1$ and $x_2⟧ = ⟦x_1⟧⟦x_2⟧$ in $C_1$) as well as rules that are illustrated implicitly via examples (e.g., $⟦x_1$ twice$⟧ = ⟦x_1⟧⟦x_1⟧$ in $C_1$). The qualifier of the examples indicates whether any implicit rules are needed to determine the reply (in which case the qualifier is **D**).

In context $C_2$, we do not provide explicit rules for $⟦x_1$ twice$⟧$ as well as $⟦x_1$ thrice$⟧$. While the rule $⟦x_1$ twice$⟧ = ⟦x_1⟧⟦x_1⟧⟦x_1⟧$ is expected to be induced from the provided examples, there is no obvious rule that can be induced for $⟦x_1$ thrice$⟧$. As a consequence, we expect the learner to reply "unknown" (?) for the request "jump thrice". At the same time, we expect the learner to reply "false" for the rule $⟦x_1$ thrice$⟧ = ⟦x_1⟧⟦x_1⟧$ because there is an example in the context that contradicts it.

**Rule space variants.** In order to cover a range of rule set complexities, we construct two versions of cSCAN (cSCAN-B and cSCAN-X), using different sizes of rule space (Table 1).

cSCAN-B (short for "cSCAN-Base") uses a fixed phrase-structure grammar equivalent to that of the original SCAN task, as reformulated in Nye et al. (2020) using 14 interpretation rules. Action sequences are kept short by allowing a token or variable to be repeated at most twice in an interpretation rule's output sequence. This ensures that cSCAN-B examples do not exceed input size of 2048 tokens or output size of 256 tokens in our baseline models.

cSCAN-X (short for "cSCAN-Extended") is based on a richer grammar space, which extends the original SCAN phrase-structure grammar with additional terminals, variations of existing rules, and an additional level that enables adverbs. Output sequences for individual interpretation rules are allowed to contain up to 4 repetitions of any given token or variable, which is the same as in original SCAN, but longer than in cSCAN-B. To keep context sizes manageable, we apply rule sampling for each context, so that the number of interpretation rules actually used in any given context is the same as in cSCAN-B, and apply a limit to the

Table 1: cSCAN rule space variants. cSCAN-B uses a fixed phrase-structure grammar equivalent to that of the original SCAN task, while keeping action sequences short. cSCAN-X is based on a richer grammar space, while using rule sampling to keep the number of rules used in any given context equivalent to the number of rules in cSCAN-B and the reformulation of original SCAN.

| Variant | Command grammar | | Action sequences | Context | Max # tokens | |
| | Depth | Terminals | Max length | Underlying rules | Input | Output |
|---|---|---|---|---|---|---|
| cSCAN-B | 6 | 13 | 4 | 14 | 2048 | 256 |
| cSCAN-X | 7 | 36 | 8 | 14 | 4096 | 512 |
| Reformulated SCAN | 6 | 13 | 8 | 14 | | |

Table 2: Key statistics of the main cSCAN datasets. For full details, see Appendix H.

| | cSCAN-B Random | | cSCAN-X Random | | cSCAN-B MCD | | cSCAN-X MCD | |
| | train | test | train | test | train | test | train | test |
|---|---|---|---|---|---|---|---|---|
| Number of examples | 100,000 | 100,000 | 100,000 | 100,000 | 100,000 | 10,000 | 99,999 | 10,000 |
| Number of contexts | 1,000 | 100 | 1,000 | 100 | 11,921 | 6,509 | 7,965 | 4,599 |
| Request: Rule (%) | 50.1 | 50.0 | 50.1 | 50.0 | 72.5 | 64.0 | 59.8 | 67.4 |
| Request: Non-rule (%) | 49.9 | 50.0 | 49.9 | 50.0 | 27.5 | 36.0 | 40.2 | 32.6 |
| Qualifier: Monotonic (%) | 46.3 | 46.0 | 50.0 | 48.0 | 26.9 | 36.6 | 37.0 | 38.9 |
| Qualifier: Defeasible (%) | 53.7 | 54.0 | 50.0 | 52.0 | 73.1 | 63.4 | 63.0 | 61.1 |
| Reply: Unknown (%) | 13.3 | 13.1 | 13.2 | 13.6 | 52.2 | 35.2 | 39.7 | 36.3 |
| Reply: Yes (%) | 21.8 | 21.8 | 21.9 | 21.5 | 12.0 | 18.1 | 12.7 | 18.5 |
| Reply: No (%) | 21.7 | 21.9 | 21.7 | 22.1 | 12.1 | 18.2 | 12.9 | 19.2 |

cumulative output sequence size across all interpretation rules used in the given context. These measures ensure that cSCAN-X examples do not exceed input size of 4096 tokens or output size of 512 tokens.

**Splitting methods.** For each of the two sizes of rule space, we prepare datasets based on two splitting methods: random and maximum compound divergence (MCD) (Keysers et al., 2020).

For the cSCAN Random variants, we generate 1200 contexts and split these contexts randomly into a train set of 1000 contexts and validation and test sets of 100 contexts each. For the train set, we generate 100 top-level examples per context, while for the validation and test contexts, we generate 1000 top-level examples per context so as to provide a denser signal of potential implications among the examples of each context for use in calculating the consistency metric. This leads to a total of 100K top-level examples in each of the train, validation and test sets.

For the cSCAN MCD variants, we apply a variation of the MCD splitting algorithm of Keysers et al. (2020) in order to evaluate the ability of the system to compositionally generalize to new rule combinations, which we consider a stronger test of the system's ability to "understand" the meaning of the rules and to apply them correctly in new situations. Specifically, we start by generating 12K contexts with 100 top-level examples each, yielding a pool of 1.2M top-level examples. We then annotate each top-level example with a set of atoms and compounds based on the phrase-structure grammar rules that were composed to form the top-level example request, and we split the set of top-level examples in such a way as to maximize the divergence in the distribution of compounds between train, validation, and test, while keeping the distribution of atoms nearly the same. Similarly to Keysers et al. (2020), we down-sample during the splitting process for more effective control of the distribution divergences, leading to a total of 100K top-level examples in train (comparable to cSCAN Random) and 10K top-level examples in each of validation and test.

**Statistics.** Table 2 gives an overview of the key statistics of the cSCAN Random datasets and representative MCD datasets. (See Appendix H for details of other cSCAN dataset variants.)

We construct the examples such that all the classes discussed in Section 2.4 are well represented. In particular, the Random datasets contain roughly equal numbers of rule vs. non-rule, as well as examples with replies of "true" (**1**) vs. "false" (**0**). There are slightly more defeasible examples than there are monotonic examples because "unknown" (?) examples are always qualified as defeasible.

Note that compared to the random split, the splitting method used in the cSCAN MCD variants leads to a somewhat less balanced dataset in terms of example classes, although each of the classes is still well

Table 3: Accuracy vs. consistency. Learner A is more accurate, but learner B is more consistent.

| Golden | Learner A | Learner B |
|---|---|---|
| $\langle C_1, [\![x_1 \text{ twice}]\!] = [\![x_1]\!][\![x_1]\!], \mathbf{1}\rangle$ | $\langle C_1, [\![x_1 \text{ twice}]\!] = [\![x_1]\!][\![x_1]\!], \mathbf{1}\rangle$ | $\langle C_1, [\![x_1 \text{ twice}]\!] = [\![x_1]\!][\![x_1]\!][\![x_1]\!], \mathbf{1}\rangle$ |
| $\langle C_1, [\![\text{walk}]\!] = \text{WALK}, \mathbf{1}\rangle$ | $\langle C_1, [\![\text{walk}]\!] = \text{WALK}, \mathbf{1}\rangle$ | $\langle C_1, [\![\text{walk}]\!] = \text{JUMP}, \mathbf{1}\rangle$ |
| $\langle C_1, \text{walk twice}, \text{WALK WALK}\rangle$ | $\langle C_1, \text{walk twice}, \text{JUMP JUMP}\rangle$ | $\langle C_1, \text{walk twice}, \text{JUMP JUMP JUMP}\rangle$ |

covered. Also, while it leads to a challenging split in terms of generalization to new top-level request patterns, it is potentially easier than the random split in terms of the contexts shown, as we do not prevent the same context from appearing in train and test, and due to the effect of down-sampling from a larger context pool, the total number of contexts that are shown in the MCD train set is an order of magnitude greater than those shown in the random train set.

## 4    METRICS

**Accuracy**    Our primary accuracy metric is example-level accuracy, where credit is given only when the reply + qualifier exactly matches the ground truth. For more nuanced error analysis, we secondarily track several accuracy variants that give credit for partially correct outputs (see Appendix N).

**Consistency**    A key aspect of learning with rules is that a learner does not just learn how to memorize and recite rules, but is actually able to combine and apply them consistently. For instance in cSCAN, if the learner believes that the rules $[\![x_1 \text{ twice}]\!] = [\![x_1]\!][\![x_1]\!]$ and $[\![\text{walk}]\!] = \text{JUMP}$ hold, a consistent learner should also believe that all combinations and applications of these rules hold, such as $[\![\text{walk twice}]\!] = \text{JUMP JUMP JUMP}$.

Note that unlike accuracy, this notion of consistency is not judging the correctness of individual predictions. Instead, it judges to what degree a whole set of predictions is consistent within itself. While a perfectly accurate learner would also be perfectly consistent, when accuracy is low to moderate, consistency can potentially be quite orthogonal to accuracy.

As an illustration, consider Table 3, which shows a golden example set as well as the predictions of two learners A and B. Learner A is more accurate because it gets 2 examples correct, whereas learner B gets none of the examples correct. At the same time, learner A is not consistent, because it is not able to correctly apply the two rules that it believes in to derive the example $\langle \text{walk twice}, \text{WALK WALK}\rangle$. In contrast, learner B is perfectly consistent, as it correctly combines the rules it believes in to derive the example $\langle \text{walk twice}, \text{JUMP JUMP JUMP}\rangle$.

To capture this notion of consistency, we introduce for any set of predictions $E \subseteq \mathcal{E}$ the *consistency metric* $\mathcal{C}(E)$, which is the percentage of subsets of $E$ that contain a logical implication, in comparison to the number of subsets of $E$ that contain an implication or a contradiction. This means that $\mathcal{C}(E)$ is a value in $[0,100]$: $\mathcal{C}(E) = 100$ says that the set E is perfectly consistent, while $\mathcal{C}(E) = 0$ says that E is completely inconsistent. (See Appendix D for a formalization of this metric and Appendix G for practicalities of calculation.)

For learner A, the example $\langle C_1, \text{walk twice}, \text{JUMP JUMP}\rangle$ contradicts the rules $\langle C_1, [\![x \text{ twice}]\!] = [\![x]\!][\![x]\!], \mathbf{1}\rangle$ and $\langle C_1, [\![\text{walk}]\!] = \text{WALK}, \mathbf{1}\rangle$, which means that the consistency of learner A is $100 \cdot {}^0/_1 = 0$. For learner B, the example $\langle C_1, \text{walk twice}, \text{JUMP JUMP JUMP}\rangle$ is implied by the rules $\langle C_1, [\![x \text{ twice}]\!] = [\![x]\!][\![x]\!][\![x]\!], \mathbf{1}\rangle$ and $\langle C_1, [\![\text{walk}]\!] = \text{JUMP}, \mathbf{1}\rangle$, which means that the consistency of learner B is $100 \cdot {}^1/_1 = 100$.

## 5    BASELINES

As baselines, we evaluate variations of T5 (Raffel et al., 2019), a Transformer encoder-decoder model (Vaswani et al., 2017), which when pre-trained on natural language served as a strong baseline on the original SCAN task (Furrer et al., 2020; Csordás et al., 2021; Ontañón et al., 2021).

The most computationally and memory intensive of the T5 architectures that we evaluate is the standard T5 architecture, which applies full self-attention in the encoder, in addition to self-attention and cross-attention in the decoder. We refer to these models as simply *T5*.

Motivated by the potentially large context size associated with CLTs, we further evaluate two variants of T5 that were designed to more efficiently scale to longer input sequences. *LongT5* (Guo et al., 2022) reduces the computational load of attention in the encoder by applying local attention within a sliding window (Ainslie et al., 2020). *LongT5-TGlobal* (Guo et al., 2022) extends LongT5 with a local-global

Table 4: Test results on cSCAN random splits by model size: $S$ (Small), $B$ (Base), and $L$ (Large).

| | | cSCAN-B Random | | | | | | cSCAN-X Random | | | | | |
| | | Accuracy | | | Consistency | | | Accuracy | | | Consistency | | |
| Model | Pretrain | S | B | L | S | B | L | S | B | L | S | B | L |
|---|---|---|---|---|---|---|---|---|---|---|---|---|---|
| T5 | True | **92.5** | **92.6** | **96.5** | **71.8** | **71.3** | **87.5** | **68.3** | **71.0** | **81.6** | **30.1** | **32.4** | **43.5** |
| | False | 19.3 | 17.9 | | 0.0 | 0.0 | | 16.8 | 17.2 | | 0.1 | 0.0 | |
| LongT5-TGlobal | False | 20.0 | 17.0 | | 0.2 | 0.0 | | 18.0 | 18.8 | | 0.2 | 0.1 | |
| LongT5 | False | 19.5 | 21.4 | | 0.0 | 0.1 | | 18.1 | 16.7 | | 0.1 | 0.1 | |
| T5 w/o Context | True | | 26.8 | | | 0.1 | | | 23.8 | | | 0.4 | |

attention sparsity pattern (Ainslie et al., 2020; Zaheer et al., 2020). The global attention is designed to facilitate propagation of attention across the full input sequence within two hops, in comparison with pure local attention, in which propagation of attention is limited by the size of the attention window.

For each architecture, we evaluate at minimum two sizes: Small (60M parameters) and Base (220M parameters). For the best-performing T5 architecture, we further evaluate on size Large (770M parameters). We report results for both pre-trained and non-pretrained versions of standard T5, but only non-pretrained versions of the long variants, as we failed to find a converging setup for the pre-trained versions (see Appendix L for details). For comparison, we also evaluate a naive variant of T5-Base (*T5 w/o Context*), which considers only the request, while ignoring the context. Additional details of the baseline configurations are provided in Appendix L and of the input-output format in Appendix O.

## 6 EXPERIMENTAL RESULTS AND DISCUSSION

### 6.1 CONSISTENCY AND SCALING TO LARGER RULE SETS

As a first set of experiments, we compare baseline performance on the random splits of the cSCAN-B and cSCAN-X datasets. The results are shown in Table 4.

On the smaller cSCAN-B rule space, it can be seen that the provided 100K examples are sufficient for a pre-trained full-attention Transformer to achieve accuracies in excess of 90%, with accuracies increasing steadily with model size. Even in this relatively simple setting, however, several challenges can be observed. First, it appears that appropriate pre-training of the model is critical, as all the Transformer variants when trained from scratch managed to learn only superficial statistical correlations, as evidenced by them failing to outperform a naive baseline. Second, regarding LongT5 and LongT5-TGlobal, while it is possible that performance could be improved through more thorough hyperparameter tuning, our initial results show these to struggle on the conceptual learning task. Specifically, non-pretrained versions suffer from the same poor performance as non-pretrained T5, while when fine-tuning from an existing checkpoint, we were not able to find a converging setup. One possible explanation is that unlike document summarization tasks, for which LongT5 produced strong results (Guo et al., 2022), CLTs may depend heavily on full attention over the context. If so, this could pose challenges in scaling to real-world conceptual learning tasks with even larger contexts. Third, while consistency scores for the evaluated models correlate roughly with accuracy, significantly more headroom is seen in consistency, with even the best-performing T5-Large scoring under 0.9, while the naive baseline and all non-pretrained models score close to 0.

On the cSCAN-X rule space, accuracy drops significantly for all sizes of T5, suggesting that scaling to larger rule sets will be a challenge. Consistency continues to correlate with accuracy for these models, but drops rapidly as performance degrades. Non-pretrained models continue to fail to outperform the naive baseline.

### 6.2 COMPOSITIONAL GENERALIZATION

As a second set of experiments, we evaluate the ability for baseline solutions to compositionally generalize on CLTs using the cSCAN-B MCD and cSCAN-X MCD datasets (Table 5).

Prior research on semantic parsing tasks has shown that while pre-trained Transformers exhibit strong performance on specialized cases of compositional generalization, they tend to struggle with more complex forms of compositional generalization, as reflected in low performance on MCD splits when an appropriate notion of "atom" and "compound" is identified (Furrer et al., 2020). Here we show that in the context of a conceptual learning task, one form of compositional generalization that is challenging for T5-based

models is generalization to new syntactic patterns in the request, even in the relatively easy setting where the same contexts could appear in train as in test.

Specifically, as can be seen in the cSCAN-B MCD results, when switching from random to MCD split, accuracy drops from over 90% to less than 70%, even for the largest of the pre-trained T5 models, illustrating that compositional generalization is a challenge for these models, independently of the size of the rule space. Accuracy on cSCAN-X MCD is roughly similar to both cSCAN-B MCD and cSCAN-X Random, suggesting that the challenges of compositional generalization and rule space size do not necessary compound.

Table 5: Test accuracy on different cSCAN MCD splits by model size: $S$ (Small), $B$ (Base), and $L$ (Large).

| Model | Pretrain | cSCAN-B MCD | | | cSCAN-X MCD | | |
|---|---|---|---|---|---|---|---|
| | | $S$ | $B$ | $L$ | $S$ | $B$ | $L$ |
| T5 | True | **53.8** | **54.7** | **67.6** | **70.7** | **69.2** | **76.4** |
| | False | 40.1 | 38.4 | | 41.6 | 39.9 | |
| LongT5-TGlobal | False | 40.6 | 39.3 | | 41.0 | 40.9 | |
| LongT5 | False | 40.7 | 38.0 | | 40.9 | 41.2 | |
| T5 w/o Context | True | | 46.2 | | | 42.7 | |

Note also that while accuracies for non-pretrained models are somewhat higher on the MCD splits than on the random splits, this is not actually a sign of stronger performance, but rather simply an artifact of the mix of example classes that occur in the different splits, due to the down-sampling that is performed when constructing the MCD split. As shown in Table 8, a side effect of this splitting algorithm was a relative increase in the proportion of defeasible rule examples vs. monotonic or non-rule examples and in the proportion of examples with "unknown" as the reply. This leads to an increase in the accuracy achievable by the naive "T5 w/o Context" baseline. On the MCD splits, like the random splits, none of the non-pretrained models manage to outperform the naive baseline.

For the MCD splits, we do not report consistency metrics, as due to technicalities of the MCD splitting algorithm, there ended up being insufficient signal for logical implications among the test examples, leading the consistency metric to be undefined in most cases. (See Appendix F for details.)

## 7 RELATED WORK

Here we discuss the most closely related lines of research. See Appendix K for more related work.

**Tasks providing knowledge as context.** In representing the input of a CLT as a request paired with a context, we build on a long tradition of QA and reasoning task formulations that provide knowledge relevant to a task via various forms of context, such as a text passage (Kwiatkowski et al., 2019; Weston et al., 2015), set of natural language statements (Talmor et al., 2020), knowledge graph fragment (Sinha et al., 2020), or grid world (Ruis et al., 2020). Our formulation of a CLT is novel in adopting a set-like context mixing rules and examples, which varies materially across examples.

**Meta-learning.** The presence of examples within the context of an example gives CLTs a nested structure that allows us to view CLTs through the lens of meta-learning or "learning to learn" (Thrun & Pratt, 1998). In this view, top-level examples that share the same context correspond to to an episode where the context examples are the training examples and the top-level examples (w/o the context) are the test examples. Closely related to cSCAN are two pieces of work that apply meta-learning to SCAN (Lake, 2019; Nye et al., 2020). Our approach differs in that we include in the context a mixture of rules and examples, and we use the synthetically-generated contexts to define a new task, rather than as a means to improve accuracy on the original SCAN task.

## 8 CONCLUSIONS AND FUTURE WORK

In this paper, we presented the cSCAN benchmark as a first instance of a "conceptual learning task" (CLT), following a task format motivated by a personal assistant use case. Through experiments on baseline solutions, we identified several challenge areas with headroom for improvement. As next steps, we are interested in exploring solutions to CLTs, including prompting of large language models, neuro-symbolic solutions, and improved ways of handling large set-like contexts. In parallel, we are interested in exploring CLTs based on a wider range of base tasks and rule formats, including non-synthetic tasks and tasks that draw on a full KB as context.

## 9 REPRODUCIBILITY STATEMENT

**Experiments** Appendix L describes the details of the baseline configurations that we used, together with other details of the environment in which we ran the experiments reported in this paper, while Appendix O provides details of the input-output format. Upon paper acceptance, we plan to release on GitHub both the cSCAN datasets and the code needed to reproduce the experiments.

**Dataset generation** The cSCAN datasets themselves were synthetically generated using a configuration-driven Python program described in Appendix F, which we also plan to open-source, together with the specific configurations used for each of the cSCAN datasets. While regeneration of the datasets is not necessary for reproducing our experiment results, researchers can use this code to generate new conceptual learning datasets based either on the existing cSCAN grammar spaces or on modified grammars. When the code is run with the provided configurations, it can reproduce the generation of datasets with statistically comparable content to the official cSCAN datasets.

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

APPENDIX

## A  CLT DESIGN CHOICES

Here we discuss the rationale behind the design choices made in our definition of a Conceptual Learning Task (CLT).

**Splitting the input into a set-like context and a request.** The main goal of CLTs is to test whether a learner is capable of learning from explicitly provided knowledge consisting of rules and examples. Assuming a supervised learning setup as a basis, the applicable knowledge must somehow be provided in the input of each example. It is therefore quite natural to split the input into two parts: context and request. Since the examples and rules that form the background knowledge do not usually have a specific order (in the movie recommendation example for instance, it does not matter whether we are first told that Top Gun is rated PG-13 or that Jerry Maguire is rated R), representing the context as an unordered set is a natural choice.

**Representing rule assertions as examples.** Another important property of CLTs is the ability to ask the learner explicitly whether a certain rule holds (in a given context). One straightforward way to achieve this is to include examples where the request asks for the validity of a certain rule and the output provides the corresponding truth value. These kind of examples also provide us with a natural way to assert (or refute) rules in the context, which allows us to represent the context simply as a set of examples (rather than a heterogeneous set containing both examples and a dedicated representation of rule assertions and refutations).

**Distinguishing monotonic and defeasible replies.** Once we include the context as part of the input, we can distinguish between two different methods of how the learner may determine the reply for a given request: deduction and induction. For deduction, the learner infers the reply for a given request from the context using deductive reasoning alone. As an illustration, consider an example where we assert in the context that Top Gun is a PG-13 movie and then ask for the rating of Top Gun in the request (see Appendix D for a formalization of the logic axioms underlying deductive reasoning).

For induction, the information provided by the context is not sufficient to unambiguously determine the reply for a given request. As an illustration, suppose that we ask for the rating of Jerry Maguire in an example whose context asserts that both Mission Impossible 1 and Jerry Maguire are movies starring Tom Cruise and that Mission Impossible 1 is rated PG-13. This information is not sufficient for us to deduce the answer. Instead, the learner needs to rely on inductive bias to determine whether it should speculatively generalize the PG-13 rating from Mission Impossible 1 to Jerry Maguire or whether it should play it safe and say that it doesn't know.

Deductive reasoning is always monotonic w.r.t. the context (i.e., new knowledge cannot lead to a different reply), so we use the qualifier $M$ to indicate deductive replies. Inductive reasoning is always defeasible w.r.t. the context (i.e., new knowledge may lead to a different reply), so we use the qualifier $D$ to indicate inductive replies.

## B  MOTIVATING EXAMPLE AS CLT

Here we show what the motivating example from the introduction (Figure 1) could look like when formulated in the syntax of a CLT. Here we take $\mathcal{Q}$ to be the space of natural language statements and questions.

Context containing assertions of relevant knowledge:

$C1 = \{\langle$"R rated movies are not appropriate for kids who are less than 17 years old.",$\mathbf{1}\rangle$

$\langle$"The age of a person is the time that has passed since the person was born.",$\mathbf{1}\rangle$

$\langle$"The current date is June 3, 2021.",$\mathbf{1}\rangle$

$\langle$"Mission Impossible 1 – 6 are PG-13 rated action movies starring Tom Cruise.",$\mathbf{1}\rangle$

$\langle$"Jerry Maguire is an R rated comedy starring Tom Cruise.",$\mathbf{1}\rangle$

$\langle$"Anna was born in January of 2008 and Tom Cruise is her favorite actor.",$\mathbf{1}\rangle$

$\langle$"Anna saw Mission Impossible 1 – 5 and liked all of them.",$\mathbf{1}\rangle$

...

$\}$

Top-level example set containing a mixture of non-rule examples (natural language QA for movie recommendations) and rule examples (which probe the model's understanding of intermediate steps in the recommendation process):

$$E = \{\langle C1, \text{"What movie could Anna watch?"}, \text{"Mission Impossible 6"}, \mathbf{D} \rangle$$
$$\langle C1, \text{"Anna is 14 years old."}, \mathbf{1}, \mathbf{M} \rangle$$
$$\langle C1, \text{"Anna is at least 17 years old."}, \mathbf{0}, \mathbf{M} \rangle$$
$$\langle C1, \text{"Jerry Maguire is appropriate for 14-year-olds."}, \mathbf{0}, \mathbf{M} \rangle$$
$$\langle C1, \text{"Anna will like Mission Impossible 6."}, \mathbf{1}, \mathbf{D} \rangle$$
$$\langle C1, \text{"Tom Cruise is Anna's best friend."}, \mathbf{?}, \mathbf{D} \rangle$$
$$...$$
$$\}$$

The above is a relatively straightforward translation of the motivating example into CLT syntax, while using the format of rule examples for all of the context examples and for all of the intermediate questions. If we assume that the request space $\mathcal{Q}$ and reply space $\mathcal{R}$ include natural language questions and answers for querying about background knowledge, as well as for the end goal of providing movie recommendations, the above example could be alternatively expanded to represent some of the background knowledge in non-rule format (e.g., $\langle$"What is the rating of Jerry Maguire?", "R"$\rangle$) and/or to represent some of the intermediate questions as non-rule top-level examples (e.g., $\langle C1,$"How old is Anna?", "14 years", $\mathbf{M}\rangle$).

## C  GENERALIZATION OF CLTS TO SUPPORT NESTED CONTEXTS

As discussed in Section 2, since the context of each top-level example in a CLT is itself represented as a set of "examples", we have similar but slightly different notions of "example" at two different levels in a CLT:

- **Top-level examples**, which we represent as quadruples of $\langle context, request, reply, qualifier \rangle$.
- **Context examples**, which in Section 2 we require to be unconditional and monotonic and which we thus represent as simply $\langle request, reply \rangle$ pairs.

While for the simple cSCAN task, it was sufficient to provide only unconditional monotonic examples in the context, in the general case, one could imagine extensions of the notion of a CLT to allow inclusion of conditional and/or defeasible examples in the context as well. In this more general view of a CLT, we can drop the distinction between top-level examples and context examples, and instead adopt a recursive structure in which each example is a quadruple of $\langle context, request, reply, qualifier \rangle$, while each context is a set of examples.

One motivation for this more general view of CLTs is if we were to think of each top-level CLT example as representing one observation of the behavior of a personal assistant with very strong decision-making capabilities (which we might call the "teacher"), whose behavior at all times is conditioned on the knowledge available to it, and whose knowledge is stored in a large and growing set-like knowledge base. In this view, the context of the top-level example can be thought of as a snapshot of the relevant contents of the assistant's knowledge base at the time that the assistant received the given request and output the given reply and qualifier.

Now let us further suppose that we have another personal assistant (which we might call the "student"), which itself has some kind of growing set-like knowledge base, whose contents may or may not agree with the contents of the teacher's knowledge base. One interesting question is how the student can go about selectively "learning" from the teacher, so as to emulate its decision-making capabilities, without necessarily accepting wholesale the full contents of its knowledge base, which may also include information that is transient, situation-specific, or debatable, and which may thus not be relevant or appropriate for the student to adopt. One natural approach could be to simply select the top-level examples of interest (i.e., to select the specific instances of the teacher's behavior that the student wishes to emulate) and assert those examples in the student's knowledge base. This would be the equivalent of storing the knowledge

that "if such and such things (i.e., the contents of the example's context) were true, then when faced with the given request, this would be the appropriate reply and the appropriate qualifier".

Under this approach, the student's set-like knowledge base would now come to contain full CLT examples (including context and qualifier). Continuing with the view of a "context" as being a snapshot of some or all of an assistant's knowledge base, our contexts could now contain examples that themselves contain non-empty contexts. The maximum depth to which we allow such recursive structures to continue could be considered an arbitrary choice in the design of a conceptual learning system or task.

## C.1 Generalized CLT

Below is a formalization of this more general definition of a CLT, along with some shorthand notations, by which the simpler CLT definition from Section 2 can be seen as just a special case of the general definition.

- The **task** $\mathcal{T} = \{e_1,...,e_N\}$ is a finite set of examples $e_k \in \mathcal{E}$, where $\mathcal{E} = \mathcal{I} \times \mathcal{O}$ denotes the set of possible examples, with $\mathcal{I}$ being the set of possible inputs and $\mathcal{O}$ the possible outputs.

- Each **example** $e_k = \langle i_k, o_k \rangle$ is a pair consisting of an input $i_k \in \mathcal{I}$ and an output $o_k \in \mathcal{O}$.

- Each **input** $i_k = \langle C, q \rangle$ is a pair consisting of a context $C \in 2^{\mathcal{E}}$ and a request $q \in \mathcal{Q}$. The set of possible requests $\mathcal{Q}$ can be partitioned into rule requests $\mathcal{Q}_R \subseteq \mathcal{Q}$ (i.e., requests that ask whether a certain rule holds) and non-rule requests $\mathcal{Q}_N \subseteq \mathcal{Q}$.

- Each **output** $o_k = \langle r, u \rangle$ is a pair consisting of the reply $r \in \mathcal{R}$ and a qualifier $u \in U$. The set of possible replies $\mathcal{R}$ must contain dedicated elements representing true (**1**), false (**0**), and unknown (**?**), which are the only valid replies for rule requests $q \in \mathcal{Q}_R$. The set of qualifiers $\mathcal{U} = \{\mathbf{M}, \mathbf{D}\}$ indicates whether the reply follows monotonically from the context (**M**) or whether it is defeasible (**D**).

For conciseness, we use the flat notation $\langle C, q, r, u \rangle$ to mean the nested pairs $\langle \langle C, q \rangle, \langle r, u \rangle \rangle$, we use $\langle C, q, r \rangle$ as a shorthand for the monotonic example $\langle C, q, r, \mathbf{M} \rangle$, and we use $\langle q, r \rangle$ as a shorthand for the unconditional monotonic example $\langle \varnothing, q, r, \mathbf{M} \rangle$. Hence, the example $\langle C, q, r \rangle$ means that given the context $C$ or any superset of $C$, the request $q$ is translated to the reply $r$, while $\langle q, r \rangle$ means that under all circumstances, the request $q$ is translated to the reply $r$.

Note that even though the context is strictly a set of examples, it may still contain any rule $q \in \mathcal{Q}_R$ by means of including the example $\langle \varnothing, q, \mathbf{1}, \mathbf{M} \rangle$, which asserts the rule $q$ to be true unconditionally. Similarly, we can express that the rule $q$ does not hold in a context by including the example $\langle \varnothing, q, \mathbf{0}, \mathbf{M} \rangle$.

We refer to an example as *unconditional* if it has an empty context.

Note also that since examples within a context are of the same form as top-level examples, they may in principle themselves contain contexts up to arbitrary levels of nesting. We can, however, for any given CLT, choose a maximum level to which we allow such nesting to occur. In cSCAN, for example, contexts contain only unconditional examples, so that there is no nesting of contexts.

## C.2 Additional Shorthand for Rule Assertions in a Context

To express the assertion of a rule $q \in \mathcal{Q}_R$ in contexts more concisely, we use $q$ to mean its unconditional monotonic assertion $\langle \varnothing, q, \mathbf{1}, \mathbf{M} \rangle$. For instance, $\langle \{q\}, q', r \rangle$ stands for $\langle \{\langle \varnothing, q, \mathbf{1}, \mathbf{M} \rangle\}, q', r \rangle$.

In the example from Figure 1, if we take $\mathcal{Q}$ to be the space of natural language statements and questions, then with the above shorthand the following would be equivalent:

- $\langle \{\langle \varnothing, \text{"Anna was born in January of 2008."}, \mathbf{1}, \mathbf{M} \rangle\}, \text{"Who is Anna's best friend?"}, \mathbf{?}, \mathbf{D} \rangle$

- $\langle \{\text{"Anna was born in January of 2008."}\}, \text{"Who is Anna's best friend?"}, \mathbf{?}, \mathbf{D} \rangle$

## D Formalization of Consistency Requirements and Metrics

In this appendix, we formalize the consistency requirements and consistency metrics of CLTs by mapping example sets into classical propositional logic. We start by summarizing the definitions and axioms of propositional logic, and we specify functions to capture the meaning of rules and the inductive bias. This

allows us to map example sets to logical formulas, which in turn allows us to formalize the consistency requirements as well as the metrics for measuring consistency of a learner.

For the purposes of this formalization, we assume the more general form of a CLT described in Appendix C, for which the simpler CLT form used for cSCAN follows as a special case.

Note that while we provide this formalization for reference purposes and to facilitate future investigation into the formal properties of CLTs, it is possible in practice to use CLTs and the related consistency metric without detailed consideration of this formalization.

## D.1 TERMINOLOGY

If not otherwise indicated, we assume that $C,C' \in 2^{\mathcal{E}}$ are arbitrary contexts, $q,q' \in \mathcal{Q}$ are arbitrary requests, $r,r' \in \mathcal{R}$ are arbitrary replies, $u,u' \in U$ are arbitrary qualifiers, $E,E',F \in \mathcal{E}$ are arbitrary example sets, and $e,e',f$ are arbitrary examples.

Furthermore, we assume that $\mathbf{M}(e)$ and $\mathbf{D}(e)$ denote the monotonic and defeasible variants of the example $e$, respectively. This means that for any $u \in \mathcal{U}$, $\mathbf{M}(\langle C,q,r,u \rangle) = \langle C,q,r,\mathbf{M} \rangle$ and $\mathbf{D}(\langle C,q,r,u \rangle) = \langle C,q,r,\mathbf{D} \rangle$.

## D.2 CLASSICAL PROPOSITIONAL LOGIC

For reference in later proofs, we summarize the basic definitions and axioms of Lukasiewicz classical propositional logic (Klement, 2004), which we adopt as-is, with only a change of symbols for the logical connectives to avoid ambiguity with the $\wedge$, $\vee$, and $\neg$ symbols that we use elsewhere in this document in first order logic statements.

We assume a set of propositional variables $\mathcal{V}$ which represent atomic formulas. General logical formulas $\mathcal{L}$ are recursively constructed from these atomic formulas using the logical connectives $\rightarrow$ (implication), $\llcorner$ (negation), $\underline{\wedge}$ (and), $\underline{\vee}$ (or), and $\equiv$ (equivalence). In our formalization, we assume the primitive connectives $\rightarrow$ and the constant $\perp$ (falsum), and we define the other connectives as follows (assuming that $x,y,z \in \mathcal{L}$).

$$\llcorner x := x \rightarrow \perp \tag{1}$$

$$x \underline{\vee} y := \llcorner x \rightarrow y \tag{2}$$

$$x \underline{\wedge} y := \llcorner(x \rightarrow \llcorner y) \tag{3}$$

$$x \equiv y := (x \rightarrow y) \underline{\wedge} (y \rightarrow x) \tag{4}$$

$$\top := \llcorner \perp \tag{5}$$

To formulate the propositional logic axioms and inference rules, we use the notation $x \vdash y$ to express that we can infer $y$ from $x$ and we use $\vdash x$ to express that $x$ is a tautology.

$$\vdash x \rightarrow (y \rightarrow x) \tag{6}$$

$$\vdash (x \rightarrow (y \rightarrow z)) \rightarrow ((x \rightarrow y) \rightarrow (x \rightarrow z)) \tag{7}$$

$$\vdash (\llcorner x \rightarrow \llcorner y) \rightarrow (y \rightarrow x) \tag{8}$$

$$x,(x \rightarrow y) \vdash y \tag{9}$$

The axioms (6), (7), (8) form the Lukasiewicz system, while the inference rule (9) is modus ponens.

## D.3 SEMANTICS OF RULES AND INDUCTIVE BIAS

To map example sets to logical formulas, we need to be able to refer to the semantics of rules and the inductive bias.

For the semantics of rules, we assume a grounding function $\mathcal{G} : \mathcal{Q}_R \rightarrow 2^{\mathcal{E}}$, which maps each rule to an equivalent set of examples. This means that in a consistent task, a rule $q \in \mathcal{Q}_R$ holds (i.e., $\langle \varnothing,q,\mathbf{1} \rangle$ is a valid example) if and only if each example in $\mathcal{G}(q)$ is valid. It also means that if the context of an example $e \in \mathcal{E}$

contains the assertion $\langle\varnothing,q,\mathbf{1}\rangle$ of a rule $q\in Q_R$, we can obtain an equivalent example $e'$ by replacing the assertion of $q$ with the examples $\mathcal{G}(q)$. We provide the grounding function of cSCAN in Appendix E.3.

Since a CLT requires a learner to induce as well as deduce rules, every CLT inherently assumes some form of inductive bias, which determines the criteria based upon which a learner is expected to induce a rule to be "true" vs. considering it to be "unknown". While the inductive bias of the task is in principle arbitrary (being dependant on the choice of the task designer or the needs of the real-world use case), the learner will need to be able to emulate this bias in order to perform well on the task. We formalize this domain-specific inductive bias through a bias function $\mathcal{B}\colon 2^{\mathcal{E}}\to 2^{\mathcal{E}_D}$, which maps each set of examples to the set of examples that are expected to be induced. We use $\mathcal{E}_M,\mathcal{E}_D\subseteq\mathcal{E}$ to denote the subsets of all monotonic and defeasible examples, respectively. We provide the bias function of cSCAN in Appendix E.4.

While the grounding and bias functions are needed to precisely formalize the consistency requirements and metrics, they are not a requirement for a conceptual learning task. This is important because providing complete grounding and bias functions may not be feasible or practical for more realistic CLTs. Instead, we may only provide partial functions, which means that the formalization in this section will only approximate the true consistency requirements and consistency metrics.

### D.4 MAPPING EXAMPLE SETS TO LOGICAL FORMULAS

We treat each example as a propositional variable (i.e, $\langle C,q,r,u\rangle\in\mathcal{V}$). This allows us to define the embedding function $\mathcal{M}\colon 2^{\mathcal{E}}\to\mathcal{L}$ as follows.

$$\mathcal{M}(\varnothing):=\top \tag{10}$$

$$\mathcal{M}(\{\langle C,q,r,u\rangle\}):=\langle C,q,r,u\rangle \tag{11}$$

$$\mathcal{M}(E\cup E'):=\mathcal{M}(E)\underset{\sim}{\wedge}\mathcal{M}(E') \tag{12}$$

The empty set maps to true (10), a set of size one maps to its only element (11), and union maps to logical conjunction (12). In addition, our embedding adheres to the following axioms.

$$\vdash\langle C,q,r,\mathbf{M}\rangle\equiv(\mathcal{M}(C)\to\langle\varnothing,q,r,\mathbf{M}\rangle) \tag{13}$$

$$\vdash\langle C,q,r,\mathbf{M}\rangle\to\langle C,q,r,\mathbf{D}\rangle \tag{14}$$

$$\mathcal{M}(C)\equiv\mathcal{M}(C')\colon\ \vdash\langle C,q,r,\mathbf{D}\rangle\equiv\langle C',q,r,\mathbf{D}\rangle \tag{15}$$

$$r\neq r'\colon\ \vdash\langle C,q,r,\mathbf{D}\rangle\underset{\sim}{\wedge}\langle C,q,r',\mathbf{D}\rangle\to(C\to\bot) \tag{16}$$

$$\vdash\langle\varnothing,q,\mathbf{1},\mathbf{M}\rangle\equiv\mathcal{M}(\mathcal{G}(q)) \tag{17}$$

$$\vdash\mathcal{M}(E)\to\mathcal{M}(\mathcal{B}(E)) \tag{18}$$

Axiom (13) says that the context of a monotonic example becomes the antecedent of a logical implication, and axiom (14) specifies that a monotonic examples implies the corresponding defeasible example. Axiom (15) says that defeasible examples that differ only in equivalent contexts are equivalent. Axiom (16) specifies that defeasible examples are functional or the context must be contradictory, which means that each request with a non-contradictory context must have a unique reply. Finally, axiom (17) specifies that the assertion of a rule is equivalent to its grounding, and axiom (18) says that each set of examples implies the set of examples that can be induced using the inductive bias.

Together with the axioms of propositional logic, we obtain the following theorems.

$$\vdash\langle C,q,r,\mathbf{M}\rangle\to\langle C\cup C',q,r,\mathbf{M}\rangle \tag{19}$$

$$\vdash\mathcal{M}(\{\langle C,q,\mathbf{1},\mathbf{M}\rangle\})\equiv\mathcal{M}(\{\langle C\cup C',q',r',u'\rangle\colon\ \langle C',q',r',u'\rangle\in\mathcal{G}(q)\}) \tag{20}$$

$$r\neq r'\colon\ \vdash\langle C,q,r,u\rangle\underset{\sim}{\wedge}\langle C,q,r',u\rangle\to(C\to\bot) \tag{21}$$

Theorem (19) is obtained from axioms (6), (13) and (12), and it says that monotonic examples behave monotonically w.r.t. their context. This allows us to rewrite the grounding axiom (17) to obtain theorem (20). The last theorem (21) says that functionality applies independently of the qualifier.

### D.5 CONSISTENCY METRIC

As discussed in Section 4, the consistency metric $\mathcal{C}(E)$ differs from the standard accuracy metric in that it does not judge the correctness of a learner's predictions on individual examples, but rather measures the degree to which a complete set of examples E is consistent w.r.t. to the axioms of classical propositional logic.

In essence, we define $\mathcal{C}(E)$ to be the percentage of subsets of $E$ that contain a logical implication, in comparison to the number of subsets of $E$ that contain an implication or a contradiction.

Note that the definition of the consistency metric provided in equation (26) could essentially be applied as-is to any arbitrary CLT, provided there is some way to identify the "implications" ($impl(E)$) and "contradictions" ($cont(E)$) among the learner's predictions. In practice, task designers may be free to apply any reasonable heuristic to identify such implications and contradictions. The formal definition of the consistency metric below can be considered an ideal, which we seek to emulate closely in the consistency metric implementation provided for cSCAN, as described in Appendix G.

$$
\begin{aligned}
E \hookrightarrow f :&\Leftrightarrow \nvdash \llcorner\mathcal{M}(E) \wedge \vdash \mathcal{M}(E) \to \mathcal{M}(\{f\}) \wedge (\nexists E' \subsetneq E : \vdash \mathcal{M}(E') \to \mathcal{M}(\{f\})) \wedge \\
&\quad \neg(f \in \mathcal{E}_D \wedge \vdash \mathcal{M}(E) \to \mathcal{M}(\{\mathbf{M}(f)\}))
\end{aligned}
\tag{22}
$$

$$
\begin{aligned}
E \nrightarrow f :&\Leftrightarrow (\nvdash \llcorner\mathcal{M}(E) \wedge \vdash \mathcal{M}(E) \to \llcorner\mathcal{M}(\{f\}) \wedge (\nexists E' \subsetneq E : \mathcal{M}(E') \to \llcorner\mathcal{M}(\{f\}))) \vee \\
&\quad (f \in \mathcal{E}_D \wedge E \hookrightarrow \mathbf{M}(f))
\end{aligned}
\tag{23}
$$

$$
impl(E) := \{F \subseteq E : (\exists f \in F : F \setminus \{f\} \hookrightarrow \{f\})\}
\tag{24}
$$

$$
cont(E) := \{F \subseteq E : (\exists f \in F : F \setminus \{f\} \nrightarrow \{f\})\}
\tag{25}
$$

$$
\mathcal{C}(E) := \begin{cases} 100 \cdot \frac{|impl(E)|}{|impl(E)| + |cont(E)|}, & |impl(E)| + |cont(E)| > 0 \\ NaN, & \text{otherwise} \end{cases}
\tag{26}
$$

Minimal implication $E \hookrightarrow f$ means that the non-contradictory example set $E$ implies the example $f$ and that there is no strict subset of $E$ that has the same property. Furthermore, we require matching qualifiers, which means that $E$ must not imply the monotonic variant of $f$ if the latter is marked as defeasible (22).

The minimal contradiction $E \nrightarrow f$ can be met by either of the following two conditions (23). First, $E \nrightarrow f$ holds if the non-contradictory example set $E$ implies the negation of $f$ and there is no strict subset of $E$ that has the same property. Secondly, $E \nrightarrow f$ holds if there is a qualifier mismatch, i.e., if $f$ is marked defeasible but $E$ minimally implies the monotonic variant of $f$. Note that we do not consider it to be a qualifier mismatch if $f$ is marked as monotonic and $E$ implies only the defeasible variant because there may be other evidence outside of $E$ that may justify the monotonic qualifier.

This allows us to define $impl(E)$ to be the set of subsets $F \subseteq E$ that contain a minimal implication (24) and $cont(E)$ to be the set of subsets $F \subseteq E$ that contain a minimal contradiction (25). Finally, the consistency metric $\mathcal{C}(E)$ is the percentage of subsets of $E$ that contain implications, in comparison with the number of subsets of $E$ that contain implications or contradictions (26). If the set $E$ does not contain any implications or contradictions, then the consistency metric is not defined.

**Illustration.** As an illustration, consider the following example set $E$, which consists of 4 rule assertions and 8 examples. (We assume the syntax and semantics of cSCAN, which is formally specified in Appendix E.)

$$
\begin{aligned}
E := \{ &\langle C_1, [\![u\ twice]\!] = [\![u]\!][\![u]\!], \mathbf{1} \rangle, &\text{(a)} \\
&\langle C_1, [\![x\ after\ y]\!] = [\![y]\!][\![x]\!], \mathbf{1} \rangle, &\text{(b)} \\
&\langle C_1, [\![walk]\!] = WALK, \mathbf{1} \rangle, &\text{(c)} \\
&\langle C_1, [\![jump]\!] = JUMP, \mathbf{1} \rangle, &\text{(d)} \\
&\langle C_1, walk, WALK \rangle, &\text{(e)} \\
&\langle C_1, eat\ twice, EAT\ EAT \rangle, &\text{(f)} \\
&\langle C_1, walk\ twice, WALK\ WALK\ WALK\ WALK \rangle, &\text{(g)} \\
&\langle C_1, jump\ twice, JUMP\ JUMP, \mathbf{D} \rangle, &\text{(h)} \\
&\langle C_1, walk\ after\ walk, WALK\ WALK \rangle, &\text{(i)}
\end{aligned}
$$

$$\langle C_1, \text{jump after walk}, \text{WALK JUMP}, \mathbf{D} \rangle, \tag{j}$$

$$\langle C_1, \text{walk after jump}, \text{WALK JUMP} \rangle, \tag{k}$$

$$\langle C_1, \text{walk twice after jump}, \text{JUMP WALK WALK} \rangle \} \tag{l}$$

In this example set there are a total of 5 minimal implication sets (with the implied example indicated in green above):

- $\{c,e\}$
- $\{b,c,i\}$
- $\{b,e,i\}$
- $\{a,b,c,d,l\}$
- $\{a,b,e,d,l\}$

And there are a total of 8 minimal contradiction sets (with the contradictory example indicated in red above), of which the following are because of an inconsistent reply:

- $\{a,c,g\}$
- $\{a,e,g\}$
- $\{b,c,d,k\}$
- $\{b,e,d,k\}$
- $\{b,d,g,l\}$

while the following are because of an inconsistent qualifier:

- $\{a,d,h\}$
- $\{b,c,d,j\}$
- $\{b,e,d,j\}$

Based on this, the consistency of example set $E$ would be $\mathcal{C}(E) = 100 \cdot {}^5/_{13} \approx 38.5$.

Note that example $d$ is both implied and contradicted. Also, since examples $c$ and $e$ are semantically equivalent according to cSCAN's grounding function (Appendix E.3), each minimal implication or contradiction set that involves example $c$ can be written alternatively using example $e$, causing such sets to appear in pairs in the lists above.

### D.6 CLT: DEFINITION AND CONSISTENCY REQUIREMENTS

To obtain a precise measure for the consistency of the predictions produced by a given learner, it is important that the task $\mathcal{T}$ itself be consistent. Specifically, this means that each example $e \in \mathcal{T}$ must be consistent on its own, and the example set $\mathcal{T}$ as a whole must be consistent. For the purposes of this paper, we further require that contexts are non-contradictory.

We capture these requirements using a predicate $consistent_{CLT}(\mathcal{T})$, which is recursively defined as follows.

$$consistent_{CLT}(\{\langle C,q,r,u \rangle\}) :\Leftrightarrow consistent_{CLT}(C) \wedge \nvdash \llcorner \langle C,q,r,u \rangle \wedge \tag{27}$$

$$(u = \mathbf{M} \Leftrightarrow \vdash \mathcal{M}(\{\langle C,q,r,u \rangle\})) \tag{28}$$

$$consistent_{CLT}(E) :\Leftrightarrow (\forall e \in E : consistent_{CLT}(\{e\})) \wedge (\mathcal{C}(E) = 100 \vee \mathcal{C}(E) = NaN) \tag{29}$$

We first define the consistency of an individual example $\langle C,q,r,u \rangle \in \mathcal{E}$, which requires that its context $C$ is consistent (and therefore non-contradictory), that the example itself is not a contradiction, and that it is qualified as monotonic if and only if it maps to a tautology (28). Then, we define that an example set $E \subseteq \mathcal{E}$ is consistent if and only if each individual example $e \in E$ is consistent and the example set is consistent as a whole (29) (i.e., $\mathcal{C}(E)$ must be 100 or undefined).

$$
\begin{array}{ll}
[\![\text{walk}]\!] = \text{WALK} & [\![x_1 \text{ and } x_2]\!] = [\![x_1]\!][\![x_2]\!] \\
[\![\text{run}]\!] = \text{RUN} & [\![x_1 \text{ after } x_2]\!] = [\![x_2]\!][\![x_1]\!] \\
[\![\text{jump}]\!] = \text{JUMP} & [\![\text{turn opposite left}]\!] = \text{LTURN LTURN} \\
[\![\text{look}]\!] = \text{LOOK} & [\![\text{turn opposite right}]\!] = \text{RTURN RTURN} \\
[\![\text{turn left}]\!] = \text{LTURN} & [\![\text{turn around left}]\!] = \text{LTURN LTURN LTURN LTURN} \\
[\![\text{turn right}]\!] = \text{RTURN} & [\![\text{turn around right}]\!] = \text{RTURN RTURN RTURN RTURN} \\
[\![x_1 \text{ left}]\!] = \text{LTURN}[\![x_1]\!] & [\![x_1 \text{ opposite left}]\!] = [\![\text{turn opposite left}]\!][\![x_1]\!] \\
[\![x_1 \text{ right}]\!] = \text{RTURN}[\![x_1]\!] & [\![x_1 \text{ opposite right}]\!] = [\![\text{turn opposite right}]\!][\![x_1]\!] \\
[\![x_1 \text{ twice}]\!] = [\![x_1]\!][\![x_1]\!] & [\![x_1 \text{ around left}]\!] = \text{LTURN}[\![x_1]\!]\ \text{LTURN}[\![x_1]\!]\ \text{LTURN}[\![x_1]\!]\ \text{LTURN}[\![x_1]\!] \\
[\![x_1 \text{ thrice}]\!] = [\![x_1]\!][\![x_1]\!][\![x_1]\!] & [\![x_1 \text{ around right}]\!] = \text{RTURN}[\![x_1]\!]\ \text{RTURN}[\![x_1]\!]\ \text{RTURN}[\![x_1]\!]\ \text{RTURN}[\![x_1]\!]
\end{array}
$$

Figure 4: SCAN interpretation rules as provided by Lake & Baroni (2017). Double brackets denote the interpretation function translating SCAN's linguistic commands into sequences of actions. Symbols $x1$ and $x2$ denote variables.

Note that while the above formulation of CLT consistency requirements is sufficient for cSCAN, it could be desirable to adjust this for more complex tasks, which we leave for future work. In particular, while this definition of consistency provides only basic control over the behavior of defeasible examples, we could imagine defining stricter consistency requirements for defeasible examples, e.g., by requiring them to adhere to the KLM properties (Sarit Kraus & Magidor, 1990; Casini et al., 2021). Also, while in this paper, for simplicity, we require contexts to be non-contradictory, this requirement is not strictly necessary for the task to be consistent or for the consistency metrics to be meaningful, as long as the top-level examples in the dataset do not contain a contradiction or contradict each other. Taken a step farther, for dealing with real-world datasets which may contain noise, it may be desirable to relax the requirement that the task be strictly consistent. In such an approach, the "requirements" of a CLT can be considered more as an aspiration rather than as strict requirements. The consistency metric could in that case still be calculated, but would be less reliable an indicator of learner consistency compared to the case where the task is consistent.

# E  SPECIFICATION OF CSCAN

This appendix contains a specification of the cSCAN task. It consists of the phrase-structure grammars to generate valid requests and replies, the compositional grounding function (which defines the meaning of the explicit rules), and the bias function (which defines the inductive bias). Together with the formalization of simple CLTs provided in Appendix D, this specifies the complete behavior of cSCAN.

Note that in this section, as in Appendix D, we assume the more general CLT formalism described in Appendix C, in which contexts can contain examples of the same form as the top-level examples. While the cSCAN specification could be expressed equivalently in terms of the simplified CLT formalism of Section 2, the more general formalism allows us to express some aspects of the specification more concisely, as we can thus describe the semantics of both top-level examples and context examples using the same grounding function.

For readability, we make use of the shorthand notation described in Section C.2 to allow expressing some of the more verbose examples more concisely.

Note also that while we provide the complete formal specification of cSCAN here as a reference, it is not necessary in general to provide a specification at this level of detail when defining future CLTs.

## E.1  RULE SPACES

Most of the specification of cSCAN is identical between cSCAN-B and cSCAN-X. In this section, we summarize the points that differ between the two, together with some notes on the original SCAN task for comparison.

**Original SCAN**

In the original SCAN task, natural language commands are generated by a fixed phrase-structure grammar as described in Lake & Baroni (2017), which is equivalent to the phrase-structure grammar shown for cSCAN-B in the top left in Figure 5. (Note that for readability, we renamed the non-terminals in this

| S → T and T | T → U twice | U → W opposite X | V → W X | W → walk | X → left | **Y → x1** |
|---|---|---|---|---|---|---|
| S → T after T | T → U thrice | U → W around X | V → W | W → run | X → right | **Y → x2** |
| S → T | T → U | U → V | | W → jump | | **Y → x3** |
| | | | | W → look | | **Y → x4** |
| | | | | W → turn | | |
| | | | | **W → Y** | **X → Y** | |

Figure 5: The green area on the top left is the phrase-structure grammar to generate the cSCAN-B commands. The red area on the lower right is the extension to generate the left-hand side of cSCAN-B rules.

presentation to proceed alphabetically, beginning with "S" as the traditional start symbol.) The mapping from command to action sequence follows the fixed set of interpretation rules shown in Figure 4.

As can be seen in the figure, action sequences in the original SCAN task are constructed from a set of 6 possible actions.

$$\mathcal{A} := \{\text{WALK,RUN,JUMP,LOOK,LTURN,RTURN}\}$$

**cSCAN-B**

In cSCAN-B, the set of non-rule requests $\mathcal{Q}_N$ consists of all natural language commands that are generated by the phrase-structure grammar shown on the top left in Figure 5, which is equivalent to the original SCAN phrase structure grammar, as reformulated in Nye et al. (2020). The mapping from command to action sequence varies from context to context.

cSCAN-B constructs action sequences from the same 6 actions used in the original SCAN task, plus several additional ones provided for diversity. (In a nod to earlier research on the SCAN task, we follow the lead of Nye et al. (2020) in using as these additional actions the ones that appear in the "MiniSCAN" task of Lake et al. (2019).)

$$\mathcal{A} := \{\text{WALK,RUN,JUMP,LOOK,LTURN,RTURN,}$$
$$\text{RED,YELLOW,GREEN,BLUE,PURPLE,PINK,BLACK,WHITE}\}$$

**cSCAN-X**

In cSCAN-X, the set of non-rule requests $\mathcal{Q}_N$ consists of all natural language commands that are generated by the phrase-structure grammar shown on the top left in Figure 6. The mapping from command to action sequence varies from context to context, similarly to cSCAN-B.

cSCAN-X constructs action sequences from a set of 13 possible actions.

$$\mathcal{A} := \{\text{WALK,RUN,JUMP,LOOK,LTURN,RTURN,}$$
$$\text{DRIVE,RIDE,FLY,LEAP,PEEK,UTURN,DTURN}\}$$

E.2 REQUESTS AND REPLIES

For simplicity, this section focuses on a formal description of cSCAN-B. The specification of cSCAN-X would follow the same form, with the exception of the differences described in Appendix E.1 above.

**Non-rule requests.** In cSCAN, the set of non-rule requests $\mathcal{Q}_N$ consists of all natural language commands that are generated by the phrase-structure grammar shown on the top left in Figure 5. Note that for convenience of generation, the grammar adopted here is based on the alternative formulation of the SCAN grammar from Nye et al. (2020). This generates a slightly larger set of commands than the original SCAN grammar from Lake & Baroni (2017), as it includes commands such as "turn" and "turn and turn".

**Replies.** Since cSCAN is a CLT, the set of replies $\mathcal{R}$ includes the dedicated replies **1** (true), **0** (false), and **?** (unknown). In addition, $\mathcal{R}$ contains all sequences consisting of actions $\mathcal{A}$ (described in Appendix E.2 above) separated by space with a maximum sequence length $N$:

| S → T and T | T → U twice | U → V opposite X | V → W fast | W → walk | X → left | **Y → x1** |
| S → T after T | T → U thrice | U → V around X | **V → W cautiously** | W → run | X → right | **Y → x2** |
| **S → T before T** | **T → U 2x** | U → V X | **V → W drunkenly** | W → jump | **X → up** | **Y → x3** |
| **S → T following T** | **T → U 3x** | **U → V 90 X** | **V → W zigzag** | W → look | **X → down** | **Y → x4** |
| **S → T between T** | **T → U 4x** | **U → V 180 X** | V → W | W → turn | | |
| **S → T framing T** | **T → U 5x** | **U → V 270 X** | | **W → drive** | | |
| S → T | T → U | **U → V 360 X** | | **W → ride** | | |
| | | U → V | | **W → fly** | | |
| | | | | **W → leap** | | |
| | | | | **W → peek** | | |
| | | | | **W → Y** | **X → Y** | |

Figure 6: The green area on the top left is the phrase-structure grammar to generate the cSCAN-X commands. The red area on the lower right is the extension to generate the left-hand side of cSCAN-X rules. The items in blue boldface are those which do not appear in cSCAN-B.

$$\mathcal{A}^* := \{a_1\ a_2\ ...\ a_k : 1 \leq k < N, a_i \in \mathcal{A}\} \tag{30}$$

$$\mathcal{R} := \{\mathbf{1}, \mathbf{0}, \mathbf{?}\} \cup \mathcal{A}^* \tag{31}$$

**Rule requests.** All cSCAN rule requests are of the form $q = r$, where $q$ is an element of the left-hand side (LHS) expressions $\mathcal{Q}_N^*$ and $r$ is an element of the right-hand side (RHS) expressions $\mathcal{R}^*$.

The set of LHS expressions $\mathcal{Q}_N^*$ is generated by the extended phrase-structure grammar shown in Figure 5, with the resulting phrases wrapped in brackets $[\![\ ]\!]$. Note that this is almost the same grammar that is used to generate the cSCAN commands $Q_N$ with the only difference being that it allows generating expressions with variables from the set $\mathcal{X} := \{x1, x2, x3, x4\}$. Examples of LHS expressions are $[\![x_1\ \text{twice}]\!]$ and $[\![\text{walk opposite left}]\!]$.

The set of RHS expressions $\mathcal{R}^*$ consists of sequences of maximum length $N$ whose elements are separated by space and consist of actions $A$ and LHS expressions $\mathcal{Q}_N^*$. Examples of RHS expressions are "WALK $[\![x\ \text{twice}]\!]$ LTURN" and "WALK JUMP $[\![\text{run left}]\!]$ $[\![\text{look thrice}]\!]$". Note that RHS expressions are a superset of LHS expressions, i.e., $\mathcal{Q}_N^* \subset \mathcal{R}^*$.

$$\mathcal{R}^* := \{a_1\ a_2\ ...\ a_K : 1 \leq K < N, a_i \in \mathcal{A} \cup \mathcal{Q}_N^*\} \tag{32}$$

We assume a function $var : \mathcal{R}^* \to 2^{\mathcal{X}}$, which returns the set of variables used by a given RHS expression. This allows us to define the set of rule requests $\mathcal{Q}_R$ as an LHS and an RHS expression with matching variables:

$$\mathcal{Q}_R := \{q = r : q \in \mathcal{Q}_N^*, r \in \mathcal{R}^*, var(q) = var(r)\} \tag{33}$$

### E.3 GROUNDING OF RULES

As discussed in Appendix E.3, we specify the semantics of cSCAN rules via a grounding function $\mathcal{G} : \mathcal{Q}_R \to 2^{\mathcal{E}}$, which maps each rule to an equivalent set of examples. For rules where the LHS consists of a command without any variables, we can define the grounding as an example that provides the interpretation of this command. For example, the rule $[\![\text{walk}]\!] = \text{WALK}$ is grounded as follows:

$$\mathcal{G}([\![\text{walk}]\!] = \text{WALK}) := \{\langle \varnothing, \text{walk}, \text{WALK} \rangle\}$$

Note that because we determine consistency based on propositional logic equivalence (see Appendix D), we could specify equivalent groundings that include additional examples that are logically implied by $\{\langle \varnothing, \text{walk}, \text{WALK} \rangle\}$. For instance, we could add variants with non-empty contexts (e.g., $\langle \{[\![\text{run}]\!] = \text{RUN}\}, \text{walk}, \text{WALK} \rangle$), which follow from monotonicity.

For rules containing variables, the grounding may consist of hundreds or thousands of examples, even if we do not include any examples that are logically implied. This is because the variable can be replaced with any command that leads to a valid LHS. For example, the rule $[\![x_1 \text{ twice}]\!] = [\![x_1]\!][\![x_1]\!]$ can be grounded as follows:

$$\mathcal{G}([\![x_1 \text{ twice}]\!] = [\![x_1]\!][\![x_1]\!]) := \{ \langle \{[\![\text{run}]\!] = \text{RUN}\}, \text{run twice}, \text{RUN RUN} \rangle,$$
$$\langle \{[\![\text{run}]\!] = \text{JUMP}\}, \text{run twice}, \text{JUMP JUMP} \rangle,$$
$$\langle \{[\![\text{run left}]\!] = \text{LTURN RUN}\},$$
$$\text{run left twice}, \text{LTURN RUN LTURN RUN} \rangle,$$
$$...\}$$

The first example in this grounding can be read as: if $[\![\text{run}]\!]$ is translated to "RUN" then "run twice" is translated to "RUN RUN".

**Compositional grounding function of cSCAN.** Since the semantics of cSCAN rules is compositional (which is a requirement for all CLTs), we are able to specify the grounding function in a complete yet concise fashion using the following helper constructs. For simplicity, we focus here on describing the compositional grounding function for cSCAN-B. The grounding function for cSCAN-X would follow the same general form.

- For any RHS sequence $r \in \mathcal{R}^*$, we use $lhs(r)$ to denote the set of elements of $r$ that are LHS expressions $\mathcal{Q}_N^*$. For example, $lhs(\text{WALK JUMP }[\![\text{run left}]\!][\![\text{look thrice}]\!]) = \{[\![\text{run left}]\!], [\![\text{look thrice}]\!]\}$.

- For any subset of LHS expressions $Q \subseteq \mathcal{Q}_N^*$, we use $c2a(Q)$ to denote the set of all possible functions $f : Q \to \mathcal{A}^*$ that map each expression in $Q$ to an action sequence in $\mathcal{A}^*$. (Note that this also applies to the empty set, i.e., there is exactly one function in $c2a(\varnothing)$.)

- We define subsets of the commands $\mathcal{Q}_N$ generated by the phrase-structure grammar in Figure 5. For each $z \in \{T,U,V,W,X\}$ the set $\mathcal{Q}_N^z \subset \mathcal{Q}_N$ denotes the commands that are generated when starting from the symbol $z$ (rather than S).

- For each $z \in \{T,U,V,W,X\}$, we assume a function $var^z : \mathcal{Q}_N^* \to 2^{\mathcal{X}}$, which map each LHS expression $q \in \mathcal{Q}_N^*$ to a subset of its variables $var(q)$. For any expression $q \in \mathcal{Q}_N^*$, the following holds:
  - $var^T(q)$ consists of the variables that are generated using the rule path T $\to$ U, U $\to$ V, V $\to$ W, W $\to$ Y.
  - $var^U(q)$ consists of the variables that are generated using the rule path U $\to$ V, V $\to$ W, W $\to$ Y.
  - $var^V(q)$ consists of the variables that are generated using the rule path V $\to$ W, W $\to$ Y.
  - $var^W(q)$ consists of the variables that are generated using the rule W $\to$ Y.
  - $var^X(q)$ consists of the variables that are generated using the rule X $\to$ Y.

- For all LHS expressions $q \in \mathcal{Q}_N^*$ we use $v2c(q)$ to denote the set of all possible functions $f : var(q) \to \mathcal{Q}_N$ that map each variable in $var(q)$ to a command in $\mathcal{Q}_N$ such that $\forall z \in \{T,U,V,W,X\} : f(var^z(q)) \in \mathcal{Q}_N^z$. Note that this restriction of the mapping ensures that variable substitution does not break the implicit precedence rules that apply to the interpretation of commands (see the discussion below for more detail).

- For any RHS expression $r \in \mathcal{R}^*$ and any partial function $f : R^* \cup \mathcal{X} \nrightarrow \mathcal{R}^*$, we use $subst(r,f)$ to denote the RHS expression that we obtain by lexically substituting in $r$ all occurrences of $r' \in \text{dom}(f)$ with $f(r')$.

These constructs allow us to define the grounding of all rules $(q = r) \in \mathcal{Q}_R$ as follows:

$$\mathcal{G}(q = r) := \begin{cases} \cup_{f \in c2a(lhs(r))}\{\langle \cup_{q' \in \text{dom}(f)}\{\langle \varnothing, q' = f(q'), 1\rangle\}, q = subst(r,f), \mathbf{1}\rangle\}, & var(q) = \varnothing \\ \cup_{f \in v2c(q)}\{\langle \varnothing, subst(q,f), subst(r,f)\rangle\}, & \text{otherwise} \end{cases} \tag{34}$$

**Discussion and illustration.** The first case of definition 34 applies when the rule $q = r$ does not contain any variables. It says that whenever a LHS expression $q'$ occurs in the RHS expression $r$, we can substitute $q'$ with an arbitrary action sequence $f(q')$ as long as we make sure that the context contains a rule asserting that $q'$ is interpreted as $f(q')$.

As an illustration, consider the grounding of the rule $[\![\text{run twice and jump}]\!] = [\![\text{run}]\!][\![\text{run}]\!][\![\text{jump}]\!]\ \text{LTURN}$ that is shown below. The set of LHS expressions on the right-hand side of this rule is {run, jump}. The first example of the grounding can be read as: if $[\![\text{run}]\!]$ maps to "RUN" and $[\![\text{jump}]\!]$ maps to "JUMP" then the RHS of the rule becomes "RUN RUN JUMP LTURN".

$$\mathcal{G}([\![\text{run twice and jump}]\!] = [\![\text{run}]\!][\![\text{run}]\!][\![\text{jump}]\!]\ \text{LTURN}) =$$
$$\{\langle\{\langle\varnothing,[\![\text{run}]\!] = \text{RUN}, \mathbf{1}\rangle, \langle\varnothing,[\![\text{jump}]\!] = \text{JUMP}, \mathbf{1}\rangle\},$$
$$[\![\text{run twice and jump}]\!] = \text{RUN RUN JUMP LTURN}, \mathbf{1}\rangle,$$
$$\langle\{\langle\varnothing,[\![\text{run}]\!] = \text{LOOK}, \mathbf{1}\rangle, \langle\varnothing,[\![\text{jump}]\!] = \text{WALK}, \mathbf{1}\rangle\},$$
$$[\![\text{run twice and jump}]\!] = \text{LOOK LOOK WALK LTURN}, \mathbf{1}\rangle,$$
$$\langle\{\langle\varnothing,[\![\text{run}]\!] = \text{RUN}, \mathbf{1}\rangle, \langle\varnothing,[\![\text{jump}]\!] = \text{RUN RUN}, \mathbf{1}\rangle\},$$
$$[\![\text{run twice and jump}]\!] = \text{RUN RUN RUN RUN LTURN}, \mathbf{1}\rangle,$$
$$...\}$$

The second case of definition 34 applies when the rule $q = r$ contains a set of variables. It says that we can substitute any variable $x$ with a command $f(x)$ that corresponds to the same non-terminal in the parse tree.

As an illustration, consider the grounding of the rule $[\![x\ \text{twice}]\!] = [\![x]\!][\![x]\!]\ \text{LTURN}$ shown below. In the first example of the grounding, we replace the variable $x$ with the command "run", in the second example we replace it with "jump around left", and in the last example we replace it with "look thrice".

$$\mathcal{G}([\![x_1\ \text{twice}]\!] = [\![x_1]\!][\![x_1]\!]\ \text{LTURN}) =$$
$$\{\langle\varnothing,[\![\text{run twice}]\!] = [\![\text{run}]\!][\![\text{run}]\!]\ \text{LTURN}, \mathbf{1}\rangle,$$
$$\langle\varnothing,[\![\text{jump around left twice}]\!] = [\![\text{jump around left}]\!][\![\text{jump around left}]\!]\ \text{LTURN}, \mathbf{1}\rangle,$$
$$\langle\varnothing,[\![\text{look thrice twice}]\!] = [\![\text{look thrice}]\!][\![\text{look thrice}]\!]\ \text{LTURN}, \mathbf{1}\rangle,$$
$$...\}$$

Note that because $x_1 \in var^U([\![x_1\ \text{twice}]\!])$ and $x_1 \notin var^T([\![x_1\ \text{twice}]\!])$, the variable $x_1$ cannot be substituted with commands such as "walk and jump" that are in $Q_N^T$ but not in $Q_N^U$. This is important because the grounding would otherwise contain examples that violate the higher precedence of "twice" when compared to "and", which is not what we intended. I.e., our formalization makes sure that:

$$\langle\varnothing,[\![\text{walk and jump twice}]\!] = [\![\text{walk and jump}]\!][\![\text{walk and jump}]\!]\ \text{LTURN}, \mathbf{1}\rangle \notin$$
$$\mathcal{G}([\![x_1\ \text{twice}]\!] = [\![x_1]\!][\![x_1]\!]\ \text{LTURN})$$

## E.4 INDUCTIVE BIAS

As discussed in Appendix E.4, the inductive bias of a CLT is the criteria based upon which a learner is expected to induce a rule to be "true" as opposed to considering it to be "unknown". For cSCAN, we adopt a simple set of criteria, based loosely on our own intuition, in which we consider there to be sufficient evidence to support induction of a rule if there are examples in the context that could be explained by the given rule (in combination with other rules that are explicitly or implicitly provided in the context) and which in total illustrate at least 4 different substitutions of each of the rule's variables. The number 4 is arbitrary, but based on our intuition that we would be comfortable in generalizing rules from a relatively small number of examples, but that 2 examples is not quite enough to justify inducing a general pattern. We chose the threshold of 4 rather than 3 because we use the inductive bias criteria internally for determining the minimum number of illustrating examples to include in the context for each rule that we intend to be induced to be "true", and we wanted to avoid penalizing a learner that is slightly cautious in its inductions.

At the same time, we took care to avoid illustrating any rules' variable substitutions exactly three times, so as to avoid penalizing a learner that is just slightly on the aggressive side in its inductions. In this way, while we describe the task's inductive bias formally as requiring a minimum of 4 illustrated variable substitutions, a learner could succeed on the cSCAN task by adopting a minimum threshold of either 3 or 4 substitutions.

We formalize this domain-specific inductive bias of cSCAN with the bias function $\mathcal{B} \colon 2^{\mathcal{E}} \to 2^{\mathcal{E}_D}$. To avoid circularity, we assume a variation $\vdash^*$ of the logic embedding defined in Appendix D.4 with the only difference being that $\vdash^*$ does not use axiom (18), which depends on the inductive bias. Similarly, we use $\hookrightarrow^*$ to denote the variant of the minimal implication $\hookrightarrow$ that is based on $\to^*$ (rather than $\to$).

As we mentioned in Section 3.2, we expect the learner to induce a rule if it has been illustrated with a sufficient number of variable substitutions. To formalize this, we define the function $num_{subst} \colon \mathcal{X} \times Q_R \times 2^{\mathcal{Q}} \to \mathbb{N}$ such that $num_{subst}(x,q,Q)$ is the number of different expressions that can be substituted for the variable $x$ in the rule $q$ to obtain a rule in $Q$ (35). We also define the function $min_{subst} \colon Q_R \times 2^{\mathcal{Q}} \to \mathbb{N}$ such that $min_{subst}(q,Q)$ is the minimum number of such substitutions for any variable used in $q$ (36).

$$num_{subst}(x,q,Q) := |\{f(x) \in \mathcal{Q}_N \colon (\exists f \in \mathcal{X} \rightarrowtail \mathcal{R}^* \colon subst(q,f) \in Q)\}| \tag{35}$$

$$min_{subst}(q,Q) := \begin{cases} min_{x \in var(q)}(num_{subst}(x,q,Q)), & var(q) > 0 \\ 0, & \text{otherwise} \end{cases} \tag{36}$$

For any $E \subseteq \mathcal{E}$, the bias function $\mathcal{B}(E)$ is then defined as follows:

$$\mathcal{Q}_1(E,C) := \cup\{Q \subseteq \mathcal{Q}_R \colon (\exists e \in E \colon (E \setminus \{e\}) \cup \{\langle C,q,\mathbf{1}\rangle \colon q \in Q\} \hookrightarrow^* e)\} \tag{37}$$

$$\mathcal{B}_1(E) := \{\langle C,q,\mathbf{1},\mathbf{D}\rangle \in \mathcal{E} \colon q \in \mathcal{Q}_1(E,C) \wedge min_{subst}(q,\mathcal{Q}_1(E,C)) \geq 4\} \tag{38}$$

$$\mathcal{B}_?(E) := \{\langle C,q,\mathbf{?},\mathbf{D}\rangle \in \mathcal{E} \colon (\forall r \in \{\mathbf{1},\mathbf{0}\} \colon \nvdash^* \mathcal{M}(E \cup \mathcal{B}_1(E)) \to \mathcal{M}(\langle C,q,r,\mathbf{D}\rangle))\} \tag{39}$$

$$\mathcal{B}(E) := \mathcal{B}_1(E) \cup \mathcal{B}_?(E) \tag{40}$$

The set $\mathcal{Q}_1(E,C)$ is the set of candidate rules with context $C$ that we may want to induce from the example set $E$. Specifically, it is the union of all sets of rules $Q$ that, together with $E \setminus \{e\}$, allow us to (minimally) explain some example $e \in E$ (37).

The set $\mathcal{B}_1(E)$ contains the assertions of all rules that are expected to be induced to be true based on the examples provided by $E$. Specifically, it consists of the assertions $\langle C,q,\mathbf{1},\mathbf{D}\rangle$ of all the candidate rules $q$ for which $\mathcal{Q}_1(E,C)$ contains instances with at least 4 different substitutions for each variable (38).

$\mathcal{B}_?(E)$ contains the assertions of all rules for which $E$ does not provide any evidence about whether or not they hold. Specifically, it contains an example $\langle C,q,\mathbf{?},\mathbf{D}\rangle$ for all contexts $C$ and rules $q$ for which $E$ does not imply a clear answer, i.e., $\mathbf{1}$ or $\mathbf{0}$ (39). Finally, $\mathcal{B}(E)$ is the union of all the rules that should be induced to be true and all the rules that should be induced to be unknown (40).

**Discussion.** Because all cSCAN contexts exclusively contain unconditional, monotonic examples, it is sufficient to induce rules with the empty context, i.e., we only need to consider $\mathcal{Q}_1(E,\varnothing)$.

The consistency criteria in $\mathcal{B}_1$ is relatively simplistic, which makes the task easier for humans. Indeed, we only need to make sure that each induced rule $q$ is consistent with $E$ but do not need to check whether the induced rules are consistent with each other. This allows us to greedily induce rules one at a time without worrying about potential conflicts among them.

# F  DATASET GENERATION

The dataset generation process is config-driven, with a different dataset spec config being defined for each of the cSCAN datasets.

All cSCAN examples are automatically generated by a Python program built on the NLTK library (Bird et al., 2009).[2] Generation is performed via the following steps.

---

[2]To be released on GitHub upon paper acceptance.

**Context generation**     For efficiency, and to aid in generating clusters of related examples for calculation of the consistency metric, we generate examples in batches, in which we first generate a context and then generate multiple top-level examples that share that same context. Each context is created by first randomly generating a coherent set of interpretation rules of similar form to those shown in Figure 4 for the original SCAN task. While many of these rules may never be shown directly to the learner, this initial rule set serves as a kind of "basis rule set" from which the truth value of all other possible rules can be derived. We then randomly choose which of those basis rules to (a) provide explicitly in the context via an example that asserts that rule to be "true", (b) illustrate implicitly through some set of context examples sufficient to satisfy the task's inductive bias (see Appendix E.4 for details), (c) illustrate insufficiently or not at all, or (d) contradict in one or more cases, so that it should be inferrable to be "false".

By generating contexts via the above procedure, we ensure that the basis rules cover each of the different possible replies and qualifiers for rule examples: monotonically "true" (case a), defeasibly "true" (case b), "unknown" (case c), and either monotonically or defeasibly "false" (case d). By extension, this also ensures that we achieve a diverse mixture of possible replies and qualifiers across the much larger set of rules that could be derived from different combinations of those basis rules.

We ensure that the exact ratio between the different above cases varies randomly from context to context, while achieving across the dataset as a whole the desired ratio that is configured in the dataset spec.

**Top-level example generation**     Once we have fixed a context, we then randomly generate a set of (request, reply, qualifier) triples corresponding to the top-level examples that we wish to generate using the given context. To aid in sampling such examples, we first construct a pair of inference engines in which we exhaustively generate all possible examples that would be considered "true" either monotonically or defeasibly (respectively) based on the given context. To generate a non-rule example or positive rule example, we then randomly sample a "true" example from the full set of examples that were inferred by one of those inference engines. To construct a negative rule example, we first sample a positive example and then apply one of a number of different heuristics to construct an example that is similar to that positive example, but which is not among the examples inferred by the inference engine.

These two inference engines encapsulate the logic needed to ensure that each top-level example satisfies the consistency criteria stated in Section 2.3 and agrees with the domain-specific inductive bias.

**Sub-sampling**     In order to ensure that the different example classes are evenly represented in the dataset, we perform the top-level example generation process described above in separate streams, each of which is dedicated to generating one specific example class (e.g., "positive monotonic rule examples" or "non-rule examples with reply of unknown", etc.). We then sub-sample examples from each of the different streams in order to achieve the desired ratio of examples from each of the different classes. We ensure that the exact ratio between the different classes varies randomly from context to context, while matching the desired ratio in the dataset overall.

**Splitting**     Due to interdependencies between the splitting algorithm and the example generation process (see, e.g., the notes on "additional top-level example generation" below), we perform the train-validation-test split as one step of the dataset generation process, rather than generating a single dataset and then splitting it in multiple ways.

For performing MCD splits, we build on the open-sourced Python implementation of the MCD algorithm from Shaw et al. (2020). While their version of the MCD algorithm consists of initially performing a random split and then iteratively swapping examples to increase compound divergence, however, we found that we were able to achieve higher compound divergence on the cSCAN dataset by implementing an algorithm closer to that of the original one described in Keysers et al. (2020). In this approach, we begin with empty train and test sets and then iteratively select examples from a large example pool to add to one of the two sets, while once in every three steps selecting an example from one of the two sets to remove and put back in the pool. At each addition or removal step, we select from among a random sample of 200 examples the one whose addition or removal would maximize divergence at that stage, while keeping atom divergence low. One of the advantages of the insertion/deletion approach is that, in cases where it is acceptable to use only a portion of the available examples, the process can be stopped early, which can result in train and test splits with significantly higher compound divergence than would be possible if the algorithm were constrained to use all of the examples from the example pool. As described in Section 3.3, we do perform such sub-sampling when constructing our MCD splits, beginning with a pool of 1.2M top-level examples and ending with 100K top-level examples in train and 10K top-level examples in each of validation and test.

While our above algorithm largely emulates the one described in Keysers et al. (2020), we do introduce one additional enhancement to enable generating a 3-way compound divergence split between train, validation and test. This is in contrast to Keysers et al. (2020), which performs only a single stage of MCD splitting for maximizing compound divergence between train and test. In our approach, we perform two stages of MCD splitting, with the goal of constructing a train set, validation set, and test set, with high pairwise compound divergence between any two of the three. In the first stage, we split the example pool into a train+validation pool and a test set, with the objective of maximizing the compound divergence between the two. In the second stage, we keep the test set fixed, while further splitting the train+validation pool into a train set and a validation set, with the joint objective of maximizing compound divergence between train and validation and maximizing compound divergence between train and test. We perform down-sampling in each of the two stages to increase the compound divergences that we are able to achieve. Appendix H contain statistics on the MCD splits, along with other details of the datasets.

**Additional top-level example generation**    For the cSCAN Random variants, after performing a random split, we augment the validation and test sets by generating additional top-level examples for each validation and test context, using the same example generation logic described above. This allows us to achieve a higher density of logically related examples in the validation and test sets, so as to yield a larger number of potential implications and contradictions for use in calculating the consistency metric. We do not perform this step for the cSCAN MCD variants, however, to avoid impacting the compound divergences between the train, validation and test sets. For this reason, we focus our investigation of consistency in this paper on the cSCAN random splits, as it is only in the random splits that we are able to identify a significant number of implications and contradictions. Improving the sampling and splitting algorithms to enable investigation of consistency in MCD splits could be a topic for future work.

## G    CONSISTENCY METRIC CALCULATION

While the consistency metric as described in Appendix D in its most general form can be prohibitively expensive to calculate if approached naively, a close approximation of it can be calculated efficiently through the application of several task-specific assumptions.

First of all, in the consistency metric calculation we use with cSCAN, we consider for simplicity only implications and contradictions among predictions for top-level examples that share the same context. While in general implications and contradictions can occur even among examples with different contexts (particularly if the examples are monotonic and if one context is a superset of the other), due to the way in which we construct the cSCAN dataset, such situations are extremely unlikely to occur. By focusing only on identifying implications and contradictions among examples sharing the same context, we are able to cleanly partition the dataset into independent clusters of examples, such that we can analyze each cluster efficiently in parallel. Dealing with clusters of examples that share the same context also simplifies analysis in that we now only need to consider the implications and contradictions among the request-reply pairs, while effectively ignoring the context.

As the cSCAN validation and test sets contain up to 1000 top-level examples per context, however, it would still be prohibitively expensive to enumerate each of the possible subsets of these examples to identify the sets that involve a "minimal" implication or contradiction. Instead, we find that we are able to identify the implications and contradictions much more efficiently by seeding an inference engine with the rule assertions that would correspond to the up to 1000 top-level request-reply pairs (ignoring negative rule replies and unknown replies for simplicity) and then performing exhaustive forward inference to determine all possible rules that could be inferred from combinations of these asserted rules. This is essentially the same inference process that is used in top-level example generation (as described in Appendix F), except that we take the additional step of tracking the provenance of each inferred rule, and we continue the inference process so as to generate all possible provenances of each rule (rather than omitting reprocessing of rules that have already been inferred via a different route). For the purposes of this consistency calculation, it is sufficient to consider as provenance the set of asserted rules that led to the given inference. Once the exhaustive inference process is complete, we then check each of the asserted rules (i.e., each of the top-level predictions) against the contents of the inference engine. If the asserted rule was also inferred from some other rules, then we take the full set of inference provenances for that rule, filter out any inference provenances that are supersets of some other provenance, and then treat each of those remaining provenances together with the asserted rule as one "minimal implication" (i.e., as one minimal implication set $F \in impl(E)$ as defined in Equation 24). Similarly, if a rule was inferred that shares the same left-hand side as the asserted but contains a different right-hand side, then we look at

Table 6: Atom and compound divergence of the cSCAN MCD datasets between pairs of splits.

|  | train vs test | | train vs validation | | validation vs test | |
|---|---|---|---|---|---|---|
|  | Atom | Compound | Atom | Compound | Atom | Compound |
| cSCAN-B MCD | 0.066 | 0.685 | 0.039 | 0.634 | 0.030 | 0.562 |
| cSCAN-X MCD | 0.050 | 0.722 | 0.046 | 0.726 | 0.067 | 0.724 |

Table 7: Atom and compound counts of the cSCAN MCD datasets. The "held out" column records the number of compounds that appear in the test split but not in the train split.

|  | Min. occurrence of atom | | | # Compounds | | | | held out |
|---|---|---|---|---|---|---|---|---|
|  | train | valid | test | train | valid | test | all | held out |
| cSCAN-B MCD | 4180 | 1110 | 1440 | 88 | 91 | 118 | 138 | 47 |
| cSCAN-X MCD | 755 | 92 | 455 | 729 | 758 | 1125 | 1754 | 752 |

the provenances of each such inferred rule, filter out any that are supersets of some other provenance of the same rule, and then treat each of those remaining provenances together with the asserted rule as one "minimal contradiction" (i.e., as one minimal implication set $F \in cont(E)$ as defined in Equation 25).

Despite the relatively large number of rules with which we seed each inference engine and the extra expense of tracking multiple rule provenances, we find that we are able to complete exhaustive inference quickly in practice, due to the fact that the majority of the rules that are asserted in top-level examples tend to be more specific than the rules that are typically asserted inside of a context, and thus lead to only a limited number of interactions, which ends up being comparable to or less than the computational cost of the inference involved in constructing the contexts in the first place. In practice, when calculating the consistency metric in parallel using a different work unit for each context, we find that we are able to calculate the consistency metric for a full cSCAN experiment within a few minutes.

## H  DATASET DETAILS

Table 6 shows atom divergence and compound divergence between pairs of splits.

Table 7 shows the minimal number of times an atom occurs in each split, the number of compounds in each split, and the number of compounds in the test split that are held out from the train split.

Table 8 shows details of all splits of all datasets used in the cSCAN experiments.

Table 8: Key statistics of the cSCAN datasets.

| | cSCAN-B Random | | | cSCAN-X Random | | |
|---|---|---|---|---|---|---|
| | train | valid | test | train | valid | test |
| Number of examples | 100,000 | 100,000 | 100,000 | 100,000 | 99,756 | 100,000 |
| Number of contexts | 1,000 | 100 | 100 | 1,000 | 100 | 100 |
| Request: Rule (%) | 50.1 | 50.0 | 50.0 | 50.1 | 49.9 | 50.0 |
| Request: Non-rule (%) | 49.9 | 50.0 | 50.0 | 49.9 | 50.1 | 50.0 |
| Qualifier: Monotonic (%) | 46.3 | 45.5 | 46.0 | 50.0 | 50.0 | 48.0 |
| Qualifier: Defeasible (%) | 53.7 | 54.5 | 54.0 | 50.0 | 50.0 | 52.0 |
| Reply: Unknown (%) | 13.3 | 13.3 | 13.1 | 13.2 | 12.9 | 13.6 |
| Reply: Yes (%) | 21.8 | 21.3 | 21.8 | 21.9 | 22.2 | 21.5 |
| Reply: No (%) | 21.7 | 22.1 | 21.9 | 21.7 | 21.4 | 22.1 |

| | cSCAN-B MCD | | | cSCAN-X MCD | | |
|---|---|---|---|---|---|---|
| | train | valid | test | train | valid | test |
| Number of examples | 100,000 | 10,000 | 10,000 | 99,999 | 10,000 | 10,000 |
| Number of contexts | 11,921 | 6,115 | 6,509 | 7,965 | 4,627 | 4,599 |
| Request: Rule (%) | 72.5 | 76.5 | 64.0 | 59.8 | 73.8 | 67.4 |
| Request: Non-rule (%) | 27.5 | 23.5 | 36.0 | 40.2 | 26.2 | 32.6 |
| Qualifier: Monotonic (%) | 26.9 | 19.0 | 36.6 | 37.0 | 29.9 | 38.9 |
| Qualifier: Defeasible (%) | 73.1 | 81.0 | 63.4 | 63.0 | 70.1 | 61.1 |
| Reply: Unknown (%) | 52.2 | 67.2 | 35.2 | 39.7 | 50.6 | 36.3 |
| Reply: Yes (%) | 12.0 | 7.2 | 18.1 | 12.7 | 14.3 | 18.5 |
| Reply: No (%) | 12.1 | 7.8 | 18.2 | 12.9 | 13.7 | 19.2 |

| | cSCAN-B 100 Contexts | | | cSCAN-B 8000 Contexts | | |
|---|---|---|---|---|---|---|
| | train | valid | test | train | valid | test |
| Number of examples | 10,000 | 100,000 | 100,000 | 800,000 | 99,713 | 100,000 |
| Number of contexts | 100 | 100 | 100 | 8,000 | 100 | 100 |
| Request: Rule (%) | 50.1 | 50.1 | 50.1 | 50.1 | 49.9 | 50.1 |
| Request: Non-rule (%) | 49.9 | 49.9 | 49.9 | 49.9 | 50.1 | 49.9 |
| Qualifier: Monotonic (%) | 44.6 | 45.6 | 45.5 | 46.1 | 45.5 | 44.9 |
| Qualifier: Defeasible (%) | 55.4 | 54.4 | 54.5 | 53.9 | 54.5 | 55.1 |
| Reply: Unknown (%) | 12.8 | 13.0 | 13.6 | 13.4 | 14.2 | 13.6 |
| Reply: Yes (%) | 22.1 | 21.9 | 21.1 | 21.8 | 21.4 | 21.1 |
| Reply: No (%) | 21.8 | 21.7 | 22.3 | 21.6 | 21.6 | 22.2 |

# I  ANALYSIS OF REALISTIC CSCAN EXAMPLES

In this appendix, we provide realistic examples from a slightly earlier version of the cSCAN dataset and outline a systematic strategy for solving them. We selected a total of 22 examples (Figure 8) that are based on a single context of size 23 (Figure 7). We made sure that the different classes (see Section 2.4) are well represented among the selected examples. This means that the examples cover all valid combinations of request types, replies, and qualifiers.

To illustrate how cSCAN can be solved by humans, we outline one possible strategy below. This strategy is based on the assumption that we are aware of the grammar (Appendix E.2 and Figure 5), the rule semantics (Appendix E.3), and the inductive bias (Appendix E.4) of cSCAN.

## I.1  MONOTONIC EXAMPLES

In a first step, we check whether a given example can be deduced directly from the context. This is the case for all requests that exclusively contain syntactic constructs for which the behavior is completely determined by explicit rules in the context. In our context $C$, the rules (C1) through (C11) completely determine the behavior of all constructs except "look", "thrice", and "and". This means that we can deduce the reply for the examples (E1), (E2), (E3), (E4), (E5), and (E7), which we consequently mark as monotonic.

As an illustration, consider the request "jump around left twice" from example (E1). Once we know that $[\![jump]\!] = \text{JUMP}$ (C3), $[\![left]\!] = \text{PURPLE}$ (C1), $[\![x1 \text{ around } x2]\!] = [\![x1]\!][\![x2]\!][\![x1]\!]$ (C8), and $[\![x1 \text{ twice}]\!] = [\![x1]\!]$ (C10), we can immediately deduce that "jump around left twice" is translated to "JUMP PURPLE JUMP". Note that some of the examples are closely related to one another. For example, the example (E1) is an instance of the rule $[\![x1 \text{ around } x2 \text{ twice}]\!] = [\![x1]\!][\![x2]\!][\![x1]\!]$ from example (E5).

The examples (E6), (E8), and (E9) contain some of the syntactic constructs that are not completely determined (i.e., "look", "thrice", and "and"), but they can still be deduced from the context and are thus marked as monotonic. The example $\langle C, [\![look \text{ left}]\!] = \text{WHITE WHITE PURPLE}, \mathbf{1}, \mathbf{M}\rangle$ (E6) follows directly from the rules $[\![left]\!] = \text{PURPLE}$ (C1) and $[\![look \text{ } x1]\!] = \text{WHITE WHITE } [\![x1]\!]$ (C12).

Similarly, the example $\langle C, [\![look]\!] = \text{WHITE YELLOW}, \mathbf{0}, \mathbf{M}\rangle$ (E8) can be deduced from the examples (C7), and (C12). Indeed, if $[\![look]\!] = \text{WHITE YELLOW}$ were true, (C7) would allow us to deduce the rule $[\![look \text{ } x1]\!] = \text{WHITE YELLOW WHITE YELLOW } [\![x1]\!]$, which contradicts (C12).

Finally, (E9) can be deduced from (C2), (C8), and (C14) because $[\![x1 \text{ thrice}]\!] = [\![x1]\!][\![x1]\!]$ would imply that $[\![x1 \text{ around right thrice}]\!] = [\![x1]\!] \text{ GREEN } [\![x1]\!][\![x1]\!] \text{ GREEN } [\![x1]\!]$, which contradicts (C14).

## I.2  DEFEASIBLE EXAMPLES

To determine the reply of examples that cannot be deduced from the context, we check whether the context allows us to induce some rules for the constructs "look", "thrice", and "and", which are not fully determined.

**Inducing rules for "look", "thrice", and "and".** We start with the primitive command "look" and identify all context examples that contain "look" but none of the other partially-determined constructs "thrice" and "and". This yields the rules (C12) and (C13). Together with (C7), the rule (C12) tells us that "look" must either be "WHITE" or undefined (i.e., **?**), and the same holds for (C13). While these two examples alone are not sufficient under the task's inductive bias (Appendix E.4) to justify inducing the general form of the "look" rule, we will proceed with the assumption that $[\![look]\!] = \text{WHITE}$ for the time being.

Next, we apply the same process to the syntactic construct "thrice". This means that we identify all context examples that contain "thrice" but not "look" and "and", which are (C14), (C15), and (C16). The first two of them tell us that "$[\![x1 \text{ thrice}]\!]$" must either translate to "$[\![x1]\!][\![x1]\!][\![x1]\!]$" or be undefined. However, example (C16) is incompatible with the rule $[\![x1 \text{ thrice}]\!] = [\![x1]\!][\![x1]\!][\![x1]\!]$. In accordance with the inductive bias, we therefore cannot induce a general rule for "thrice".

Now, we apply the process to "and", which means that we look at the rules (C17) and (C18). Together with the example (C10), they both tell us that "$[\![x1 \text{ and } x2]\!]$" must either translate to "$[\![x2]\!][\![x2]\!][\![x1]\!]$" or be undefined. So, we proceed with the assumption that $[\![x1 \text{ and } x2]\!] = [\![x2]\!][\![x2]\!][\![x1]\!]$.

As a next step, we look at the examples (C19) and (C20) whose requests contain both "and" and "look" but not "thrice". Both of them follow from our current assumptions $[\![look]\!] = \text{WHITE}$ and $[\![x1 \text{ and } x2]\!] = [\![x2]\!][\![x2]\!][\![x1]\!]$. This means that we have now found four examples in the context that agree with our

(C1)     ⟨⟦left⟧ = PURPLE, **1**⟩
(C2)     ⟨⟦right⟧ = GREEN, **1**⟩
(C3)     ⟨⟦jump⟧ = JUMP, **1**⟩
(C4)     ⟨⟦run⟧ = LTURN, **1**⟩
(C5)     ⟨⟦turn⟧ = YELLOW, **1**⟩
(C6)     ⟨⟦walk⟧ = RUN, **1**⟩
(C7)     ⟨⟦x1 x2⟧ = ⟦x1⟧ ⟦x1⟧ ⟦x2⟧, **1**⟩
(C8)     ⟨⟦x1 around x2⟧ = ⟦x1⟧ ⟦x2⟧ ⟦x1⟧, **1**⟩
(C9)     ⟨⟦x1 opposite x2⟧ = ⟦x1⟧ ⟦x1⟧ ⟦x2⟧, **1**⟩
(C10)    ⟨⟦x1 twice⟧ = ⟦x1⟧, **1**⟩
(C11)    ⟨⟦x1 after x2⟧ = ⟦x2⟧ ⟦x1⟧ ⟦x2⟧ ⟦x1⟧, **1**⟩

(C12)    ⟨⟦look x1⟧ = WHITE WHITE ⟦x1⟧, **1**⟩
(C13)    ⟨⟦look opposite x1⟧ = WHITE WHITE ⟦x1⟧, **1**⟩

(C14)    ⟨⟦x1 around right thrice⟧ = ⟦x1⟧ GREEN ⟦x1⟧ ⟦x1⟧ GREEN ⟦x1⟧ ⟦x1⟧ GREEN ⟦x1⟧, **1**⟩
(C15)    ⟨⟦x1 opposite right thrice⟧ = ⟦x1⟧ ⟦x1⟧ GREEN ⟦x1⟧ ⟦x1⟧ GREEN ⟦x1⟧ ⟦x1⟧ GREEN, **1**⟩
(C16)    ⟨run right twice after walk thrice, RUN RUN LTURN LTURN GREEN RUN RUN LTURN LTURN GREEN⟩

(C17)    ⟨⟦x1 and x2 twice⟧ = ⟦x2⟧ ⟦x2⟧ ⟦x1⟧, **1**⟩
(C18)    ⟨⟦x1 twice and x2⟧ = ⟦x2⟧ ⟦x2⟧ ⟦x1⟧, **1**⟩

(C19)    ⟨jump around right and look around left, WHITE PURPLE WHITE WHITE PURPLE WHITE JUMP GREEN JUMP⟩
(C20)    ⟨jump opposite left twice and look around left, WHITE PURPLE WHITE WHITE PURPLE WHITE JUMP JUMP PURPLE⟩

(C21)    ⟨look left thrice, WHITE WHITE PURPLE WHITE WHITE PURPLE WHITE WHITE PURPLE⟩
(C22)    ⟨walk twice and jump left thrice, JUMP JUMP PURPLE JUMP JUMP PURPLE RUN⟩
(C23)    ⟨turn left thrice and run, LTURN LTURN YELLOW YELLOW PURPLE YELLOW YELLOW PURPLE YELLOW⟩

Figure 7: A context *C* from the cSCAN dataset. The examples that make up the context are sorted, starting with 11 explicit rules that specify the complete behavior of all syntactic constructs except "look", "thrice", and "and". The remaining examples are sorted such that we first have the examples that do not contain "thrice" and "and", then the examples that do not contain "and", and finally the remaining examples.

**Monotonic (non-rules)**
(E1)     ⟨C, jump around left twice, JUMP PURPLE JUMP, **M**⟩
(E2)     ⟨C, walk around right after run opposite left twice, LTURN LTURN PURPLE RUN GREEN RUN LTURN LTURN PURPLE RUN GREEN RUN, **M**⟩
(E3)     ⟨C, run left and jump around right twice, JUMP GREEN JUMP JUMP GREEN JUMP LTURN LTURN PURPLE, **M**⟩

**Monotonic (rules)**
(E4)     ⟨C, ⟦x1 opposite left⟧ = ⟦x1⟧ ⟦x1⟧ PURPLE, **1**, **M**⟩
(E5)     ⟨C, ⟦x1 around x2 twice⟧ = ⟦x1⟧ ⟦x2⟧ ⟦x1⟧, **1**, **M**⟩
(E6)     ⟨C, ⟦look left⟧ = WHITE WHITE PURPLE, **1**, **M**⟩
(E7)     ⟨C, ⟦run around x1⟧ = LTURN ⟦x1⟧ ⟦x1⟧ LTURN, **0**, **M**⟩
(E8)     ⟨C, ⟦look⟧ = WHITE YELLOW, **0**, **M**⟩
(E9)     ⟨C, ⟦x1 thrice⟧ = ⟦x1⟧ ⟦x1⟧, **0**, **M**⟩

**Defeasible (non-rules)**
(E10)    ⟨C, look opposite left twice, WHITE WHITE PURPLE, **D**⟩
(E11)    ⟨C, run right and look around right, WHITE GREEN WHITE WHITE GREEN WHITE LTURN LTURN GREEN, **D**⟩
(E12)    ⟨C, look left twice after look, WHITE WHITE WHITE PURPLE WHITE WHITE WHITE PURPLE, **D**⟩
(E13)    ⟨C, run left thrice, **?**, **D**⟩
(E14)    ⟨C, run opposite right and jump around left thrice, **?**, **D**⟩
(E15)    ⟨C, walk around right thrice after turn around left twice, **?**, **D**⟩

**Defeasible (rules)**
(E16)    ⟨C, ⟦look⟧ = WHITE, **1**, **D**⟩
(E17)    ⟨C, ⟦look around x1⟧ = WHITE ⟦x1⟧ WHITE, **1**, **D**⟩
(E18)    ⟨C, ⟦x1 and x2 thrice⟧ = ⟦x1⟧ ⟦x2⟧ ⟦x2⟧ ⟦x1⟧, **1**, **D**⟩
(E19)    ⟨C, ⟦look opposite x1 thrice⟧ = WHITE WHITE, **0**, **D**⟩
(E20)    ⟨C, ⟦x1 thrice⟧ = ⟦x1⟧ ⟦x1⟧ ⟦x1⟧, **?**, **D**⟩
(E21)    ⟨C, ⟦x1 after x2 thrice⟧ = ⟦x2⟧ ⟦x2⟧ ⟦x2⟧ ⟦x1⟧ ⟦x2⟧ ⟦x2⟧ ⟦x2⟧ ⟦x1⟧, **?**, **D**⟩
(E22)    ⟨C, ⟦x1 around x2 thrice and x3 twice⟧ = ⟦x3⟧ ⟦x3⟧ ⟦x1⟧ ⟦x2⟧ ⟦x1⟧ YELLOW ⟦x1⟧ ⟦x2⟧ ⟦x1⟧, **?**, **D**⟩

Figure 8: Examples from the cSCAN dataset. The examples are based on context *C* shown in Figure 7 above, and they are grouped by request type and qualifier.

assumed rules for "look" and "and", and there are no examples that contradict them. Based on our inductive bias, this means that we induce these rules.

All the remaining examples (i.e., (C21), (C22), and (C23)) contain the construct "thrice", for which we cannot induce a general rule.

**Applying the induced rules for "look" and "and".** Using the induced rules for "look" and "and", we can determine a concrete reply for the examples (E10) through (E12) as well as (E16) through (E19). These examples are marked as defeasible because they are based (among others) on induced rules, which means that the reply might change if the context is expanded.

**Using reply "unknown" for examples containing "thrice".** Because we are not able to induce a generic rule for "thrice", we are not able to determine a concrete reply for the examples (E13) through (E14) and (E20) through (E22). Instead we use the reply "unknown" (**?**) and mark them as defeasible because the reply might again change if the context is expanded.

## J   EFFECT OF DATA SIZE

As an additional set of experiments, we investigate the impact of training data size on the performance of the relatively strong pre-trained T5-Base (220M parameters) baseline. In each experiment, we use a dataset generated with a similar mix of examples as in cSCAN-B, but with a varying number of contexts and examples in the train set. As can be seen in the results in Table 9, while the model is able to achieve high accuracy and relatively high consistency as the train size approaches 800K examples, performance drops dramatically as the train decreases from 100K examples down to 10K examples. This suggests that finding ways to reduce models' dependency on large amounts of task-specific training data will be an additional important theme for future work in conceptual learning.

Table 9: Accuracy as a function of training data size. Results from experiments with pre-trained T5-Base (220M parameters) on datasets of similar form to cSCAN-B, but of varying size.

| Contexts | Examples | Accuracy | Consistency |
|---------|----------|----------|-------------|
| 100 | 10,000 | 38.6 | 6.9 |
| 1,000 | 100,000 | 92.6 | 71.3 |
| 8,000 | 800,000 | 97.7 | 91.9 |

## K   EXTENDED RELATED WORK

### K.1   COMPARISON WITH EXISTING DATASETS

In representing the input of a CLT as a request paired with a context, we build on a long tradition of QA and reasoning task formulations that provide knowledge relevant to a task via various forms of context, such as a text passage (Kwiatkowski et al., 2019; Weston et al., 2015; Dua et al., 2019; Sinha et al., 2019; Yang et al., 2018; Rajpurkar et al., 2016; Levesque et al., 2012), logical premise (Roemmele et al., 2011; Dagan et al., 2005; Bowman et al., 2015), set of natural language statements (Talmor et al., 2020), knowledge graph fragment (Sinha et al., 2020), antecedent description grammar (Cohen, 1994), dialog (Semantic Machines et al., 2020; Budzianowski et al., 2018), image (Antol et al., 2015; Johnson et al., 2017; Hudson & Manning, 2019; Bahdanau et al., 2019a;b), grid world (Ruis et al., 2020), or DB schema (Yu et al., 2018).

Here, to give a better sense of how a CLT is similar to and different from these existing task formulations, we make a closer comparison of cSCAN with several representative NLU tasks that provide explicit knowledge as part of the input and satisfy some of the desired properties for a CLT formulated in Section 2.1. This is illustrated in Figure 9.

Note that we focus our comparison here specifically on the defining features of a CLT. This should not be construed as a commentary on the overall usefulness or quality of these benchmarks. Indeed, many of the benchmarks described here have complementary strengths which cSCAN lacks, such as being based on true natural language, supporting multi-modal input, illustrating specific domains of reasoning, or covering more complex reasoning or syntax.

Talmor et al. present a series of "Leap-of-Thought" tasks (Talmor et al., 2020) where the learner needs to judge yes/no hypotheses by performing inference over knowledge that is obtained implicitly from language model pretraining while part of the knowledge is also provided explicitly using a context containing natural

| | Context (explicit knowledge) | | | | Request | Output | |
|---|---|---|---|---|---|---|---|
| | **Set-like** | **Variable** (incompatible across examples) | **Rules** (rules assertable) | **Examples** (request / reply pairs assertable) | **Rules** (rules requestable) | **Unknown** | **Deductive vs. Inductive** |
| Leap-of-Thought | ✅ | ❌ | ✅ | ✅ | ✅ | ❌ | ❌ |
| GraphLog | ✅ | ✅ | ❌ | ✅ | ❌ | ❌ | ❌ |
| bAbI | ❌ | ✅ | (✅) | ❌ | ❌ | ❌ | ✅ |
| gSCAN | ❌ | ✅ | (✅) | ❌ | ❌ | ❌ | ❌ |
| Natural Questions | ❌ | ❌ | (✅) | ❌ | ❌ | ❌ | ❌ |
| **cSCAN** | ✅ | ✅ | ✅ | ✅ | ✅ | ✅ | ✅ |

Figure 9: Comparison of cSCAN and other NLU task against the key features of conceptual learning. For the context (explicit knowledge), we distinguish whether it is organized as a set of independent units rather than a monolithic block (column 1) and whether different examples use different and sometimes contradictory knowledge (column 2). We also distinguish whether the context contains both rules (column 3) as well as examples (column 4). For the request, we indicate whether the task contains examples that explicitly test the truth value of rules (column 5). For the output, we indicate whether the task requires identifying whether a request cannot be answered based on the given context (column 6) and whether it distinguishes between inductive and deductive reasoning.

language statements. This setup satisfies various CLT properties. In particular, arbitrary hypotheses can be asserted in the context, and the requests can ask for the truth value of any context statement (examples and rules coincide in this task because all examples are yes/no hypotheses). However, the knowledge is constant across all examples (albeit different parts of this knowledge are provided explicitly for different examples). Therefore, Leap-of-Thought tasks do not investigate whether a learner is able to adapt to transient knowledge, e.g., explicit knowledge that may contradict the implicit knowledge obtained during language model pretraining (e.g., an actor was married and is now divorced). Similarly, these tasks exclusively focus on monotonic inference: they do not test whether the learner is able to induce defeasible hypotheses from the explicit knowledge nor do they test the ability to identify certain hypotheses as "unknown".

GraphLog (Sinha et al., 2020) is a benchmark suite based on tasks that are quite similar to CLTs. The learner is presented with part of a graph consisting of labeled edges (context) and then needs to predict a the label for an edge that is not part of the context. For example, the context may contain two "father-child" relations and the learner needs to predict the "grandfather-grandchild" relation. As for cSCAN, the graph is constructed automatically based on a set of first-order logic rules, which make sure that each task is consistent. However, in contrast to cSCAN, the underlying rules cannot be part of the context, nor are they expressible as requests. In the CLT terminology, this means that GraphLog tests only the case where the context consists of examples and the learner has to induce new examples. It does not support the case where the context contains rules and the learner has to consistently combine and apply these rules deductively.

The bAbI tasks (Weston et al., 2015) also require the learner to answer a question using variable context consisting of natural language statements. Each context consists of a sequence of relatively simple factual statements that are, at least for some of the tasks, order-dependent (Dehghani et al., 2018). This means that a bAbI context does not directly correspond to a set-based context that we are using with CLTs. One way to bridge this gap is to consider a sequence of bAbI statements as a single "macro rule", but the truth value of these rules cannot be requested. bAbI contains some tasks that require deductive reasoning and some tasks that require inductive reasoning.

The gSCAN task (Ruis et al., 2020) is an extension of SCAN where the learner is provided with a context that describes a spacial configuration of objects in order to translate commands into a sequences of actions. However, as with bAbI, the context consists of a single dedicated structure describing the spacial configuration as a whole rather than a set of rules that describe the different objects that are part of the spacial configuration one by one. This makes the language to specify the context disjoint from the request language.

As an example of a reading comprehension task, Natural Questions (Kwiatkowski et al., 2019) is a benchmark where the learner is given a Wikipedia page as context and then needs to answer a natural language question by outputting a long answer (e.g., the paragraph containing the answer) as well as a short answer. As with bAbI, the context is not a set of independent rules but instead a sequence of inter-dependent statements, which makes this benchmark quite different from a CLT.

## K.2 OTHER RELATED WORK

**Compositional generalization.** Our evaluation of cSCAN MCD splits builds on existing research in measuring the ability of machine learning models to compositionally generalize (Keysers et al., 2020; Lake & Baroni, 2017). In response to compositional generalization benchmarks such as SCAN, a range of techniques have been proposed which have not yet been evaluated on cSCAN. Some of these solutions involve specialized architectures for enforcing a compositional bias (Qiu et al., 2022; Chen et al., 2020; Liu et al., 2020; Nye et al., 2021). Such architectures are appealing due to their potential for achieving a principled solution to compositional generalization, but would require some effort to adapt to the context and heterogenous output of a CLT. In the past, some specialized architectures have shown limited success when transferring to new tasks, compared to more general techniques such as language model pre-training (Furrer et al., 2020). In the latter category, there is some promise shown by recent developments in decompositional prompting techniques, which have led to strong results on the SCAN MCD splits using off-the-shelf large language models (Zhou et al., 2022).

**Instruction following.** In evaluating the ability for a system to apply rules to a task, our work relates to research in building systems that learn to follow instructions (Goldwasser & Roth, 2014), including recent research in the instruction-following capabilities of large language models (Wei et al., 2022; Ouyang et al., 2022; Wang et al., 2022). Our approach differs in that we provide in the context a set of rules, each of which may be applicable to some part of the task, and we evaluate the ability to infer new rules in addition to applying the rules to an underlying task.

**Meta-learning.** Meta-learning or "learning to learn" generally refers to a setup where the learner is provided with a family of tasks, which are also called episodes, each of which comes with its own set of training examples and test examples. A learner is able to "learn to learn" if its performance for each task increases both with increasing training data and with an increasing number of tasks (Thrun & Pratt, 1998; Hospedales et al., 2021; Finn et al., 2017).

The presence of examples within the context of an example gives CLTs a nested structure that allows us to view CLTs through the lens of the meta-learning setup. In this view, top-level examples that share the same context correspond to an episode where the context examples are the training examples and the top-level examples (w/o the context) are the test examples.

Closely related to cSCAN are two pieces of work that apply meta-learning to SCAN, both of which generate large numbers of SCAN-like grammars, from which they construct meta-learning episodes, similarly to how we generate cSCAN contexts based on SCAN-like grammars. Lake (2019) uses a memory-augmented network to attend to the train examples of each episode, while Nye et al. (2020) trains for each episode a program synthesis model that outputs the underlying rules of the task. Our approach differs in that we include in the context a mixture of rules and examples, rather than just examples of the underlying task, and we use the synthetically-generated contexts to define a new task for evaluating the ability of the system to generalize to many different rule sets, rather than using meta-learning techniques as a means to improve accuracy on the original SCAN task.

**Rule induction and logic deduction tasks.** CLTs require the learner to judge whether a certain rule can be induced from a given set of observations (i.e., examples provided as part of the context). This is similar to a rule induction task (Cohen, 1995; Reddy & Tadepalli, 1998; Grzymala-Busse, 2010), with the main caveat that the learner only needs to verify rules rather than generate them. CLTs also require the learner to apply rules and judge whether a rule may be deductively obtained from a set of other rules.

**Interpretable ML models.** CLTs make part of the rules that govern the underlying tasks explicit, which allows us to "introspect" the behavior of the learner by asking, as part of the task, whether or not a certain rule holds (inductively or deductively). This means that the question of whether and why a certain model behaves correctly or incorrectly can be broken down into two parts: (a) did the model learn the right rules and (b) is it able to apply these rules consistently. This is related to, yet different from, other efforts to make ML models more interpretable. For example, Sushil et al. (Sushil et al., 2018) propose a method to

induce if-then-else rules to explain the behavior of ML models. However, unlike for CLTs, this method is external to the actual task: it does not reflect whether the model claims a certain rule to be true but instead identifies the if-then-else rules between different input features and class labels that are the most important for classification according to the model.

**Consistency.** Our consistency metric is related to research into evaluating and improving the consistency of neural networks. One closely related work is Li et al. (2019), which evaluates consistency by generating clusters of examples that by construction are related via logical symmetry, transitivity, or some other logical constraint. Based on these examples, they calculate a "conditional violation" metric, which similarly to our consistency metric is a ratio of violated (in our case, satisfied) constraints vs. the total number of logical constraints. Our approach differs in that we gather the logical constraints automatically using symbolic inference over the generated examples, rather than depending on a specific algorithm for generating related examples.

**Handling of large contexts.** Due to the potentially large context size in CLTs, another relevant area of research is how to deal with very large contexts, including large set-like contexts. One line of research in this area involves modifying the Transformer architecture to be able to handle longer inputs more efficiently (Tay et al., 2020a;b). We evaluated two such architecture variants in our LongT5 and LongT5-Global baselines (Guo et al., 2022), but many other such variants have been proposed (Gu et al., 2021; Zaheer et al., 2020; Choromanski et al., 2020; Wang et al., 2020). In cases where the context takes the form of a set or a graph, some approaches seek to explicitly take into account this structure by encoding the input structure in positional embeddings (Herzig et al., 2020), guiding the Transformer's attention via the structural relations within the input (Ainslie et al., 2020), or message-passing in graph neural networks (Gilmer et al., 2017; Battaglia et al., 2018). Other lines of research seek to more efficiently deal with large pools of potentially relevant knowledge by either performing cross-attention from portions of the input to knowledge stored in neural memory (Verga et al., 2020) or by retrieving only the most relevant material from a knowledge base or text corpus for concatenation to the input (Guu et al., 2020; Pasupat et al., 2021).

## L  REPRODUCIBILITY

**Hardware and training period**     Table-10 shows the different hardware used for each experiment and the training period (in steps).

**T5 versions**     For our T5 baselines, we use T5X (Roberts et al., 2022), which is a re-implementation of T5 in JAX (Bradbury et al., 2018) using Flax (Heek et al., 2020). Table 11 presents the configurations of different T5 variants that we used in our experiments. For the full-attention version of T5, we experiment with both fine-tuning from a standard pre-trained checkpoint and training from scratch.

For LongT5 and LongT5-TGlobal, while we initially evaluated with both fine-tuning from a standard pre-trained checkpoint and training from scratch, when fine-tuning, we failed to find a setup in which the models converge on the train set, possibly due to poor compatibility between the cSCAN task and the summarization-oriented PEGASUS Principle Sentences Generation pre-training objective (Zhang et al., 2019) used in LongT5. For this reason, we only report results on LongT5 and LongT5-TGlobal models trained from scratch.

For each of the architectures, we evaluate at minimum two sizes: Small (60M parameters) and Base (220M parameters). For the best-performing T5 architecture, we further evaluate on size Large (770M parameters). We omitted experiments on Large variants of the other architectures for reasons of computational cost, as the poor performance on the Small and Base sizes suggest that it is unlikely for the performance of LongT5, LongT5-TGlobal or the non-pretrained version of T5 to improve significantly with model size alone.

**Hyperparameters**     Table 12 summarizes the hyperparameters used for each of the baselines. The reasons for choosing those hyperparameters are:

- **Config:** We chose the config version based on experiments on an earlier version of the dataset. In that version, we found that T5 achieved higher performance using the T5.1.0 config, while LongT5 and LongT5-TGlobal performed better on the T5.1.1 config.

- **Learning Rate:** A constant learning rate is the standard way to fine-tune pre-trained T5 models. For non-pretrained models we found that a constant learning rate performed as well as the inverse square root decay with linear warmup scheduler (the standard learning rate scheduler for pretraining T5).

Table 10: Hardware and training period (in steps).

| Model | Size | Pretrain | Dataset | TPU Type | TPU Slices | Training Steps | Training Epochs |
|---|---|---|---|---|---|---|---|
| T5 | S | True | cSCAN-B | TPU V2 | 64 | 150,000 | 192 |
| T5 | B | True | cSCAN-B | TPU V3 | 16 | 50,000 | 64 |
| T5 | L | True | cSCAN-B | TPU V4 | 64 | 50,000 | 64 |
| T5 | S | False | cSCAN-B | TPU V2 | 64 | 300,000 | 384 |
| T5 | B | False | cSCAN-B | TPU V3 | 16 | 150,000 | 192 |
| LongT5 | S | False | cSCAN-B | TPU V2 | 64 | 300,000 | 384 |
| LongT5 | B | False | cSCAN-B | TPU V3 | 16 | 150,000 | 192 |
| LongT5-TGlobal | S | False | cSCAN-B | TPU V2 | 64 | 300,000 | 384 |
| LongT5-TGlobal | B | False | cSCAN-B | TPU V3 | 16 | 150,000 | 192 |
| T5 | S | True | cSCAN-X | TPU V2 | 64 | 150,000 | 192 |
| T5 | B | True | cSCAN-X | TPU V3 | 16 | 50,000 | 64 |
| T5 | L | True | cSCAN-X | TPU V4 | 128 | 50,000 | 64 |
| T5 | S | False | cSCAN-X | TPU V2 | 64 | 300,000 | 384 |
| T5 | B | False | cSCAN-X | TPU V3 | 16 | 150,000 | 192 |
| LongT5 | S | False | cSCAN-X | TPU V2 | 64 | 300,000 | 384 |
| LongT5 | B | False | cSCAN-X | TPU V3 | 16 | 150,000 | 192 |
| LongT5-TGlobal | S | False | cSCAN-X | TPU V2 | 64 | 300,000 | 384 |
| LongT5-TGlobal | B | False | cSCAN-X | TPU V3 | 16 | 150,000 | 192 |
| T5 | S | True | cSCAN-B MCD | TPU V2 | 64 | 150,000 | 192 |
| T5 | B | True | cSCAN-B MCD | TPU V3 | 16 | 50,000 | 64 |
| T5 | L | True | cSCAN-B MCD | TPU V4 | 64 | 50,000 | 64 |
| T5 | S | False | cSCAN-B MCD | TPU V2 | 64 | 300,000 | 384 |
| T5 | B | False | cSCAN-B MCD | TPU V3 | 16 | 150,000 | 192 |
| LongT5 | S | False | cSCAN-B MCD | TPU V2 | 64 | 300,000 | 384 |
| LongT5 | B | False | cSCAN-B MCD | TPU V3 | 16 | 150,000 | 192 |
| LongT5-TGlobal | S | False | cSCAN-B MCD | TPU V2 | 64 | 300,000 | 384 |
| LongT5-TGlobal | B | False | cSCAN-B MCD | TPU V3 | 16 | 150,000 | 192 |
| T5 | S | True | cSCAN-X MCD | TPU V2 | 64 | 150,000 | 192 |
| T5 | B | True | cSCAN-X MCD | TPU V3 | 16 | 50,000 | 64 |
| T5 | L | True | cSCAN-X MCD | TPU V4 | 64 | 50,000 | 64 |
| T5 | S | False | cSCAN-X MCD | TPU V2 | 64 | 300,000 | 384 |
| T5 | B | False | cSCAN-X MCD | TPU V3 | 16 | 150,000 | 192 |
| LongT5 | S | False | cSCAN-X MCD | TPU V2 | 64 | 300,000 | 384 |
| LongT5 | B | False | cSCAN-X MCD | TPU V3 | 16 | 150,000 | 192 |
| LongT5-TGlobal | S | False | cSCAN-X MCD | TPU V2 | 64 | 300,000 | 384 |
| LongT5-TGlobal | B | False | cSCAN-X MCD | TPU V3 | 16 | 150,000 | 192 |
| T5 | B | True | cSCAN-B 100 Contexts | TPU V3 | 16 | 100,000 | 1280 |
| T5 | B | True | cSCAN-B 5000 Contexts | TPU V3 | 16 | 100,000 | 25.6 |
| T5 | B | True | cSCAN-B 8000 Contexts | TPU V3 | 16 | 100,000 | 16 |

Table 11: Configurations of different T5 variants.

| Name | #Parameters | #Layers | $d_{model}$ | $d_{ff}$ | #heads |
|---|---|---|---|---|---|
| Small | 60M | 6 | 512 | 2048 | 8 |
| Base | 220M | 12 | 768 | 3072 | 12 |
| Large | 770M | 24 | 1024 | 4096 | 16 |

Table 12: Hyperparameters used in the cSCAN baselines.

| | T5 (Pretrained) | | | T5 (Non-Pretrained) | | LongT5 | | LongT5-TGlobal | |
| | Small | Base | Large | Small | Base | Small | Base | Small | Base |
|---|---|---|---|---|---|---|---|---|---|
| Config | T5.1.0 | T5.1.0 | T5.1.0 | T5.1.0 | T5.1.0 | T5.1.1 | T5.1.1 | T5.1.1 | T5.1.1 |
| Learning Rate Scheduler | Constant | Constant | Constant | Constant | Constant | Constant | Constant | Constant | Constant |
| Learning Rate | 0.001 | 0.001 | 0.001 | 0.001 | 0.001 | 0.001 | 0.001 | 0.001 | 0.001 |
| Batch Size | 128 | 128 | 128 | 128 | 128 | 128 | 128 | 128 | 128 |
| Loss Normalizing Factor | 233472 | 233472 | 233472 | 233472 | 233472 | 233472 | 233472 | 233472 | 233472 |
| Initial Checkpoint | 1000000 | 999900 | 1000700 | 0 | 0 | 0 | 0 | 0 | 0 |

- **Loss Normalizing Factor:** By instruction of T5 creators, we used a fixed loss normalizing factor of

$$PretrainingBatchSize(2048) \times TargetTokensLength(114) = 233472$$

for fine-tuning and we used the same value for non-pretrained models to maintain consistency.

Note that for the small models using T5.1.1 config, we set the number of layers and heads to 6 and 8 respectively to match those set in T5.1.0 config.

**Tokenization** All models use the pretrained SentencePiece tokenizer (Kudo & Richardson, 2018) provided by T5, which is pretrained to cover the English, French, German and Romanian languages with 32,000 tokens. We also tried using a simple whitespace tokenizer, which resulted in similar performance when compared to the pretrained T5 tokenizer.

Table 13: Breakdown of accuracy by example characteristics on cSCAN-B Random.

| Model | Pretrain | Size | Knownness | | | Request/Reply Type | | |
|---|---|---|---|---|---|---|---|---|
| | | | Defeasible | Monotonic | Unknown | Neg. Rule | Pos. Rule | Non-Rule |
| T5 | True | S | 90.2 | 93.9 | 97.4 | 79.8 | 96.5 | 95.8 |
| | | B | 90.5 | 93.9 | 97.7 | 78.8 | 96.9 | 96.2 |
| | | L | **95.3** | **97.7** | **97.7** | **91.9** | **98.1** | **97.6** |
| | False | S | 13.9 | 16.4 | 75.0 | 32.0 | 27.7 | 2.9 |
| | | B | 15.0 | 12.3 | 74.4 | 33.4 | 20.5 | 2.7 |
| LongT5-TGlobal | False | S | 16.3 | 15.9 | 74.1 | 28.8 | 34.2 | 3.0 |
| | | B | 13.9 | 13.1 | 68.7 | 29.5 | 24.0 | 2.2 |
| LongT5 | False | S | 17.1 | 14.0 | 73.8 | 31.2 | 29.5 | 3.0 |
| | | B | 16.1 | 17.8 | 80.6 | 33.1 | 32.6 | 3.9 |
| T5 w/o Context | True | B | 19.3 | 23.5 | 83.0 | 40.6 | 44.9 | 5.8 |

Table 14: Breakdown of accuracy by example characteristics on cSCAN-X Random.

| Model | Pretrain | Size | Knownness | | | Request/Reply Type | | |
|---|---|---|---|---|---|---|---|---|
| | | | Defeasible | Monotonic | Unknown | Neg. Rule | Pos. Rule | Non-Rule |
| T5 | True | S | 59.2 | 68.7 | 97.8 | 43.3 | 69.2 | 75.2 |
| | | B | 62.6 | 71.6 | **98.4** | 46.5 | 79.4 | 74.8 |
| | | L | **74.9** | **83.7** | 97.7 | **69.5** | **91.4** | **80.6** |
| | False | S | 10.3 | 16.4 | 63.0 | 29.0 | 25.1 | 1.9 |
| | | B | 11.2 | 14.8 | 72.7 | 29.5 | 22.6 | 2.2 |
| LongT5-TGlobal | False | S | 13.0 | 14.6 | 72.4 | 28.1 | 26.8 | 2.7 |
| | | B | 9.4 | 18.2 | 78.6 | 28.1 | 28.5 | 2.7 |
| LongT5 | False | S | 11.7 | 16.3 | 70.8 | 32.8 | 23.5 | 2.5 |
| | | B | 12.3 | 15.2 | 57.8 | 29.5 | 25.4 | 2.0 |
| T5 w/o Context | True | B | 17.0 | 20.6 | 83.3 | 35.5 | 39.9 | 4.0 |

## M  BREAKDOWN METRICS

### M.1  CSCAN-B RANDOM

Table 13 shows breakdown of accuracy by example characteristics on cSCAN-B Random. From the breakdowns for T5-Large (770M parameters), we can see that for a sufficiently large model with full attention, it is feasible to achieve high accuracy across example classes. Across the pre-trained T5 models, however, we can notice several trends:

- Examples with reply of "unknown" tend to be easier than other examples.

- Negative rule examples, i.e., those with reply of "false", are the most challenging class of example.

- Accuracy on non-rule examples is roughly similar to accuracy on rule examples, despite the fact that rule examples involve classification into a much smaller space of possible replies (which would make them more amenable to solving through guessing).

In contrast, when looking at the results of the models that failed to outperform the naive baseline, we can see that for these, accuracy on rule examples is significantly higher than that on non-rule examples, with the highest accuracy on examples with the reply of "unknown", consistent with the view that these models are relying on guessing based on superficial example characteristics.

Table 15: Breakdown of accuracy by example characteristics on cSCAN-B MCD.

| Model | Pretrain | Size | Knownness | | | Request/Reply Type | | |
|---|---|---|---|---|---|---|---|---|
| | | | Defeasible | Monotonic | Unknown | Neg. Rule | Pos. Rule | Non-Rule |
| T5 | True | S | 38.2 | 24.0 | 99.8 | 60.0 | 39.7 | 22.5 |
| | | B | 38.8 | 25.3 | **99.9** | 59.8 | 35.3 | 27.2 |
| | | L | **55.3** | **46.8** | 99.8 | **75.6** | **60.4** | **42.4** |
| | False | S | 17.9 | 14.3 | 96.6 | 25.8 | 30.3 | 8.9 |
| | | B | 13.8 | 14.7 | 97.4 | 26.7 | 24.2 | 6.2 |
| LongT5-TGlobal | False | S | 18.4 | 14.0 | 98.5 | 30.3 | 25.7 | 8.9 |
| | | B | 13.1 | 13.2 | 97.4 | 29.8 | 16.7 | 10.8 |
| LongT5 | False | S | 16.7 | 15.0 | 97.2 | 36.8 | 18.7 | 10.3 |
| | | B | 13.4 | 14.0 | 97.6 | 42.1 | 6.5 | 6.7 |
| T5 w/o Context | True | B | 25.2 | 19.5 | 96.6 | 38.0 | 40.2 | 14.6 |

## M.2 cSCAN-X RANDOM

Table 14 shows breakdown of accuracy by example characteristics on cSCAN-X Random. From the pre-trained T5 models, we can see the following trends:

- Examples with reply of "unknown" continue to be much easier than other examples. In fact, the models are able to answer these examples with comparably high accuracy on cSCAN-X as on cSCAN-B.

- Negative rule examples continue to be the most challenging class of example.

- While the models struggle with both rule examples and non-rule examples, accuracies are even lower on rule examples than on non-rule examples, again despite the fact that rule examples would be more amenable to random guessing.

For the models that failed to outperform the naive baseline, the pattern is similar on cSCAN-X as on cSCAN-B.

## M.3 cSCAN-B MCD

Table 15 shows breakdown of accuracy by example characteristics on cSCAN-B MCD. From these results, we can see the following trends:

- All models achieve particularly high accuracy on examples with reply "unknown", as would be expected from the dataset stats shown in Table 2, where we can see that this class of examples makes up over 50% of the examples in the cSCAN-B MCD train set. This makes the answer of "unknown" a natural guess in any situation where the model is unsure.

- Given that T5 w/o Context is able to achieve significantly higher than zero accuracy on rule examples with replies other than "unknown", however, it is clear that the model is doing more than simply predicting "unknown" every time. Rather, it appears that a moderate amount of statistical clues must be available in the request itself to allow some degree of "educated guessing" of the reply, particularly in the case of rule examples.

- For all models, accuracy on non-rule examples lags significantly behind accuracy on rule examples. This is in contrast to the cSCAN Random datasets, where the stronger-performing pre-trained models frequently performed better on non-rule examples than on rule examples. One reason for this difference is likely the fact that the train set for cSCAN-B MCD is skewed toward rule examples, which make up somewhat over 70% of the dataset. Taken in light of the observation above about the naive T5 w/o Context baseline, however, the poor performance on non-rule examples also suggests that the T5 baselines may be achieving even less proper "understanding" of the examples than one would have thought from looking at the overall accuracy numbers alone, and is likely relying to a large degree on "educated guessing", based on statistical clues from the

Table 16: Breakdown of accuracy by example characteristics on cSCAN-X MCD.

| Model | Pretrain | Size | Knownness | | | Request/Reply Type | | |
| | | | Defeasible | Monotonic | Unknown | Neg. Rule | Pos. Rule | Non-Rule |
|---|---|---|---|---|---|---|---|---|
| T5 | True | S | 51.6 | 57.1 | 99.5 | **65.0** | 37.8 | 66.6 |
| | | B | 50.3 | 54.2 | **99.7** | 14.3 | 83.1 | 66.0 |
| | | L | **60.0** | **67.4** | 99.7 | 36.2 | **88.6** | **73.4** |
| | False | S | 13.3 | 19.4 | 96.5 | 29.2 | 28.0 | 6.8 |
| | | B | 13.5 | 17.3 | 94.8 | 28.7 | 24.4 | 5.6 |
| LongT5-TGlobal | False | S | 14.6 | 17.3 | 96.4 | 28.1 | 26.5 | 6.4 |
| | | B | 14.4 | 17.8 | 95.5 | 30.6 | 24.8 | 6.5 |
| LongT5 | False | S | 13.2 | 19.0 | 94.9 | 32.8 | 23.5 | 6.3 |
| | | B | 14.7 | 18.4 | 95.1 | 32.2 | 24.8 | 6.8 |
| T5 w/o Context | True | B | 12.6 | 21.2 | 96.2 | 27.1 | 33.3 | 8.7 |

request and context, which is much easier to do on rule examples than on non-rule examples, due to the smaller space of possible replies for rule examples.

## M.4 cSCAN-X MCD

Table 16 shows breakdown of accuracy by example characteristics on cSCAN-X MCD. From these results, we can see the following trends:

- Similarly to cSCAN-B MCD, all models achieve high accuracy on examples with reply "unknown", which is again the most commonly occurring class of examples in this dataset (around 40% of examples in the train set).

- For pre-trained T5, however, accuracy on non-rule examples is significantly higher than on cSCAN-B MCD, suggesting that these models are likely benefiting from the more balanced distribution of examples in the cSCAN-X MCD dataset, where around 40% of the train examples are non-rule examples, compared with less than 30% in cSCAN-B MCD.

- The large gap in accuracy between negative and possible rule examples on pre-trained T5 suggests that while they are able to use information from the context to do a better job than the naive T5 w/o Context at distinguishing between "unknown" and "not unknown" rules, they are still relying largely on guessing for determining the rules' actual truth value.

## M.5 EFFECT OF EXAMPLE AND CONTEXT CHARACTERISTICS

Figures 10, 11, 12, and 13 show T5's performance on the test splits of cSCAN-B, cSCAN-X, cSCAN-B MCD, and cSCAN-X MCD datasets with respect to various features of the examples and the contexts, broken down into rule and non-rule examples.

The features being considered are:

- **num_rules**: The number of *distinct* rules used to create an example. For example, for the request "walk and walk", num_rules is 2: it is created with the rules: $[\![x_1 \text{ and } x_2]\!] = ...$ and $[\![\text{walk}]\!] = ...$

- **num_variables**: The number of variables in the rule. For example, for the request "walk and $x_1$", num_variables is 1.

- **derivation_level**: The number of compositions used to build an example. For example, for the request "walk and walk", derivation_level is 2: it is created by first composing $[\![x_1 \text{ and } x_2]\!] = ...$ with $[\![\text{walk}]\!] = ...$ to get $[\![\text{walk and } x_2]\!] = ...$, followed by another composition with $[\![\text{walk}]\!] = ...$.

- **frac_explicit_rules_bucket**: The fraction of explicit rules among all distinct rules used to create an example. The fractions are bucketed for legibility: frac_explicit_rules_bucket=0.5 includes all examples with a fraction of explicit rules at least 0.5 and less than 0.6.

- **input_length_bucket**: The length of the input (context + request) in tokens. The lengths are bucketed for legibility: input_length_bucket=500 includes all examples with length at least 500 and less than 600.

- **context_num_explicit_rules**: The number of rules explicitly asserted in the context. Every context is based on 14 grammar rules, so for example, context_num_explicit_rules=5 means that the context contains explicit assertions of 5 of these rules, while the other 9 rules are either illustrated indirectly via examples (such that the learner is expected to induce the rule to be true) or are not illustrated sufficiently (such that the learner is expected to consider the rule to be either false or unknown).

In all cases the accuracy appears negatively correlated num_rules and derivation_level. See text below each figure for additional observations.

## N    FINE-GRAINED EVALUATION METRICS

In addition to exact match accuracy, for more nuanced error analysis, we track several additional finer-grained metrics, including partial accuracy metrics, edit distance, and counts of implications and contradictions related to the consistency metric.

### N.1    PARTIAL ACCURACY METRICS

Sequence level accuracy is a hard metric where a single wrong token leads the entire prediction to be labeled as wrong. For this reason we define additional accuracy measures that take into account partial success in solving the actual task. These measures are:

- **Reply Accuracy:** A prediction is considered correct if the reply portion is correct.

- **Qualifier Accuracy:** A prediction is considered correct if the qualifier portion is correct.

- **Pattern Accuracy:** Pattern accuracy assigns each token an incremental ID based on the order it appears in the sequence, thus ignoring the specific predicted tokens and focusing on token variation pattern. E.g. the sequence *JUMP JUMP RUN JUMP* and *WALK WALK EAT WALK* both have the same pattern of *A A B A* where *A* replaces JUMP and WALK in the first and second sequences respectively, while *B* replaces RUN and EAT in the first and second sequence respectively.

- **Naive Accuracy:** A prediction is considered correct if it produces the same set of unique tokens as the target regardless of the order or the count. E.g. the sequences *JUMP JUMP RUN JUMP* and *RUN JUMP* both have the same unique tokens set (*JUMP* and *RUN*).

- **Token Accuracy:** Token-wise accuracy between the prediction and the target. The two sequences are aligned at the start token, and the shorter sequence is padded to have the same length as the longer sequence such that the padded tokens are considered wrong predictions.

Tables 17, 18, 20, and 20 show the performance of each baseline on all of these metrics.

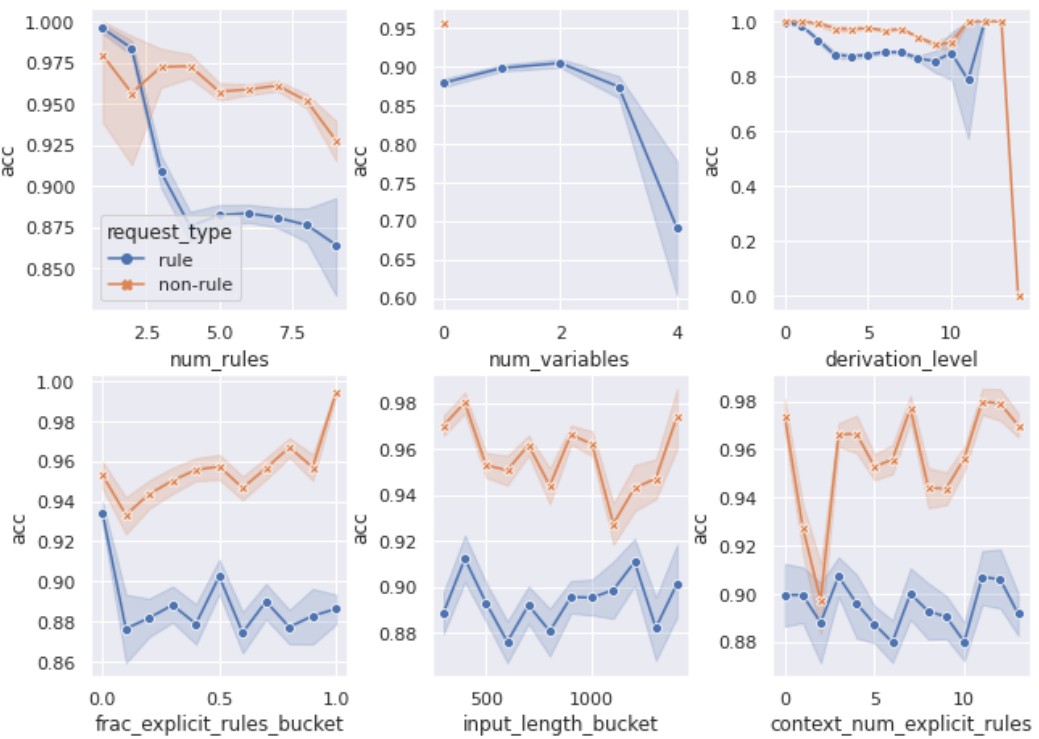

Figure 10: T5 accuracy on cSCAN-B examples. For non-rule examples, the accuracy appears positively correlated with the fraction of explicit rules (bottom-left).

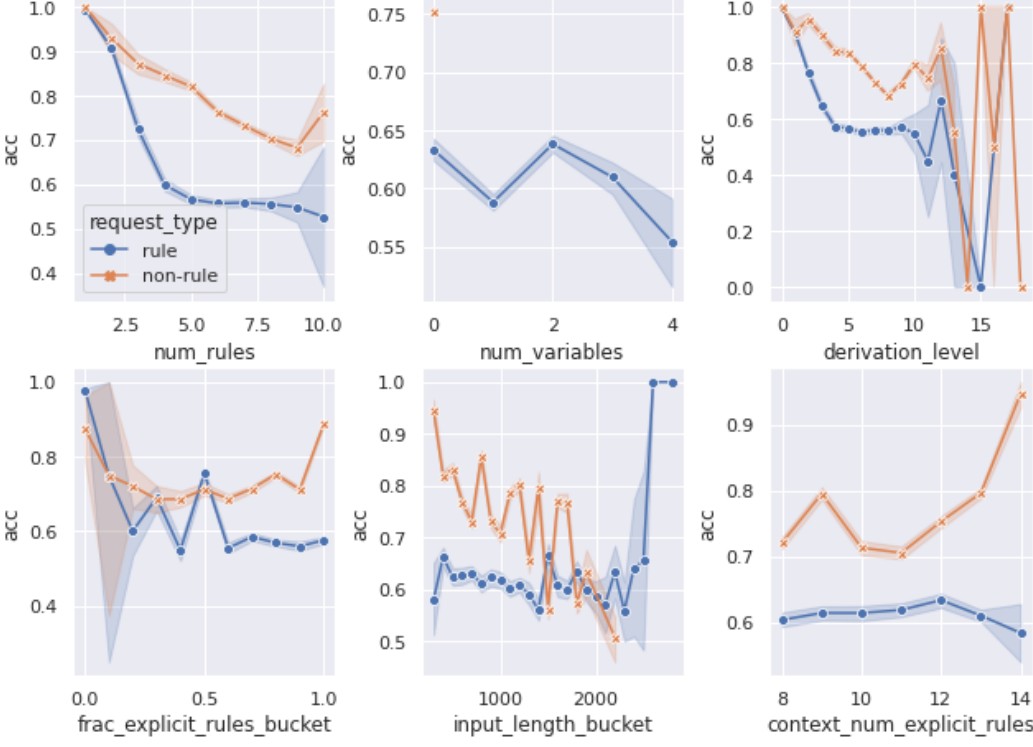

Figure 11: T5 accuracy on cSCAN-X examples. For non-rule examples, the accuracy appears positively correlated with the number of explicit examples in the context (bottom-right), and negatively correlated with the input length (bottom-center).

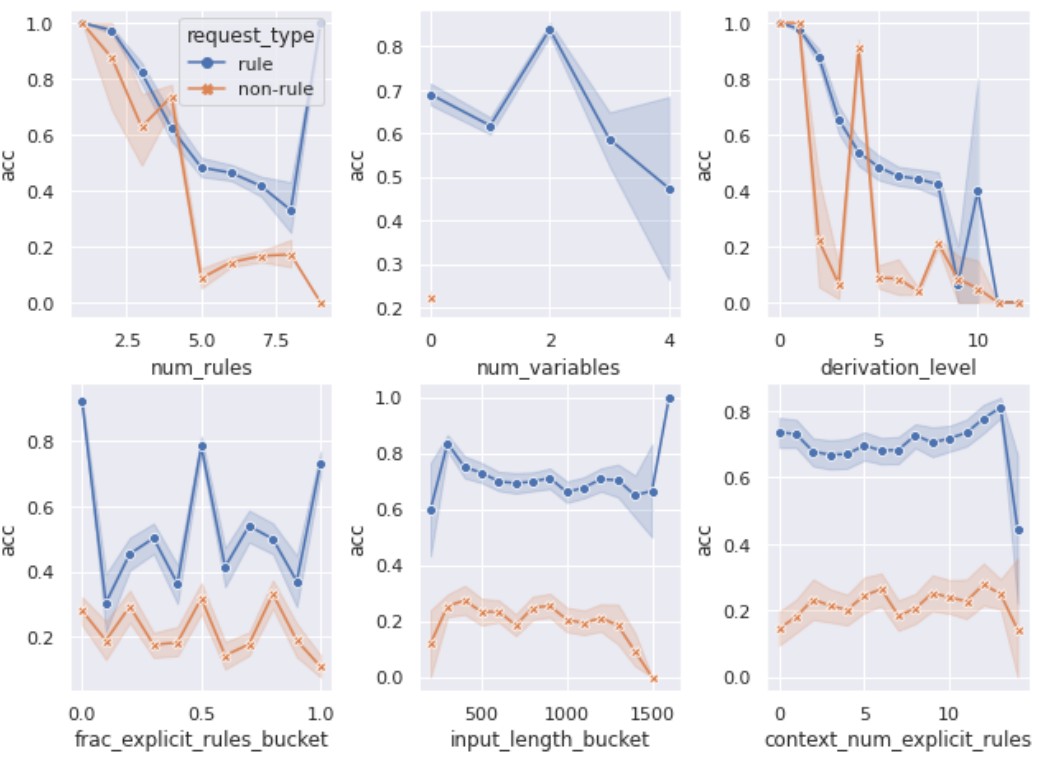

Figure 12: T5 accuracy on cSCAN-B MCD examples.

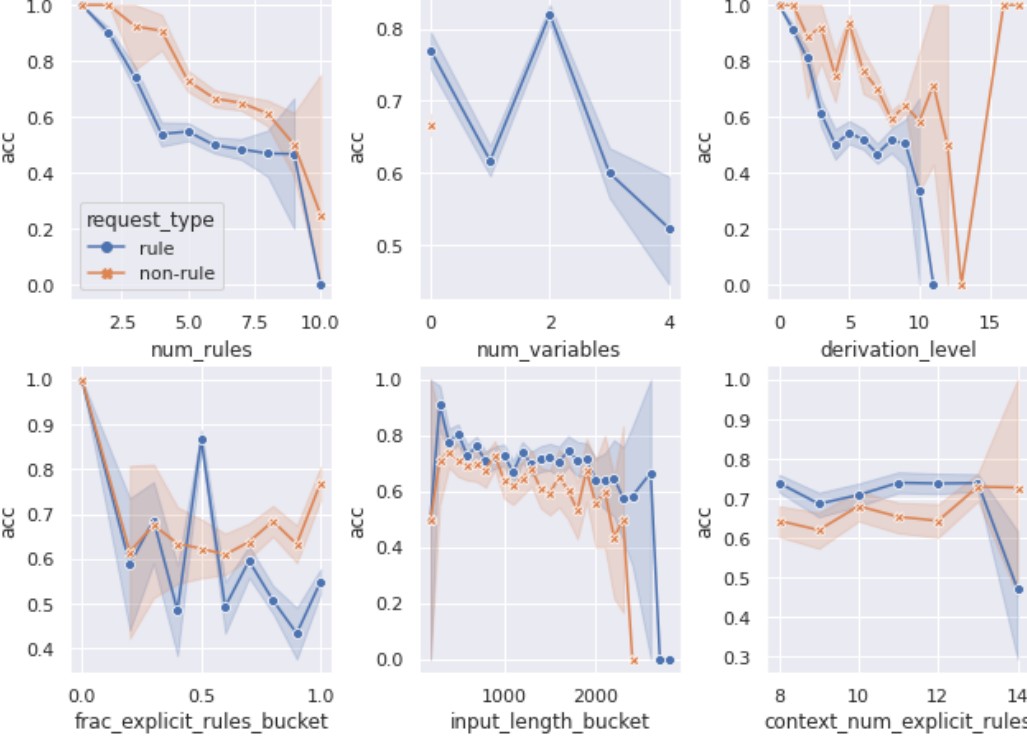

Figure 13: T5 accuracy on cSCAN-X MCD examples. For both rule and non-rule examples, the accuracy appears negatively correlated with the input length (bottom-center).

Table 17: Partial accuracy measures on cSCAN-B Random

| | | | | Partial Accuracy | | | | |
|---|---|---|---|---|---|---|---|---|
| Baseline | Pretrain | Size | Accuracy | Reply | Qualifier | Pattern | Naive | Token |
| T5 | True | *S* | 92.5 | 93.2 | 99.2 | 98.2 | 93.2 | 96.9 |
| T5 | True | *B* | 92.6 | 93.3 | 99.1 | 98.4 | 93.3 | 97.2 |
| T5 | True | *L* | **96.5** | **97.1** | **99.2** | **99.1** | **96.8** | **98.5** |
| T5 | False | *S* | 19.3 | 28.4 | 61.1 | 51.8 | 38.9 | 27.5 |
| T5 | False | *B* | 17.9 | 27.9 | 55.0 | 51.7 | 18.5 | 17.7 |
| LongT5 | False | *S* | 19.5 | 28.4 | 62.2 | 51.9 | 33.2 | 26.7 |
| LongT5 | False | *B* | 21.4 | 29.9 | 65.6 | 52.7 | 39.2 | 30.7 |
| LongT5-TGlobal | False | *S* | 20.0 | 28.7 | 62.9 | 51.8 | 43.2 | 29.5 |
| LongT5-TGlobal | False | *B* | 17.2 | 27.2 | 54.5 | 51.5 | 18.0 | 16.7 |
| T5 w/o Context | True | *B* | 26.8 | 43.8 | 54.8 | 53.3 | 26.9 | 15.5 |

Table 18: Partial accuracy measures on cSCAN-X Random

| | | | | Partial Accuracy | | | | |
|---|---|---|---|---|---|---|---|---|
| Baseline | Pretrain | Size | Accuracy | Reply | Qualifier | Pattern | Naive | Token |
| T5 | True | *S* | 68.3 | 70.1 | 96.4 | 88.5 | 77.2 | 85.7 |
| T5 | True | *B* | 71.1 | 72.7 | 96.4 | 88.1 | 79.9 | 85.0 |
| T5 | True | *L* | **81.6** | **83.6** | **97.1** | **91.2** | **88.5** | **90.2** |
| T5 | False | *S* | 16.8 | 26.8 | 53.0 | 51.0 | 20.1 | 10.5 |
| T5 | False | *B* | 17.2 | 27.1 | 52.9 | 51.1 | 19.8 | 8.6 |
| LongT5 | False | *S* | 18.1 | 27.7 | 55.8 | 51.3 | 22.9 | 11.4 |
| LongT5 | False | *B* | 16.7 | 26.6 | 53.8 | 51.1 | 20.0 | 9.4 |
| LongT5-TGlobal | False | *S* | 18.0 | 27.6 | 55.6 | 51.4 | 21.9 | 11.1 |
| LongT5-TGlobal | False | *B* | 18.8 | 28.2 | 54.8 | 51.3 | 22.0 | 8.8 |
| T5 w/o Context | True | *B* | 23.8 | 38.7 | 52.0 | 52.1 | 23.9 | 4.4 |

Table 19: Partial accuracy measures on cSCAN-B MCD

| | | | | Partial Accuracy | | | | |
|---|---|---|---|---|---|---|---|---|
| Baseline | Pretrain | Size | Accuracy | Reply | Qualifier | Pattern | Naive | Token |
| T5 | True | *S* | 53.8 | 62.0 | 77.4 | 73.8 | 67.4 | 47.0 |
| T5 | True | *B* | 54.7 | 62.9 | 78.4 | 76.0 | 67.2 | 51.0 |
| T5 | True | *L* | **67.6** | **76.7** | **83.2** | **82.7** | **76.9** | **65.4** |
| T5 | False | *S* | 40.1 | 47.7 | 68.4 | 67.4 | 50.4 | 31.8 |
| T5 | False | *B* | 38.4 | 45.7 | 64.7 | 66.0 | 39.2 | 24.7 |
| LongT5 | False | *S* | 40.7 | 47.6 | 68.5 | 68.2 | 48.5 | 31.0 |
| LongT5 | False | *B* | 38.2 | 45.2 | 67.5 | 66.8 | 45.2 | 29.6 |
| LongT5-TGlobal | False | *S* | 40.6 | 48.0 | 68.0 | 67.6 | 52.1 | 33.6 |
| LongT5-TGlobal | False | *B* | 39.3 | 45.9 | 66.9 | 68.0 | 45.6 | 30.2 |
| T5 w/o Context | True | *B* | 46.2 | 60.0 | 65.2 | 69.4 | 46.2 | 22.9 |

Table 20: Partial accuracy measures on cSCAN-X MCD

| Baseline | Pretrain | Size | Accuracy | Partial Accuracy | | | | |
| | | | | Reply | Qualifier | Pattern | Naive | Token |
|---|---|---|---|---|---|---|---|---|
| T5 | True | *S* | 70.7 | 72.8 | 95.5 | 90.1 | 78.3 | 83.2 |
| T5 | True | *B* | 69.2 | 71.0 | 95.8 | 89.8 | 76.9 | 81.4 |
| T5 | True | *L* | **76.8** | **78.4** | **97.0** | **92.2** | **82.9** | **87.2** |
| T5 | False | *S* | 41.6 | 48.7 | 69.5 | 69.7 | 51.4 | 18.7 |
| T5 | False | *B* | 40.0 | 47.6 | 66.5 | 69.0 | 42.4 | 12.3 |
| LongT5 | False | *S* | 40.9 | 48.2 | 68.1 | 69.5 | 49.9 | 17.9 |
| LongT5 | False | *B* | 41.2 | 48.5 | 68.0 | 69.7 | 46.1 | 14.3 |
| LongT5-TGlobal | False | *S* | 41.0 | 48.4 | 67.7 | 69.5 | 49.2 | 18.0 |
| LongT5-TGlobal | False | *B* | 40.9 | 48.1 | 67.8 | 69.5 | 44.8 | 12.5 |
| T5 w/o Context | True | *B* | 42.7 | 52.3 | 64.4 | 70.3 | 42.8 | 6.2 |

Table 21: Consistency breakdown for cSCAN-B Random

| Baseline | Pretrain | Size | Consistency | Implications | Contradictions |
|---|---|---|---|---|---|
| T5 | True | $S$ | 71.8 | 431 | 169 |
| T5 | True | $B$ | 71.3 | 437 | 176 |
| T5 | True | $L$ | **87.5** | **446** | **64** |
| T5 | False | $S$ | 0.0 | 0 | 881 |
| T5 | False | $B$ | 0.0 | 0 | 2107 |
| LongT5 | False | $S$ | 0.0 | 0 | 1419 |
| LongT5 | False | $B$ | 0.1 | 1 | 689 |
| LongT5-TGlobal | False | $S$ | 0.2 | 3 | 1269 |
| LongT5-TGlobal | False | $B$ | 0.0 | 1 | 3128 |
| T5 w/o Context | True | $B$ | 0.1 | 1 | 774 |

Table 22: Consistency breakdown for cSCAN-X Random

| Baseline | Pretrain | Size | Consistency | Implications | Contradictions |
|---|---|---|---|---|---|
| T5 | True | $S$ | 30.1 | 237 | 550 |
| T5 | True | $B$ | 32.3 | 268 | 561 |
| T5 | True | $L$ | **43.5** | **335** | **435** |
| T5 | False | $S$ | 0.1 | 4 | 3967 |
| T5 | False | $B$ | 0.0 | 2 | 5375 |
| LongT5 | False | $S$ | 0.1 | 2 | 1791 |
| LongT5 | False | $B$ | 0.1 | 4 | 3678 |
| LongT5-TGlobal | False | $S$ | 0.2 | 3 | 1651 |
| LongT5-TGlobal | False | $B$ | 0.1 | 3 | 3002 |
| T5 w/o Context | True | $B$ | 0.4 | 3 | 822 |

## N.2 CONSISTENCY BREAKDOWN BY IMPLICATIONS AND CONTRADICTIONS

In this section we show the number of implications and contradictions used to calculate the consistency metric. Tables 21 and 22 show this breakdown.

To further illustrate the type of inconsistencies the model is making, we show the consistency sets (implications and contradictions) size distribution in Figures 14 and 15. This shows that sets of size 2 where one prediction implies or contradicts another are the most common. Appendix P.2 expands on this by providing examples of the contradictions.

As can be seen in these tables, the consistency metric for each cSCAN Random experiment is calculated based on a minimum of 500 implications and contradictions.

As discussed in Section 6.2 and Appendix F, we do not report consistency metrics for the MCD datasets, as we are only able to achieve a high enough density of potential implications in the Random datasets, where after splitting, we augment each context with additional top-level examples from the same distribution. We do not perform this additional example generation step for the MCD datasets to avoid impacting the compound divergences between the train and test sets.

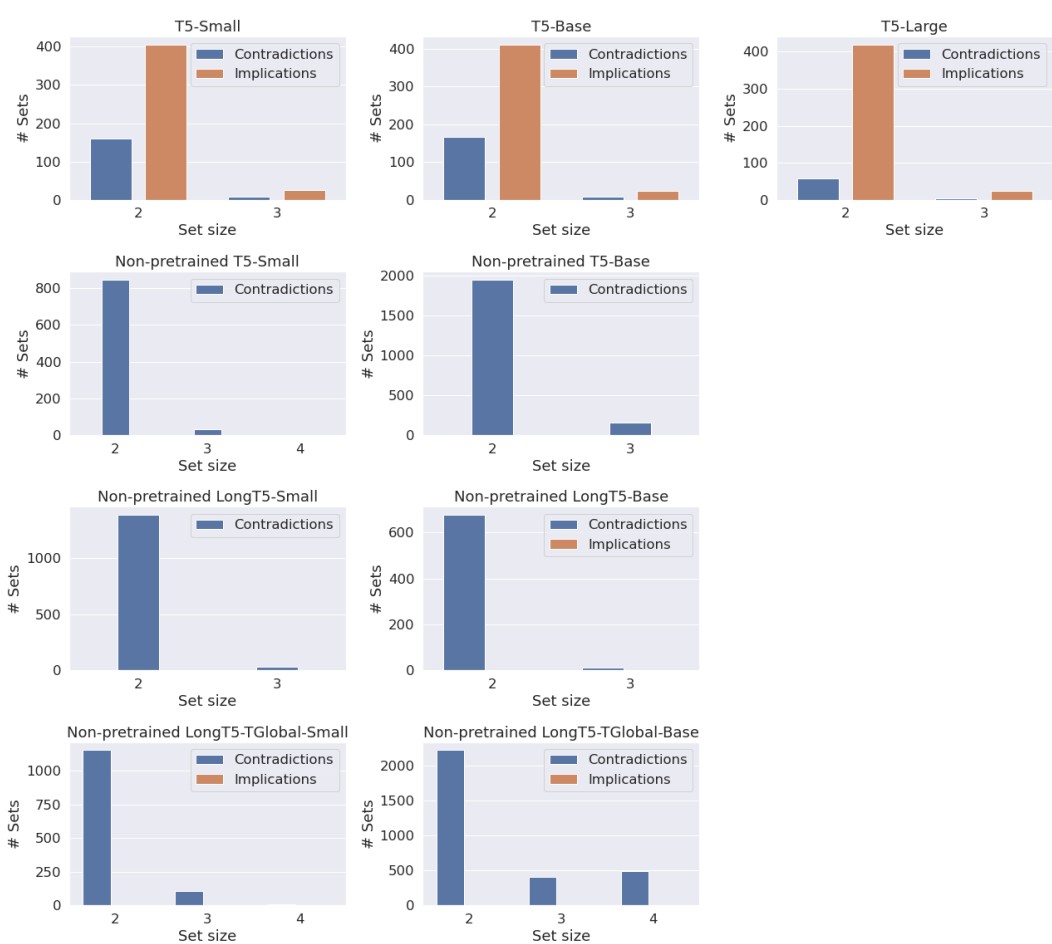

Figure 14: Consistency sets size distribution for cSCAN-B Random

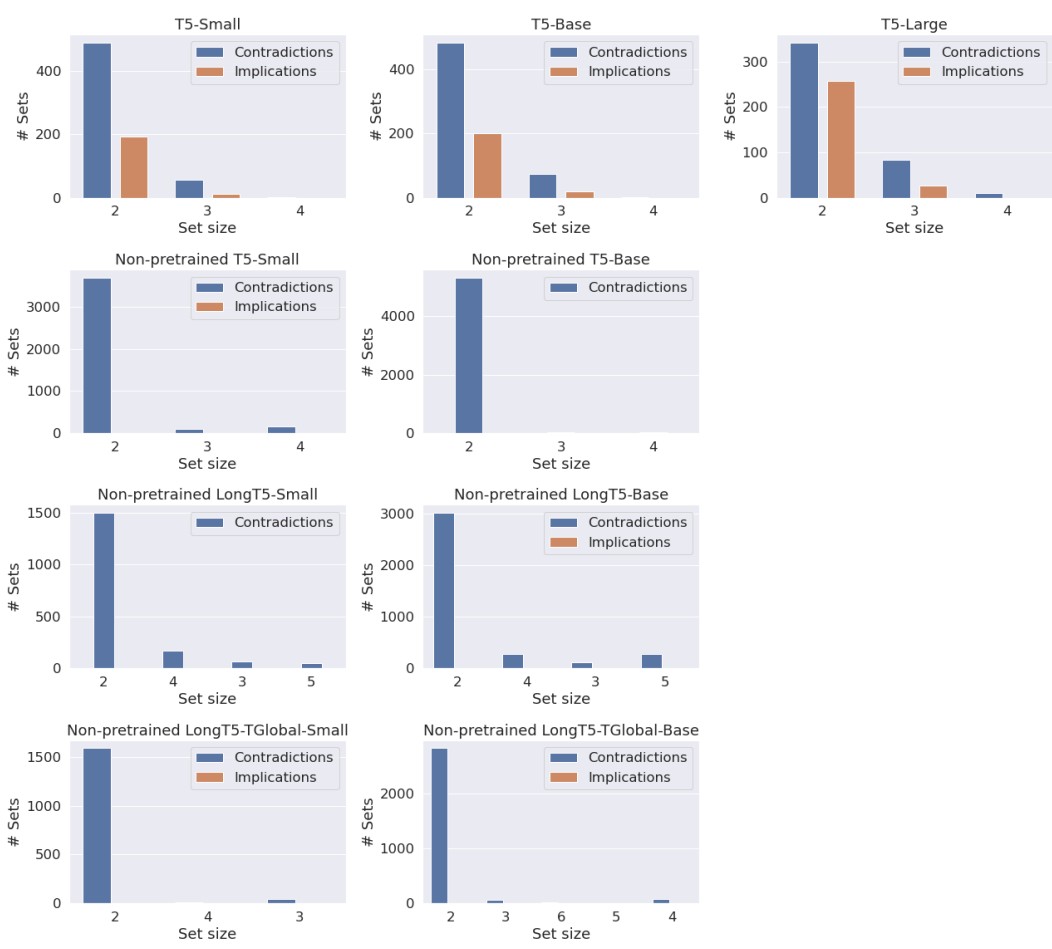

Figure 15: Consistency sets size distribution for cSCAN-X Random

Table 23: Edit distance measures for cSCAN-B Random

| Baseline | Pretrain | Size | Distance | Substitutions | Insertions | Deletions |
|---|---|---|---|---|---|---|
| T5 | True | S | 0.19 | 0.07 | 0.07 | 0.05 |
| T5 | True | B | 0.17 | 0.07 | 0.03 | 0.07 |
| T5 | True | L | **0.09** | **0.03** | **0.01** | **0.04** |
| T5 | False | S | 4.62 | 1.96 | 1.30 | 1.35 |
| T5 | False | B | 6.00 | 3.22 | 1.30 | 1.48 |
| LongT5 | False | S | 4.75 | 2.18 | 1.23 | 1.34 |
| LongT5 | False | B | 4.44 | 1.93 | 1.09 | 1.42 |
| LongT5-TGlobal | False | S | 4.30 | 1.86 | 1.18 | 1.26 |
| LongT5-TGlobal | False | B | 6.39 | 3.23 | 1.29 | 1.87 |
| T5 w/o Context | True | B | 6.19 | 2.89 | 2.22 | 1.08 |

Table 24: Edit distance measures for cSCAN-X Random

| Baseline | Pretrain | Size | Distance | Substitutions | Insertions | Deletions |
|---|---|---|---|---|---|---|
| T5 | True | S | 3.01 | 0.27 | 1.40 | 1.33 |
| T5 | True | B | 3.21 | 0.30 | 0.65 | 2.26 |
| T5 | True | L | **2.04** | **0.15** | **0.57** | **1.32** |
| T5 | False | S | 37.87 | 12.41 | 5.77 | 19.69 |
| T5 | False | B | 48.64 | 12.73 | 5.18 | 30.73 |
| LongT5 | False | S | 38.81 | 11.65 | 5.30 | 21.87 |
| LongT5 | False | B | 46.49 | 13.13 | 4.21 | 29.15 |
| LongT5-TGlobal | False | S | 36.74 | 11.59 | 6.11 | 19.04 |
| LongT5-TGlobal | False | B | 46.59 | 12.54 | 5.23 | 28.82 |
| T5 w/o Context | True | B | 44.81 | 14.17 | 8.37 | 22.27 |

Table 25: Edit distance measures for cSCAN-B MCD

| Baseline | Pretrain | Size | Distance | Substitutions | Insertions | Deletions |
|---|---|---|---|---|---|---|
| T5 | True | S | 1.96 | 0.69 | 0.82 | 0.44 |
| T5 | True | B | 1.72 | 0.60 | 0.90 | 0.22 |
| T5 | True | L | **1.23** | **0.42** | **0.63** | **0.18** |
| T5 | False | S | 3.41 | 1.26 | 1.28 | 0.86 |
| T5 | False | B | 4.29 | 2.12 | 1.07 | 1.10 |
| LongT5 | False | S | 3.39 | 1.32 | 1.47 | 0.60 |
| LongT5 | False | B | 3.64 | 1.54 | 1.23 | 0.87 |
| LongT5-TGlobal | False | S | 3.18 | 1.29 | 1.15 | 0.74 |
| LongT5-TGlobal | False | B | 3.50 | 1.28 | 1.47 | 0.75 |
| T5 w/o Context | True | B | 4.46 | 1.90 | 1.77 | 0.79 |

## N.3 EDIT DISTANCE

Edit distance is the number of edits that would need to be applied to the predicted sequence to transform it to the target sequence. There are three types of edits: Substitutions (S), Insertions (I) and Deletions (D).

Tables 23, 24, 26, and 26 show the edit distance and constituent metrics of each baseline on all of these metrics.

Table 26: Edit distance measures for cSCAN-X MCD

| Baseline | Pretrain | Size | Distance | Substitutions | Insertions | Deletions |
|----------|----------|------|----------|---------------|------------|-----------|
| T5 | True | *S* | 2.60 | 0.26 | 1.63 | **0.70** |
| T5 | True | *B* | 2.90 | 0.33 | 0.79 | 1.78 |
| T5 | True | *L* | **1.96** | **0.22** | **0.65** | 1.09 |
| T5 | False | *S* | 24.02 | 5.06 | 4.30 | 14.66 |
| T5 | False | *B* | 26.84 | 7.22 | 5.50 | 14.13 |
| LongT5 | False | *S* | 24.31 | 5.58 | 3.91 | 14.83 |
| LongT5 | False | *B* | 29.05 | 7.02 | 3.76 | 18.26 |
| LongT5-TGlobal | False | *S* | 24.21 | 5.37 | 4.37 | 14.47 |
| LongT5-TGlobal | False | *B* | 29.79 | 7.90 | 3.86 | 18.03 |
| T5 w/o Context | True | *B* | 30.25 | 9.72 | 6.49 | 14.04 |

## O    INPUT-OUTPUT FORMAT

We evaluate T5 variants, which are all encoder-decoder architectures. Here, we show how we prepare the input (that is fed to the encoder) and output (that the decoder generates) of our models, given an example from cSCAN. The example below is selected from the cSCAN-B dataset. Note that the request and the context examples are all concatenated into a single newline-separated string to form the input, and the bullet-points here (and in Appendix P.1) are just added to improve the readability of the examples.

```
Input: <Request, Context>
  turn opposite left and jump around left twice
•[left] = PURPLE
•[x1 around x2] = [x2] [x2] [x1] [x1]
•[x1 opposite x2] = [x1] [x2] [x2]
•[turn] = YELLOW
•[x1 after x2] = [x2] [x1]
•[walk] = LTURN
•[jump] = RUN
•run around left and turn opposite right twice PURPLE PURPLE JUMP JUMP YELLOW
RTURN RTURN YELLOW RTURN RTURN YELLOW RTURN RTURN
•walk opposite left twice after run thrice JUMP RUN LTURN PURPLE PURPLE LTURN PUR-
PLE PURPLE
•[run opposite left] = JUMP PURPLE PURPLE
•[run opposite x1 twice] = JUMP [x1] [x1]
•turn opposite right twice and turn around left thrice YELLOW RTURN RTURN YELLOW
RTURN RTURN YELLOW RTURN RTURN PURPLE PURPLE YELLOW YELLOW
•run right thrice JUMP RTURN RTURN
•[x1 around x2 thrice] = [x2] [x2] [x1] [x1]
•[look around x1 thrice] = [x1] [x1] PINK PINK
•look around right thrice and run opposite left RTURN RTURN PINK PINK JUMP PURPLE
PURPLE
•look around left thrice PURPLE PURPLE PINK PINK
•[look left thrice] = PINK PURPLE PURPLE
•[look opposite x1] = PINK [x1] [x1]
•run opposite right twice and turn around left twice JUMP RTURN RTURN PURPLE PUR-
PLE YELLOW YELLOW
•turn around left and walk twice PURPLE PURPLE YELLOW YELLOW LTURN LTURN JUMP
•[x1 and x2 thrice] = [x1] [x2]
•[x1 thrice and x2] = [x1] [x2]
•look thrice and run around right thrice PINK RTURN RTURN JUMP JUMP
•turn opposite right YELLOW RTURN RTURN
•[jump right] = RUN RTURN RTURN
•[x1 right] = [x1] RTURN RTURN
•turn left after look opposite right PINK RTURN RTURN YELLOW PURPLE PURPLE
•look right twice YELLOW PINK RTURN RTURN PINK RTURN RTURN
•[look x1] = PINK [x1] [x1]
•[x1 left] = [x1] PURPLE PURPLE
•walk around left twice afterz walk around left thrice PURPLE PURPLE LTURN LTURN
PURPLE PURPLE LTURN LTURN PURPLE PURPLE LTURN LTURN YELLOW
•jump right twice RUN RTURN RTURN RUN RTURN RTURN RTURN
•[look around x1 twice] = RUN [x1] [x1] PINK PINK [x1] [x1] PINK PINK
•[jump around left twice] = PINK PURPLE PURPLE RUN RUN PURPLE PURPLE RUN RUN
```

```
Output: <Reply, Qualifier>
  YELLOW PURPLE PURPLE PINK PURPLE PURPLE RUN RUN PURPLE PURPLE RUN RUN   (Reason-
ing:  Defeasible)
```

Note that we concatenate request + context, rather than context + request, so as to make the system more robust to truncation of the example string, if any example were to exceed the maximum length of T5's input buffer (although in our experiments we made sure that the example lengths did not exceed this buffer size).

In representing the context for T5, it can be noted that we omitted special syntactic tokens such as braces, angle brackets, and commas wherever possible, so as to reduce the token count and keep the format closer to natural language, to the extent that this could be done without introducing ambiguity. For context examples

that represent rule assertions, we also adopted a simplified syntax, similar to the shorthand described in Appendix C.2, consisting of a single line containing the rule request alone, while omitting the reply, since in cSCAN we only include positive rule assertions in the context (i.e., never rule examples with reply of **0** or **?**).

Note also that we do not perform any clustering of top-level examples by their context, but rather represent each top-level example in flattened form as shown here (with its context included). We then shuffle the full set of top-level examples before batching them for input into T5. This means that even when there may be 100 or more top-level examples with the same context, T5 will in general not see them all in sequence or in the same batch, but rather intermixed with top-level examples with different contexts.

# P   QUALITATIVE ERROR ANALYSIS

## P.1   ACCURACY ERROR ANALYSIS

Here, we showcase examples where our best model (T5-Large with pretraining) fails at producing the accurate results when evaluated on an example from the cSCAN-B. We show different cases where the target and the prediction from the model are different.

| Example of T5-Large failing at rule assertion (when the request is False). | |
| --- | --- |
| Input | `[run left after turn right] = LOOK WALK WHITE WHITE PINK LOOK WALK`
`•[left] = PINK`
`•[turn] = LOOK`
`•[x1 twice] = [x1] [x1]`
`•[x1 thrice] = [x1]`
`•[x1 and x2] = [x1] [x2] [x1]`
`•[look] = BLUE`
`•[x1 opposite x2] = [x2] [x1] [x1] [x2]`
`•[x1 x2] = [x1] [x2]`
`•[walk] = RTURN`
`•[x1 after x2] = [x2] [x1] [x1] [x2]`
`•[jump] = GREEN`
`•[right] = WALK`
`•[run] = WHITE`
`•walk around right and turn around left thrice RTURN WALK RTURN LOOK PINK LOOK RTURN WALK RTURN`
`•jump opposite right after look around left PINK BLUE PINK WALK GREEN GREEN WALK WALK GREEN GREEN WALK PINK BLUE PINK`
`•[jump around left] = GREEN GREEN`
`•[x1 around right twice] = WALK [x1] WALK WALK [x1] WALK` |
| target | 0 (Reasoning:  Monotonic) |
| prediction | 1 (Reasoning:  Monotonic) |
| Commentary | Since `[x1 x2] = [x1] [x2]`, `[run left]` and `[turn right]` would map to `WHITE PINK` and `LOOK WALK` respectively, and with `[x1 after x2] = [x2] [x1] [x1] [x2]`, the correct output should be `LOOK WALK WHITE PINK WHITE PINK LOOK WALK`. |

| Example of T5-Large failing at rule assertion (when the request is True). | |
|---|---|
| Input | [run and run around right thrice] = PURPLE
• [jump] = RED
• [x1 and x2] = [x2] [x1] [x2]
• [left] = LOOK
• [run] = PURPLE
• [x1 x2] = [x2] [x2] [x1]
• [x1 around x2] = [x1] [x2] [x1]
• [look] = RUN
• [x1 after x2] = [x2] [x2] [x1]
• [turn] = JUMP
• [walk] = LTURN
• [x1 twice] = [x1] [x1]
• walk opposite right GREEN LTURN
• look opposite right twice and jump twice RED RED GREEN RUN GREEN RUN RED RED
• [turn opposite right] = GREEN JUMP
• [look opposite x1] = [x1] RUN
• look right and jump around right thrice RED GREEN RED RED GREEN RED GREEN GREEN RUN RED GREEN RED RED GREEN RED
• turn opposite left twice and look thrice LOOK JUMP LOOK JUMP
• [x1 opposite right thrice] = GREEN [x1] GREEN [x1]
• [run around x1 thrice] = EMPTY_STRING
• run around right after run opposite right twice GREEN PURPLE GREEN PURPLE GREEN PURPLE GREEN PURPLE PURPLE GREEN PURPLE
• jump opposite left after turn opposite right twice GREEN JUMP GREEN JUMP GREEN JUMP GREEN JUMP LOOK RED
• [run around right] = PURPLE GREEN PURPLE
• [x1 opposite right] = GREEN [x1] |
| target | 1 (Reasoning: Monotonic) |
| prediction | 0 (Reasoning: Monotonic) |
| Commentary | Since run around x1 thrice maps to an empty string and [x1 and x2] = [x2] [x1] [x2] with [run] = PURPLE, the provided rule is True. |

| | Example of T5-Large failing at acknowledging lack of information to reply. |
|---|---|
| Input | turn around right thrice and jump opposite left thrice
•[x1 and x2] = [x1] [x2] [x1]
•[x1 around x2] = [x1] [x2] [x2]
•[x1 opposite x2] = [x1] [x2] [x2]
•[x1 twice] = [x1]
•[x1 after x2] = [x2] [x1] [x2]
•[left] = BLUE
•[x1 thrice] = [x1] [x1]
•[look] = PINK
•[run] = RED
•walk opposite right after run opposite left twice RED BLUE BLUE WHITE YEL-
LOW YELLOW RED BLUE BLUE
•walk opposite left twice and run WHITE BLUE BLUE RED WHITE BLUE BLUE
•[walk opposite x1] = WHITE [x1] [x1]
•[walk x1] = [x1] WHITE
•jump opposite left thrice and turn left thrice RTURN BLUE BLUE RTURN BLUE
BLUE BLUE RUN BLUE RUN RTURN BLUE BLUE RTURN BLUE BLUE
•turn opposite right twice after look opposite right PINK YELLOW YELLOW
RUN YELLOW YELLOW PINK YELLOW YELLOW
•[turn left] = BLUE RUN
•[turn around x1 twice] = RUN [x1] [x1]
•jump left twice after turn around left RUN BLUE BLUE BLUE LTURN RUN BLUE
BLUE
•jump around right thrice WALK WHITE YELLOW YELLOW WALK WHITE YELLOW YEL-
LOW
•[jump right] = YELLOW WALK WALK
•[jump opposite left] = YELLOW BLUE BLUE
•walk right twice and walk around left thrice YELLOW WHITE WHITE BLUE BLUE
WHITE BLUE BLUE YELLOW WHITE
•turn around right twice after jump around left WALK WALK BLUE BLUE RUN
YELLOW YELLOW WALK WALK BLUE BLUE
•[jump opposite right] = RED WALK YELLOW YELLOW
•[turn opposite right] = RUN YELLOW YELLOW
•jump right twice and jump thrice YELLOW BLACK BLACK BLACK YELLOW BLACK
•run right thrice after look around left twice PINK BLUE BLUE YELLOW RED
YELLOW RED PINK BLUE BLUE
•[x1 left] = BLUE [x1]
•[run x1] = [x1] RED |
| target | ?  (Reasoning:  Defeasible) |
| prediction | RUN YELLOW YELLOW RUN YELLOW YELLOW YELLOW BLUE BLUE YELLOW BLUE BLUE RUN
YELLOW YELLOW RUN YELLOW YELLOW (Reasoning:  Defeasible) |
| Commentary | [jump] = _ is not well illustrated by at least 2 unique substitutions,
therefore the mapping of [jump] is unknown. |

| | Example of T5-Large failing at drawing correct information to compose the reply. |
|---|---|
| Input | turn opposite left after turn around left thrice
•[x1 and x2] = [x1] [x2]
•[jump] = RTURN
•[x1 opposite x2] = [x1] [x2] [x2] [x1]
•[left] = YELLOW
•[x1 twice] = [x1]
•[turn] = RED
•[right] = WHITE
•[x1 thrice] = [x1] [x1]
•[walk] = GREEN
•[x1 after x2] = [x1] [x1] [x2] [x2]
•[run] = JUMP
•[x1 around x2] = [x2] [x1] [x2] [x1]
•look around left and walk twice YELLOW LOOK YELLOW LOOK GREEN
•look opposite left thrice WALK WALK YELLOW YELLOW WALK WALK WALK WALK YEL-LOW YELLOW WALK WALK
•[look opposite left thrice] = WALK WALK YELLOW YELLOW WALK WALK WALK WALK YELLOW YELLOW WALK WALK
•[look right] = WHITE WHITE RTURN RTURN
•walk right twice WHITE WHITE GREEN GREEN
•run left thrice and run YELLOW YELLOW JUMP JUMP YELLOW YELLOW JUMP JUMP JUMP
•[x1 left] = YELLOW YELLOW [x1] [x1]
•[turn x1] = [x1] [x1] RED RED |
| target | RED YELLOW YELLOW RED RED YELLOW YELLOW RED YELLOW RED YELLOW RED YELLOW RED YELLOW RED YELLOW RED YELLOW RED YELLOW RED YELLOW RED YELLOW RED (Reasoning: Monotonic) |
| prediction | RED YELLOW YELLOW RED RED YELLOW YELLOW RED YELLOW RED YELLOW RED YELLOW RED YELLOW RED YELLOW RED YELLOW RED YELLOW RED YELLOW RED (Reasoning:  Monotonic) |
| Commentary | •Composing [x1 around x2] = [x2] [x1] [x2] [x1] into [x1 thrice] = [x1] [x1] gives [x1 around x2 thrice] = [x2] [x1] [x2] [x1] [x2] [x1] [x2] [x1].
•Composing the above rule and [x1 opposite x2] = x1 x2 x2 x1 into x1 and x2 of [x1 after x2] = [x1] [x1] [x2] [x2] respectively results in the rule [x1 opposite x2 after x3 around x4 thrice] = [x1] [x2] [x2] [x1] [x1] [x2] [x2] [x1] [x4] [x3] [x4] [x3] [x4] [x3] [x4] [x3] [x4] [x3] [x4] [x3] [x4] [x3].
•Substituting x1 and x3 by [turn] = RED and x2 and x4 by [left] = YELLOW results in the rule [turn opposite left after turn around left thrice] = RED YELLOW YELLOW RED RED YELLOW YELLOW RED YELLOW RED YELLOW RED YELLOW RED YELLOW RED YELLOW RED YELLOW RED YELLOW RED YELLOW RED YELLOW RED.
Which varies from the predicted reply by the underlined tokens. |

| Example of T5-Large failing at inferring the type of reasoning, while the reply is correct. | |
|---|---|
| Input | `[x1 around left and x2 opposite right twice] = LOOK LOOK [x2] [x1] [x1] GREEN [x1] [x1] BLUE LOOK LOOK [x2]` 
 •`[x1 after x2] = [x1] [x2] [x1]` 
 •`[x1 thrice] = [x1] [x1]` 
 •`[x1 around x2] = [x1] [x1] [x2]` 
 •`[walk] = LTURN` 
 •`[right] = LOOK` 
 •`[left] = GREEN` 
 •`[x1 opposite x2] = [x2] [x2] [x1]` 
 •`[look] = WALK` 
 •`[x1 twice] = [x1]` 
 •`look twice and jump right WALK BLUE LOOK WALK WALK WALK BLUE LOOK` 
 •`turn twice and walk left GREEN LTURN RUN RUN GREEN LTURN` 
 •`[jump left] = BLUE BLUE GREEN` 
 •`[run left] = GREEN BLACK` 
 •`run around left twice BLACK BLACK GREEN` 
 •`turn around left and run opposite right LOOK LOOK BLACK RUN RUN GREEN RUN RUN GREEN LOOK LOOK BLACK` 
 •`[run around x1 thrice] = BLACK BLACK [x1] BLACK BLACK [x1]` 
 •`[run opposite x1] = [x1] [x1] BLACK` 
 •`walk right thrice after turn opposite left thrice LTURN LTURN LOOK LTURN LTURN LOOK GREEN GREEN RUN GREEN GREEN RUN LTURN LTURN LOOK LTURN LTURN LOOK` 
 •`turn around right twice RUN RUN LOOK` 
 •`[turn around x1] = RUN RUN [x1]` 
 •`[turn opposite left] = GREEN GREEN RUN` 
 •`run opposite left thrice and look WALK GREEN GREEN BLACK GREEN GREEN BLACK GREEN GREEN BLACK GREEN GREEN BLACK WALK` 
 •`run around right and look around right twice WALK WALK LOOK BLACK BLACK LOOK BLACK BLACK LOOK WALK WALK LOOK` 
 •`[x1 and x2 thrice] = [x2] [x2] [x1] [x1] [x2] [x2]` 
 •`[x1 and x2 opposite x3] = [x3] [x3] [x2] [x1] [x1] [x3] [x3] [x2]` 
 •`jump opposite right thrice after walk around left LOOK LOOK BLUE LOOK LOOK BLUE LTURN LTURN GREEN LOOK LOOK BLUE LOOK LOOK BLUE` 
 •`jump around right thrice and turn thrice RUN RUN BLUE BLUE LOOK BLUE BLUE LOOK BLUE BLUE LOOK BLUE BLUE LOOK RUN RUN` 
 •`[jump opposite right] = LOOK LOOK BLUE` 
 •`[jump opposite x1 twice] = [x1] [x1] BLUE` |
| target | `0 (Reasoning: Defeasible)` |
| prediction | `0 (Reasoning: Monotonic)` |
| Commentary | •The rule `[x1 and x2] = [x2] [x1] [x1] [x2]` is hidden and can be induced from the context examples highlighted by an underline. 
 •Composing `[x1 opposite x2] = [x2] [x2] [x1]` into `[x1 twice] = [x1]` results into the rule `[x1 opposite x2 twice] = [x2] [x2] [x1]` 
 •Composing `[x1 around x2] = [x1] [x1] [x2]` and `[x1 opposite x2 twice] = [x2] [x2] [x1]` into *x1* and *x2* of `[x1 and x2] = [x2] [x1] [x1] [x2]` respectively results in the rule `[x1 around x2 and x3 opposite x4 twice] = [x4] [x4] [x3] [x1] [x1] [x2] [x1] [x1] [x2] [x4] [x4] [x3]` 
 •Substituting *x2* and *x4* by `[left] = GREEN` and `[right] = LOOK` respectively results in the rule `[x1 around left and x2 opposite right twice] = LOOK LOOK [x3] [x1] [x1] GREEN [x1] [x1] GREEN LOOK LOOK [x3]`. |

## P.2    CONSISTENCY ERROR ANALYSIS

All the above examples show inaccurate output, where there is a mismatch between the expected target and model prediction. However, as discussed in Section 4, regardless of being accurate, a model may fail to stay consistent in replying to different requests. In this section we sample contradictions made by T5-Large on cSCAN-X Random and categorize the type of contradictions the model makes. Note that consistency is independent of the context and therefore it's omitted when presenting the examples.

As seen in Appendix N.2, the models produce contradictory sets at different sizes. We find it easy to analyse each size independently as the type of mistakes vary between them.

**Contradictions of size** 2     At this level, the inconsistencies are 1:1 relationships between two contradictory predictions $P1$ and $P2$. There are two types at this level

- **Type 1:** Both $P1$ and $P2$ are asserted rules with matching right-hand sides and and different right-hand sides.
- **Type 2:** One prediction is an asserted rule with no variables and the other is a rule application prediction with a request matching the right-hand side of the rule request while the reply does not match the rule request's right-hand side.

From a sample of 20 contradictory sets of size 2, 11 sets were of type 1 while 9 were found to be of type 2. Here are examples of both types

| Contradictory set of size 2: Type 1 | |
|---|---|
| Request 1 | `[drive left 5x framing x1 x2] = [x1] [x2] [x1] [x1] [x2] [x2] [x1] [x2]` `[x1] [x1] [x2] [x2] DTURN LOOK DTURN DTURN LOOK LOOK DTURN LOOK DTURN` `DTURN LOOK LOOK [x1] [x2] [x1] [x1] [x2] [x2] [x1] [x2] [x1] [x1] [x2]` `[x2] [x1] [x2] [x1] [x1] [x2] [x2] [x1] [x2] [x1] [x1] [x2] [x2]` |
| Reply 1 | `1 (Reasoning:  Monotonic)` |
| Request 2 | `[drive left 5x framing x1 x2] = [x1] [x2] [x1] [x1] [x2] [x2] [x1] [x1]` `[x1] [x1] [x2] [x2] DTURN LOOK DTURN DTURN LOOK LOOK DTURN LOOK DTURN` `DTURN LOOK LOOK [x1] [x2] [x2] [x1] [x2] [x2] [x1] [x2] [x1] [x1] [x2]` `[x2] [x1] [x2] [x1] [x1] [x2] [x2] [x1] [x2] [x1] [x1] [x2] [x2]` |
| Reply 2 | `1 (Reasoning:  Monotonic)` |
| Commentary | `The two right-hand sides are different in the underlined tokens.` |

| Contradictory set of size 2: Type 2 | |
|---|---|
| Request 1 | `[run zigzag left after run fast around left] = PEEK PEEK PEEK PEEK PEEK` `PEEK PEEK PEEK PEEK PEEK PEEK PEEK PEEK PEEK PEEK PEEK PEEK RUN PEEK` `PEEK PEEK PEEK PEEK PEEK PEEK RUN RUN RUN PEEK RUN RUN RUN PEEK RUN RUN` `RUN PEEK PEEK PEEK PEEK PEEK PEEK PEEK PEEK PEEK PEEK PEEK PEEK PEEK PEEK` `PEEK PEEK PEEK RUN PEEK PEEK PEEK PEEK PEEK PEEK` |
| Reply 1 | `1 (Reasoning:  Defeasible)` |
| Request 2 | `run zigzag left after run fast around left` |
| Reply 2 | `PEEK PEEK PEEK PEEK PEEK PEEK PEEK PEEK PEEK PEEK PEEK PEEK PEEK PEEK PEEK` `PEEK PEEK PEEK RUN PEEK PEEK PEEK PEEK PEEK PEEK PEEK RUN RUN RUN PEEK RUN` `RUN RUN PEEK RUN RUN RUN PEEK PEEK PEEK PEEK PEEK PEEK PEEK PEEK PEEK PEEK` `PEEK PEEK PEEK PEEK PEEK PEEK PEEK PEEK RUN PEEK PEEK PEEK PEEK PEEK PEEK` `(Reasoning:  Defeasible)` |
| Commentary | `Reply 2 is different from the right-hand side of Request 1 by one addi-` `tional token (highlighted with an underline).` |

**Contradictions of size** 3     At this level, the inconsistencies are 2:1 relationships with at least two rule assertions.

- **Type 1:** The composition of two rules directly contradicts a third rule.

- **Type 2:** The composition of two rules forms a rule with no variables where it's right-hand side matches the request of a rule application prediction and it's right-hand side contradicts the reply of that prediction.

From a sample of 20 contradictory sets of size 3, both Type 1 and Type 2 had 10 sets each. Here are examples of both types

| Contradictory set of size 3: Type 1 | |
|---|---|
| Request 1 | `[x1 cautiously x2] = [x1] [x1] [x1] [x2] [x2] [x2] [x1] [x1] [x1] [x2] [x2]` |
| Reply 1 | `1 (Reasoning: Monotonic)` |
| Request 2 | `[walk] = RIDE RIDE` |
| Reply 2 | `1 (Reasoning: Monotonic)` |
| Request 3 | `[walk cautiously x1] = RIDE RIDE RIDE [x1] [x1] [x1] RIDE RIDE RIDE [x1] [x1]` |
| Reply 3 | `1 (Reasoning: Monotonic)` |
| Commentary | `Composing Rule 2 into Rule 1 would result in the rule`
`•[walk cautiously x1] = RIDE RIDE RIDE RIDE RIDE RIDE [x1] [x1] [x1] RIDE RIDE RIDE RIDE RIDE RIDE [x1] [x1]`
`Which is different from Rule 3 in places highlighted by an underline.` |

| Contradictory set of size 3: Type 2 | |
|---|---|
| Request 1 | `[run opposite up thrice following x1] = LOOK LOOK RUN LOOK LOOK LOOK RUN LOOK LOOK LOOK RUN LOOK [x1] LOOK LOOK RUN LOOK [x1] LOOK LOOK RUN LOOK [x1]` |
| Reply 1 | `1 (Reasoning: Monotonic)` |
| Request 2 | `run opposite up thrice following run zigzag` |
| Reply 2 | `LOOK LOOK RUN LOOK LOOK LOOK RUN LOOK LOOK LOOK RUN LOOK LOOK LOOK RUN LOOK LOOK LOOK LOOK LOOK RUN LOOK LOOK LOOK RUN LOOK LOOK LOOK LOOK RUN LOOK LOOK LOOK LOOK LOOK RUN LOOK LOOK LOOK RUN LOOK LOOK LOOK RUN LOOK LOOK (Reasoning: Monotonic)` |
| Request 3 | `[run zigzag] = LOOK` |
| Reply 3 | `1 (Reasoning: Monotonic)` |
| Commentary | `Composing Rule 2 into Rule 3 would result in the rule`
`•[run opposite up thrice following run zigzag] = LOOK LOOK RUN LOOK LOOK LOOK RUN LOOK LOOK LOOK RUN LOOK LOOK LOOK LOOK RUN LOOK LOOK LOOK LOOK RUN LOOK LOOK`
`Which matches Request 2 in its right-hand side but is different from Reply 2 by its right-hand side in places highlighted by an underline.` |

**Contradictions of size** 4    At this level there are more complex contradictory relations as the space of possible compositions grows. we categorise 3 types of contradictions

- **Type 1:** Composing 3 rules to form a rule with no variables that contradicts a rule application prediction.
- **Type 2:** The composition of a subset set of rules contradicts the composition of another set of rules.
- **Type 3:** The composition of 3 rules directly contradicts another rule.

For this size, the model only made 10 contradictions: 5 were of type 2, 3 were of type 3 and 2 were of type 1. Here are examples for each type

| Contradictory set of size 4: Type 1 | |
|---|---|
| Request 1 | [run drunkenly down] = RTURN PEEK PEEK PEEK PEEK PEEK |
| Reply 1 | 1 (Reasoning:  Monotonic) |
| Request 2 | jump drunkenly twice after run drunkenly down |
| Reply 2 | RIDE RIDE RIDE RIDE RIDE RIDE RIDE RIDE RIDE RIDE RIDE RIDE RIDE RIDE RIDE RIDE RIDE RIDE RIDE RIDE RIDE RIDE RIDE RIDE RIDE RIDE RIDE RIDE RIDE RIDE RIDE RIDE RIDE RIDE RIDE RIDE RIDE RIDE RIDE RIDE RIDE RIDE RIDE RIDE RIDE RIDE RIDE RIDE RIDE RIDE RIDE RIDE RIDE RIDE RIDE RIDE RIDE RIDE RIDE RIDE RIDE RIDE RIDE RIDE RIDE RIDE RIDE RIDE RIDE RIDE RIDE RIDE RIDE RIDE RIDE RIDE RIDE RIDE RIDE RIDE RIDE RIDE RIDE RTURN PEEK PEEK PEEK PEEK PEEK (Reasoning:  Monotonic) |
| Request 3 | [x1 twice after x2] = [x1] [x1] [x1] [x1] [x1] [x1] [x1] [x1] [x1] [x1] [x1] [x1] [x2] |
| Reply 3 | 1 (Reasoning:  Monotonic) |
| Request 4 | [jump drunkenly] = RIDE RIDE RIDE RIDE RIDE RIDE |
| Reply 4 | 1 (Reasoning:  Monotonic) |
| Commentary | Composing rule 1 into *x2* of rule 3 and rule 4 into *x1* of rule 3 would result in the rule
•*[jump drunkenly twice after run drunkenly down] = RIDE RIDE RIDE RIDE RIDE RIDE RIDE RIDE RIDE RIDE RIDE RIDE RIDE RIDE RIDE RIDE RIDE RIDE RIDE RIDE RIDE RIDE RIDE RIDE RIDE RIDE RIDE RIDE RIDE RIDE RIDE RIDE RIDE RIDE RIDE RIDE RIDE RIDE RIDE RIDE RIDE RIDE RIDE RIDE RIDE RIDE RIDE RIDE RIDE RIDE RIDE RIDE RIDE RIDE RIDE RIDE RIDE RIDE RIDE RIDE RIDE RIDE RIDE RIDE RIDE RIDE RIDE RIDE RIDE RIDE RIDE RIDE RIDE RIDE RIDE RTURN PEEK PEEK PEEK PEEK PEEK*
Which matches Request 2 in its right-hand side but is different from Reply 2 by its right-hand side in places highlighted by an underline. |

| Contradictory set of size 4: Type 2 | |
|---|---|
| Request 1 | `[x1 cautiously x2 5x] = [x1] [x1] [x1] [x2] [x2] [x2] [x1] [x1] [x1] [x2]`
`[x2] [x1] [x1] [x1] [x2] [x2] [x2] [x1] [x1] [x1] [x2] [x2] [x1] [x1] [x1]`
`[x2] [x2] [x2] [x1] [x1] [x1] [x2] [x2] [x1] [x1] [x1] [x2] [x2] [x2] [x1]`
`[x1] [x1] [x2] [x2] [x1] [x1] [x1] [x2] [x2] [x2] [x1] [x1] [x1] [x2] [x2]`
`[x1] [x1] [x1] [x2] [x2] [x2] [x1] [x1] [x1] [x2] [x2]` |
| Reply 1 | `1 (Reasoning: Monotonic)` |
| Request 2 | `[walk cautiously x1] = RIDE RIDE RIDE [x1] [x1] [x1] RIDE RIDE RIDE [x1]`
`[x1]` |
| Reply 2 | `1 (Reasoning: Monotonic)` |
| Request 3 | `[x1 5x] = [x1] [x1] [x1] [x1] [x1] [x1]` |
| Reply 3 | `1 (Reasoning: Defeasible)` |
| Request 4 | `[walk] = RIDE RIDE` |
| Reply 4 | `1 (Reasoning: Monotonic)` |
| Commentary | Composing rule 4 into *x1* of rule 1 results in the rule
•Rule 5 = *[walk cautiously x1 5x] = RIDE RIDE RIDE RIDE RIDE RIDE*
*[x1] [x1] [x1] RIDE RIDE RIDE RIDE RIDE RIDE [x1] [x1] RIDE RIDE RIDE*
*RIDE RIDE RIDE [x1] [x1] [x1] RIDE RIDE RIDE RIDE RIDE RIDE [x1] [x1]*
*RIDE RIDE RIDE RIDE RIDE RIDE [x1] [x1] [x1] RIDE RIDE RIDE RIDE RIDE RIDE*
*[x1] [x1] RIDE RIDE RIDE RIDE RIDE RIDE [x1] [x1] [x1] RIDE RIDE RIDE*
*RIDE RIDE RIDE [x1] [x1] RIDE RIDE RIDE RIDE RIDE RIDE [x1] [x1] [x1] RIDE*
*RIDE RIDE RIDE RIDE RIDE [x1] [x1] RIDE RIDE RIDE RIDE RIDE RIDE [x1] [x1]*
*[x1] RIDE RIDE RIDE RIDE RIDE RIDE [x1] [x1]*
And composing rule 2 into rule 3 results in the rule
•Rule 6 = *[walk cautiously x1 5x] = RIDE RIDE RIDE [x1] [x1] [x1] RIDE*
*RIDE RIDE [x1] [x1] RIDE RIDE RIDE [x1] [x1] [x1] RIDE RIDE RIDE [x1] [x1]*
*RIDE RIDE RIDE [x1] [x1] [x1] RIDE RIDE RIDE [x1] [x1] RIDE RIDE RIDE [x1]*
*[x1] [x1] RIDE RIDE RIDE [x1] [x1] RIDE RIDE RIDE [x1] [x1] [x1] RIDE RIDE*
*RIDE [x1] [x1] RIDE RIDE RIDE [x1] [x1] [x1] RIDE RIDE RIDE [x1] [x1]*
Which directly contradicts rule 5 in the places highlighted by an under-
line. |

| Contradictory set of size 4: Type 3 | |
|---|---|
| Request 1 | [x1 between fly cautiously 90 x2] = [x1] [x1] WALK WALK WALK WALK WALK WALK [x2] WALK WALK WALK WALK WALK WALK WALK WALK [x2] [x1] |
| Reply 1 | 1 (Reasoning:  Monotonic) |
| Request 2 | [walk] = RIDE RIDE |
| Reply 2 | 1 (Reasoning:  Monotonic) |
| Request 3 | [x1 cautiously 90 x2 3x] = [x1] [x1] [x1] [x1] [x1] [x1] [x2] [x1] [x1] [x1] [x1] [x1] [x1] [x1] [x1] [x2] |
| Reply 3 | 1 (Reasoning:  Defeasible) |
| Request 4 | [walk cautiously 90 x1 3x between fly cautiously 90 x2] = RIDE RIDE RIDE RIDE RIDE RIDE [x1] RIDE RIDE RIDE RIDE RIDE RIDE RIDE RIDE RIDE [x1] RIDE RIDE RIDE RIDE RIDE RIDE [x1] RIDE RIDE RIDE RIDE RIDE RIDE RIDE RIDE [x1] WALK WALK WALK WALK WALK WALK [x2] WALK WALK WALK WALK WALK WALK WALK WALK WALK WALK [x2] RIDE RIDE RIDE RIDE RIDE RIDE [x1] RIDE RIDE RIDE RIDE RIDE RIDE RIDE RIDE [x1] |
| Reply 4 | 1 (Reasoning:  Monotonic) |
| Commentary | Composing rule 2 into *x1* of rule 3 results in the rule
•Rule 5 = *[walk cautiously 90 x1 3x] = RIDE RIDE RIDE RIDE RIDE RIDE RIDE RIDE RIDE RIDE RIDE RIDE [x1] RIDE RIDE RIDE RIDE RIDE RIDE RIDE RIDE RIDE RIDE RIDE RIDE RIDE RIDE RIDE RIDE [x1]*
And composing rule 5 into *x1* of rule 1 results in the rule
•Rule 6 = *[walk cautiously 90 x1 3x between fly cautiously 90 x2] = RIDE RIDE RIDE RIDE RIDE RIDE RIDE RIDE RIDE RIDE RIDE RIDE [x1] RIDE RIDE RIDE RIDE RIDE RIDE RIDE RIDE RIDE RIDE RIDE RIDE RIDE RIDE RIDE RIDE RIDE RIDE [x1] RIDE RIDE RIDE RIDE RIDE RIDE RIDE RIDE RIDE RIDE RIDE RIDE [x1] RIDE RIDE RIDE RIDE RIDE RIDE RIDE RIDE RIDE RIDE RIDE RIDE RIDE RIDE RIDE RIDE RIDE RIDE [x1] WALK WALK WALK WALK WALK WALK [x2] WALK WALK WALK WALK WALK WALK WALK WALK [x2] RIDE RIDE RIDE RIDE RIDE RIDE RIDE RIDE RIDE RIDE RIDE RIDE [x1] RIDE RIDE RIDE RIDE RIDE RIDE RIDE RIDE RIDE RIDE RIDE RIDE RIDE RIDE RIDE RIDE RIDE RIDE RIDE RIDE RIDE [x1]*
Which directly contradicts rule 4 in the places highlighted by an underline. |