# OpenReview forum: "Conceptual SCAN: Learning With and About Rules"
_ICLR.cc/2023/Conference — Submitted to ICLR 2023_

### Official Review · Reviewer_N6X6 · 2022-10-22

**Confidence:** 4
**Correctness:** 3
**Technical Novelty And Significance:** 3
**Empirical Novelty And Significance:** 2
**Recommendation:** 5

**Clarity, Quality, Novelty And Reproducibility:**

The clarity of this paper should be improved. The quality of this paper is below average. The paper proposes a novel benchmark dataset that has good reproducibility.

**Strength And Weaknesses:**

### Strengths:
- The authors provided extensive details about how this dataset is generated, hence the reproducibility of this work is solid.
- This work is motivated by an important problem. Semantic parsing by deep learning can hardly make use of domain knowledge or context such as common sense knowledge and rules. The proposed cSCAN dataset could be a candidate for evaluating it.

### Weaknesses:
- The presentation could be improved. The motivating example in natural language QA is intriguing and convincing. However, the proposed dataset is an extended version of SCAN, i.e., a set of robot navigation instructions. While reading this paper, I found myself feeling disappointed and confused when I see the gap between them. And the natural language QA example doesn't help me understand how the "rules" work in the robot navigation problem. Maybe changing the motivating example to a problem that is more related to the cSCAN dataset could improve the clarity of this paper.
  - The notations in this paper are confusing. For example, $e_k= \langle i_k, o_k \rangle \in\mathcal{E}$ are examples, which consists of both inputs $i_k$ and $o_k$. However, the $i_k$ and $o_k$ are bost consist of the context $C$, and $C\in 2^{\mathcal{E}}$ is constructed by $e_k$. This is a recursive definition and makes me confused.

**Summary Of The Paper:**

This paper extends the well-known compositional generalization benchmark dataset SCAN to cSCAN, driven by the idea of evaluating machine learning models' ability in semantic parsing with context and rule-like constraints.

**Summary Of The Review:**

This paper proposes a novel dataset to evaluate neural networks' semantic parsing ability while considering knowledge-based constraints. The problem is important to the AI community, but the presentation of this paper should be improved.

---

> ### Author Response · Authors · 2022-11-19
> **Author Response**
>
> Thank you for your thoughtful feedback.
>
> **Presentation: Connection between motivating example and cSCAN**
>
> > The motivating example in natural language QA is intriguing and convincing. However, the proposed dataset is an extended version of SCAN, i.e., a set of robot navigation instructions. While reading this paper, I found myself feeling disappointed and confused when I see the gap between them. And the natural language QA example doesn't help me understand how the "rules" work in the robot navigation problem. Maybe changing the motivating example to a problem that is more related to the cSCAN dataset could improve the clarity of this paper.
>
> Thank you for this feedback. We agree that we could have been clearer about the connection between the real-world motivating assistant use case and the cSCAN task, and we have updated the text of the introduction and the beginning of Section 3 accordingly. The motivating assistant example is one we would want to keep, as the challenges of achieving a robust personal assistant (including, but not limited to, its usage for providing movie recommendations) was indeed one of the motivations for our current line of research into conceptual learning. One of the main goals of this paper, however, is to abstract out the key properties of interest from our motivating use case into the notion of a "conceptual learning task", such that we can make progress in understanding and improving the capabilities of ML systems on the family of tasks that share these key properties, even while starting with an initial "simplest possible" CLT instance, which we generate synthetically so as to be able to carefully control the task complexity. We would argue that if we have done a good job at selecting an appropriate set of properties of the motivating use case, and of encapsulating the important dynamics of this type of task in the CLT definition, then this first ``simplest possible'' CLT could potentially be based on a very different underlying task than the motivating use case, as long as these two tasks share the properties of a CLT. We hope that the improved presentation of this motivation in the introduction and early sections of the paper can help make clearer to the reader why we find this approach compelling.
>
> **Presentation: Recursive definition of contexts**
>
> > The notations in this paper are confusing. For example, ek=⟨ik,ok⟩∈E are examples, which consists of both inputs ik and ok. However, the ik and ok are bost consist of the context C, and C∈2E is constructed by ek. This is a recursive definition and makes me confused.
>
> Based on your feedback and that of Reviewers 44yb and qoQG, we understand that the notation for CLT examples introduced in Section 2.2 was indeed more complicated than necessary for the simple cSCAN task, particularly with regard to the recursive definition of the contexts (which we do not actually make use of for cSCAN). We have accordingly updated the structural definition of CLTs presented in Section 2.2 to adopt a simpler and more concise notation, without recursion, and with just a single standard form of this notation that is used throughout the main body of the paper. The discussion of the more general form of the CLT definition (which allows for nested contexts), along with some discussion of the motivation for when this could be useful, has been now moved into Appendix C: Generalization of CLTs to Support Nested Contexts.

---

### Official Review · Reviewer_qoQG · 2022-10-24

**Confidence:** 4
**Correctness:** 3
**Technical Novelty And Significance:** 2
**Empirical Novelty And Significance:** 3
**Recommendation:** 6

**Clarity, Quality, Novelty And Reproducibility:**

The paper is fairly clear and easy to follow, and there are abundant appendices containing details for dissecting the results and reproducing the evaluations. The motivation for training ML models in such a way that they can consume both examples and rules is interesting, and while somewhat related to traditional knowledge-base approaches, novel in that it contains _new_ rules/contexts accompanying each example. This means that it is not sufficient for the agent to learn a fixed set of rules from a fixed data set, but it actually has to be able to interpret new rules at evaluation time and solve the task in the context corresponding to those new rules. Moreover, the notion of providing examples alongside the rules as additional context is also interesting and, to my knowledge, novel.

That being said, I believe that there is some room for improvement in a couple of places, in particular regarding the scope of the contribution and the experimental evaluation.

**Scope of the contribution**: in the early sections of the paper, the presentation seems to suggest that a contribution of the submission was a method for designing new CLTs and data sets for them given an existing data set. While this would have been fantastic, I don't believe that there's evidence later in the manuscript to substantiate this claim. Sure, it _may_ be possible to follow the process used for cSCAN to create other CLT data sets, but it is unclear exactly what that would entail. Given that only SCAN is explicitly demonstrated to be extensible into a CLT, I would encourage the authors to scope down their claimed contributions to more clearly state this. This way, the contribution wouldn't be a "recipe to derive a CLT from an underlying base task", but rather "the recipe we follow to derive a CLT from the underlying SCAN task". Perhaps, you could include a sentence indicating that it might be possible to use this recipe for _other_ tasks, but that this hasn't been pursued in this work.

**Experimental section**: The first (and simplest to address) comment I have is that it is not stated anywhere in the paper how the transformers consume the context rules/examples. I assume these are tokenized and passed as additional inputs alongside the top-level example? Please clarify in your response and in the updated manuscript. The second is a broader comment regarding the amount of insight from the results. At this point, there's essentially a single architecture evaluated, with some variants in the details of how attention is implemented and the number of parameters of the architecture. This provides _some_ insight, especially regarding the the impact of scale on the results, but it fails to address multiple other axes. In particular, one thing that might be interesting to consider is how neuro-symbolic / program synthesis approaches would fare in cSCAN. Since the rules are explicitly provided, I would think these methods wouldn't have an "unfair" advantage, as they typically have in other tasks where such approaches still require access to some rules. The authors could also consider including meta-learning approaches in their set of baselines.


############## Additional feedback ##############

The following points are provided as feedback to hopefully help better shape the submitted manuscript, but did not impact my recommendation in a major way.

Intro
- The idea and motivation seem interesting
- Even from this point, I was very much wondering what the systems are supposed to do with the rules. Are these going to be introduced as language input somehow? I hoped not (or at least, not just). Given their symbolic nature, some form of neuro-symbolic approach would be most suitable. Something like neural module nets or a program synthesis approach.

Sec 2.2
- I find the recursive definition of contexts odd, especially consdiering that the examples and even the introduced data set do not contain any context that is itself an example. Couldn't it then be defined as C \in 2^(Q \times R \times U), simplifying the definition? What is the advantage of including C recursively in the definition?
- I suggest that the authors include the word "defeasible" earlier, much like they did with "monotonically" in the examples from Sec. 2.1

Sec 2.3
- I think it would benefit the readability of this section to continue with the running example of the movie recommendation task.

Sec 3
- The first time the manuscript clarifies that the context varies generating multiple "tasks" seems to be when introducing cSCAN. Parhaps the authors could state something about this more explicitly earlier on for additional clarity? Particularly in the general CLT descriptions.
- Could the authors provide some insight into why two different versions of cSCAN are desireable/useful/interesting? I later came to see that this was due to the "scalability to larger rule sets". It might be worth including that here
- Interestingly, the MCD makes the problem harder in terms of generalizing to unseen top-level requests (I believe this is the same type of generalization as studied in SCAN), but it makes the problem of generalizing to new contexts easier because the contexts might be repeated between train/test. I wonder if the design makes it impossible to have these two simultaneously

Supplement
- Source code is promised but not provided at submission time

Typos/style/grammar/layout
- It's best to use "CLT" only after introducing the acronym (instead of jumping back and forth between "CLT" and "conceptual learning task")



**Strength And Weaknesses:**

########### Strengths ###########
- The idea of constructing data sets where examples come annotated with a context, which itself contains both examples and rules, is interesting
- The running example of the movie recommendation system helps motivate and provide context to the reader for some of the design choices behind CLTs
- The connections to related literature, especially in terms of contextual data sets and meta-learning, are well established and help provide additional context for the usefulness of the data set


########### Weaknesses ###########
- The experimental evaluation is not as thorough as I would have liked to see, and it seems to not provide as much insight as would be possible
- The detail of how the transformers consume context are not explicitly provided
- While the approach to creating CLTs seems to be presented as a general-purpose data-set-generating method, it is largely unclear whether it would be easily applicable beyond SCAN


**Summary Of The Paper:**

The submission presents conceptual SCAN (cSCAN) as an instance of a conceptual learning task (CLT). CLTs, as defined in the manuscript, are tasks that contain a combination of examples and rules. Concretely, each example is defined as a tuple including a context, where the context is a set of rules and examples (potentially recursively, but in practice the inner examples have an "empty" context). cSCAN follows this formulation to introduce 4 different data sets, with smaller/larger set of rules and with/without explicit split for testing compositional generalization. Variants of the T5 transformer architecture are evaluated on the data set and demonstrated to fail in all but the simplest instances of the data set.


**Summary Of The Review:**

With all the above in mind, I am leaning towards recommending the acceptance of the paper, since I do think that the data set on its own presents a valuable contribution, and there are some insights derived from the empirical evaluation. I do encourage the authors to more accurately scope their contributions in the introductory sections and to provide additional details of the evaluation. If additional experiments are feasible during the review process, those would likely be most impactful towards turning this into a higher quality manuscript.

---

> ### Author Response · Authors · 2022-11-19
> **Author Response 1: Scope of the contribution; Experimental section**
>
> Thank you for your thoughtful feedback.
>
> **Scope of the contribution:**
>
> >  In the early sections of the paper, the presentation seems to suggest that a contribution of the submission was a method for designing new CLTs and data sets for them given an existing data set. While this would have been fantastic, I don't believe that there's evidence later in the manuscript to substantiate this claim. Sure, it may be possible to follow the process used for cSCAN to create other CLT data sets, but it is unclear exactly what that would entail. Given that only SCAN is explicitly demonstrated to be extensible into a CLT, I would encourage the authors to scope down their claimed contributions to more clearly state this. This way, the contribution wouldn't be a "recipe to derive a CLT from an underlying base task", but rather "the recipe we follow to derive a CLT from the underlying SCAN task". Perhaps, you could include a sentence indicating that it might be possible to use this recipe for other tasks, but that this hasn't been pursued in this work.
>
> Thank you for pointing this out. We agree that since we have not directly demonstrated applicability of this recipe to constructing other CLTs outside of the family of cSCAN, it would be better in this paper not to claim generality of the recipe. We have updated the discussion of our paper's contributions in the abstract, introduction and elsewhere accordingly.
>
> **Experimental section: How transformers consume the context**
>
> > The first (and simplest to address) comment I have is that it is not stated anywhere in the paper how the transformers consume the context rules/examples. I assume these are tokenized and passed as additional inputs alongside the top-level example? Please clarify in your response and in the updated manuscript.
>
> We describe the way in which the context rules/examples are consumed by the transformers in Appendix M: Input-Output Format (now Appendix O in the latest draft). We appreciate, however, that these appendices could have been made more discoverable, and we have updated the paper draft to now reference them more explicitly from Section 5 and other relevant places in the paper (in addition to from the Reproducibility Statement). We have also updated Appendix O to include more detailed commentary on the input-output format.
>
> **Experimental section: Additional baselines or other experiment axes**
>
> > The second is a broader comment regarding the amount of insight from the results. At this point, there's essentially a single architecture evaluated, with some variants in the details of how attention is implemented and the number of parameters of the architecture. This provides some insight, especially regarding the the impact of scale on the results, but it fails to address multiple other axes. In particular, one thing that might be interesting to consider is how neuro-symbolic / program synthesis approaches would fare in cSCAN. Since the rules are explicitly provided, I would think these methods wouldn't have an "unfair" advantage, as they typically have in other tasks where such approaches still require access to some rules. The authors could also consider including meta-learning approaches in their set of baselines.
>
> As discussed also in our response to Reviewer TbTf, we are indeed also interested in exploring more diverse approaches to solving cSCAN, including both neuro-symbolic approaches and meta-learning approaches. As the primary contributions of this paper are the cSCAN dataset, the code base for generation of this family of datasets, and the novel task format, however, which already lead to a quite broad scope of content to cover in the paper in the current 9-page format, we would propose to avoid further expansion of the scope of this initial paper, and instead deal evaluation of the additional solution approaches in follow-up papers. While there are existing meta-learning and program synthesis or grammar induction solutions, for example, that we could build on, we do expect each of these to require some degree of adaptation to the CLT format, with a corresponding amount of explanation in the paper, in order to do them justice. We have updated the future work section to specifically mention neuro-symbolic solutions as an area of interest to explore.
>
> Regarding the more general theme of exploring more axes in our experimental results, we have now significantly expanded the content of what are now Appendices M, N and P to explore additional experimental axes in the form of breakdown analyses and more detailed qualitative error analyses. Please see the notes we shared in our response to Reviewer TbTf for details.

---

> ### Author Response · Authors · 2022-11-19
> **Author Response 2: Intro; Sec 2; Sec 3 (Context varying across examples)**
>
> **Intro: How rules are to be used**
>
> > Even from this point, I was very much wondering what the systems are supposed to do with the rules. Are these going to be introduced as language input somehow? I hoped not (or at least, not just). Given their symbolic nature, some form of neuro-symbolic approach would be most suitable. Something like neural module nets or a program synthesis approach.
>
> As described above, we agree that neuro-symbolic approaches could also be interesting to explore as future work. Regarding how the rules are consumed by our current Transformer baselines, we describe this as mentioned earlier in Appendix M: Input-Output Format (now Appendix O in the latest draft, to which we have also added further details). To help other readers who may have the same question on their minds, we have also introduced a reference to Appendix O much earlier in the paper now, in Section 2.2 (Structural Definition).
>
> **Sec 2.2: Recursive definition of contexts**
>
> > I find the recursive definition of contexts odd, especially consdiering that the examples and even the introduced data set do not contain any context that is itself an example. Couldn't it then be defined as C \in 2^(Q \times R \times U), simplifying the definition? What is the advantage of including C recursively in the definition?
>
> Thank you for this suggestion. Indeed, for cSCAN, it is not necessary to define the context and examples recursively, and we appreciate that introducing this most general form of a CLT upfront may have introduced an unnecessary amount of cognitive load for first-time readers of the paper. We have accordingly updated the structural definition of CLTs presented in Section 2.2 to adopt a simpler and more concise notation, without recursion, and with just a single standard form of this notation that is used throughout the main body of the paper. The discussion of the more general form of the CLT definition (which allows for nested contexts), along with some discussion of the motivation for when this could be useful, has been now moved into Appendix C: Generalization of CLTs to Support Nested Contexts.
>
> **Sec 2.2: Introduction of word 'defeasible'**
>
> > I suggest that the authors include the word "defeasible" earlier, much like they did with "monotonically" in the examples from Sec. 2.1
>
> Thank you for this suggestion. We have updated the paper accordingly.
>
> **Sec 2.3: Connecting explanation to original motivating example**
>
> > I think it would benefit the readability of this section to continue with the running example of the movie recommendation task.
>
> Thank you for this suggestion. We have updated the paper accordingly.
>
> **Sec 3: Context varying across examples**
>
> > The first time the manuscript clarifies that the context varies generating multiple "tasks" seems to be when introducing cSCAN. Parhaps the authors could state something about this more explicitly earlier on for additional clarity? Particularly in the general CLT descriptions.
>
> Currently, we describe the notion of the context varying across examples in Section 2.1 (Desired Properties). We were struggling to find space to repeat this point again in Section 2.2 without exceeding the page limit or sacrificing other important information, but please let us know if you still find this unclear, and we can continue to look for a way to clarify this in the camera-ready version.

---

> ### Author Response · Authors · 2022-11-19
> **Author Response 3: Sec 3 (Reason for two versions of cSCAN, MCD context split); Source code**
>
> **Sec 3: Reason for two versions of cSCAN**
>
> > Could the authors provide some insight into why two different versions of cSCAN are desireable/useful/interesting? I later came to see that this was due to the "scalability to larger rule sets". It might be worth including that here
>
> Indeed, the reason for multiple versions of cSCAN (four in total: cSCAN-[B|X] [Random|MCD]) is to enable exploration of different axes along which current ML systems may potentially struggle in the CLT setting. While some experimental axes can be explored through breakdown analyses on a single dataset (e.g., for comparing the accuracies on examples of different degrees of complexity in terms of the number of hops of reasoning, or amount of deduction vs. induction involved, etc., which we now explore in Appendix M), some axes would require constructing separate datasets with the different characteristics of interest. In particular, this included the axis of the size of the rule space (cSCAN-B vs. cSCAN-X) and the axis of the type of split (Random vs. MCD). We describe the motivation for these different versions briefly in Section 3.3, specifically in the "Rule space variants" section and the "Splitting methods" section. Some more detail on the motivations for this latter case are provided in what is now Appendix F (Dataset Generation), specifically in the sections on "Splitting" and "Additional top-level example generation". At a high level, the reason for generating different datasets for the splitting methods is that in order to get a stronger signal for our consistency metric, we found it beneficial to perform some additional example generation for each of the test contexts after splitting was complete; however, this approach would be undesirable for an MCD split, as the additional example generation would affect the compound divergence between the train and test sets; for this reason, we chose to generate separate datasets for the Random split (which we use for investigating consistency) and the MCD split (which we use for investigating compositional generalization).
>
> **Sec 3: MCD context split**
>
> > Interestingly, the MCD makes the problem harder in terms of generalizing to unseen top-level requests (I believe this is the same type of generalization as studied in SCAN), but it makes the problem of generalizing to new contexts easier because the contexts might be repeated between train/test. I wonder if the design makes it impossible to have these two simultaneously
>
> Indeed, as we currently perform MCD splits based purely on atoms and compounds derived from the top-level request alone (rather than being derived from the compositional structure of the context), our MCD splits do not tend to segregate the contexts into separate sets of train contexts and test contexts. In principle, we would expect it to be possible to construct a more comprehensive MCD split that tracks atoms and compounds based on both the context and the top-level request, which we would expect to lead to a split that is challenging from the perspective of generalizing to new contexts as well as to new top-level requests. We have not gotten as far as implementing such a split, however, and so would save that for future work.
>
> Regarding the current MCD splitting approach, aside from its relative simplicity, we were also attracted by the fact that even though the same contexts could be repeated in train and test, the task still proved to be quite challenging for the baseline systems. From that perspective, while a comprehensive MCD split would also be valuable, we would find it valuable to also maintain the current type of split which allows us to isolate the challenge of generalizing to new requests from that of generalizing to new contexts.
>
> **Supplement: Source code**
>
> > Source code is promised but not provided at submission time
>
> For reference, we have now uploaded as a supplement an anonymized version of the code used for the dataset generation and consistency metric calculation. While we still need to do a few more clean-up steps to ensure the code is runnable out-of-the-box outside of our own environment, the full logic of the dataset generation process can be seen in the attached code and corresponding unit tests. The main entry point into the dataset generation code is `cscan/generate_benchmark.py`. If there are any specific questions about the source code, we would be happy to point the reviewers to the relevant code locations.

---

> > ### Comment · Reviewer_qoQG · 2022-11-22
> > **Response to authors**
> >
> > Thank you for your response. I have read it in detail, and continue to be of the opinion that this paper is slightly above the acceptance threshold. Below, I repeat the three main concerns I raised in my review and discuss how the authors have addressed them during the rebuttal period.
> > 1. **Not thorough enough evaluation.** The authors included additional analyses in the appendices that dive deeper into their results. My main concern was regarding the types of approaches that were considered for the evaluation, and the authors contended that adding other types of approaches would have made the paper substantially longer. While I appreciate that this may well be the case, I am of the opinion that introducing a new problem definition (CLTs) and accompanying data set (cSCAN) requires assessing how the state of the art can/cannot handle the proposed problem setting. This, to me, is a critical component of the scientific contribution of this type of papers. If it is not possible to condense the contributions to fit them in a conference-length paper (which clearly may be the case, seeing as the current paper is over 60 pages long with appendices), then the authors might consider a journal publication instead.
> > 2. **Details of how transformers consume rules.** The authors clarify that this was already stated in Appendix M (now O), and as I anticipated the rules are just passed in as additional text.
> > 3. **Over-stated claim of generality of CLT data set generation.** The authors have scoped their contributions more appropriately in the revised version.
> >
> > I appreciate the authors' effort in the response period, and reiterate that I do still consider that this paper could be accepted for publication.

---

### Official Review · Reviewer_44yb · 2022-10-25

**Confidence:** 4
**Correctness:** 3
**Technical Novelty And Significance:** 2
**Empirical Novelty And Significance:** 2
**Recommendation:** 3

**Clarity, Quality, Novelty And Reproducibility:**

The paper is an extension of an existing SCAN procedure. The clarity of the paper could be improved in many parts. The used machine learning approach is not reported in the paper and this make difficult to understand the results.

**Strength And Weaknesses:**

- clarity
- experimental evaluation
+ incremental interesting approach

**Summary Of The Paper:**

The paper introduces an improvement of a learning method from samples and rules, named conceptual learning task. In particular the aim could be that of debug a movie recommendation.


**Summary Of The Review:**

The notation adopted in the paper to describe the rules is difficult to understand.

Section 3 is very difficult to follow. For instance the authors say "we generate examples automatically using a Python program". How? what is the meaning of "picking a coherent set of basis rules"? Even the points (a-b-c-d) following the previous sentence are ambiguous and not clear.

After having read the first three section is not clear what is the machine learning task definition.

Another aspect regards the experimental evaluation. It seems that the proposed approach has been evaluated on artificial data in contrast with the example of recommendation system reported in the introduction.

Finally, it should be important to stress the learning process and how it is related to the cSCAN proposed approach.

---

> ### Author Response · Authors · 2022-11-19
> **Author Response 1: Summary of paper; ML approach**
>
> Thank you for your thoughtful feedback.
>
> **Summary Of The Paper:**
>
> > The paper introduces an improvement of a learning method from samples and rules, named conceptual learning task. In particular the aim could be that of debug a movie recommendation.
>
> To clarify one point of potential misunderstanding, this paper does not introduce a new learning method, but rather provides the following novel contributions:
> * a new task format ("conceptual learning task" or "CLT") for evaluating the capabilities of an ML system to learn with and about rules, while encapsulating key properties of a motivating personal assistance use case
> * a new concrete benchmark dataset ("cSCAN") that follows this task format while carefully controlling task complexity by capturing the key properties of a CLT in a simple synthetic task, which despite its simplicity succeeds in highlighting several challenge areas that we believe to be relevant to CLTs in general, including the more complex real-world CLTs such as the motivating assistant example.
>
> Regarding the learning method, the task format and dataset are agnostic to the specific learning method that is applied. For example, as a baseline, we evaluate in this paper the widely-used existing learning method of supervised sequence-to-sequence learning using Transformers. In this approach, each top-level example (context + request) is simply flattened into a sequence that is passed into the Transformer, while the Transformer learns to output a sequence consisting of the reply + qualifier. Other learning methods that could be potentially be evaluated in future work include, for example, meta-learning or program synthesis. In this learning methods, the same cSCAN dataset would be used, and the same accuracy and consistency metrics would be applied, but the learning method may be quite different.
>
> Based in part on your feedback, we have revised the text of the introduction, which we hope will make the contributions and the motivation of the paper clearer.
>
> > The paper is an extension of an existing SCAN procedure.
>
> To clarify, the paper is not an extension of a learning procedure or dataset generation procedure from SCAN, although we do use the SCAN task as a base task from which we derive the "cSCAN" conceptual learning task. The task format presented in our paper is novel and, as illustrated in the connections to the motivating assistant use case, is not specific to the SCAN base task. Similarly, the procedure by which we synthetically generate cSCAN is also novel, with relatively minor overlap with some of the techniques used by earlier researchers in generating the SCAN dataset and other SCAN-derived datasets.
>
> As discussed above, we have revised the text of the introduction, in part based on your feedback, which we hope will make the contributions and the motivation of the paper clearer.
>
> **Machine Learning Approach:**
>
> > The used machine learning approach is not reported in the paper and this make difficult to understand the results.
> > Finally, it should be important to stress the learning process and how it is related to the cSCAN proposed approach.
>
> While as discussed above, this paper does not introduce a new machine learning approach, the approach of how we configure our T5-based baseline solutions and pass the context and examples to them for application to the cSCAN task is described briefly in Section 5 (Baselines), and in much more detail in Appendix J: Reproducibility (now Appendix L in the latest draft) and in Appendix M: Input-Output Format (now Appendix O in the latest draft). We appreciate, however, that these appendices could have been made more discoverable, and we have updated the paper draft to now reference them more explicitly from Section 5 and other relevant places in the paper (in addition to from the Reproducibility Statement). We have also updated Appendix O to include more detailed commentary on the input-output format.

---

> ### Author Response · Authors · 2022-11-19
> **Author Response 2: Notation; Procedure for generating cSCAN; Use of synthetic data**
>
> **Notation:**
>
> > The notation adopted in the paper to describe the rules is difficult to understand.
>
> Based on your feedback and that of Reviewers qoQG and N6X6, we understand that the notation for CLT examples introduced in Section 2.2 was indeed more complicated than necessary for the simple cSCAN task, particularly with regard to the recursive definition of the contexts (which we do not actually make use of for cSCAN). We appreciate also that the different shorthand notation variations that we introduced in this same section to allow more concise human-readable notation of some examples may have also contributed to the cognitive load of the new notation. We have accordingly updated the structural definition of CLTs presented in Section 2.2 to adopt a simpler and more concise notation, without recursion, and with just a single standard form of this notation that is used throughout the main body of the paper. The discussion of the more general form of the CLT definition (which allows for nested contexts), along with some discussion of the motivation for when this could be useful, has been now moved into Appendix C: Generalization of CLTs to Support Nested Contexts.
>
> **Section 3: Procedure for generating cSCAN**
>
> > Section 3 is very difficult to follow. For instance the authors say "we generate examples automatically using a Python program". How? what is the meaning of "picking a coherent set of basis rules"? Even the points (a-b-c-d) following the previous sentence are ambiguous and not clear.
> > After having read the first three section is not clear what is the machine learning task definition.
>
> The Python program referred to in Section 3.2 is the same one that implements the dataset generation procedure that Section 3.2 describes, and which is described in further detail in Appendix F: Dataset Generation (Appendix F in the latest draft). We appreciate that discoverability of Appendix F could have been improved, and we have now updated the paper draft to reference this more explicitly early on (rather than just in the Reproducibility Statement).
>
> As noted in Footnote 1, this program is also something we are planning to open-source upon paper acceptance. For reference, we have now uploaded as a supplement an anonymized version of the code used for the dataset generation and consistency metric calculation. While we still need to do a few more clean-up steps to ensure the code is runnable out-of-the-box outside of our own environment, the full logic of the dataset generation process can be seen in the attached code and corresponding unit tests. The main entry point into the dataset generation code is `cscan/generate_benchmark.py`.
>
> **Experimental evaluation: Use of synthetic data**
>
> > Another aspect regards the experimental evaluation. It seems that the proposed approach has been evaluated on artificial data in contrast with the example of recommendation system reported in the introduction.
>
> While the paper is not evaluating a new machine learning procedure, it is true that the cSCAN dataset that we present in this paper (our first concrete instance of a CLT dataset) is a simple and synthetically-generated one. We would argue, however, that this is not a weakness of the paper, but rather a strength. Specifically, one benefit of abstracting out key properties of interest from our motivating use case into the definition of a CLT, as we did, is that this opens up the possibility to study the capabilities of current ML methods on this family of tasks by starting with a very simple synthetic-generated CLT instance, that enables exploration of ML system performance on CLTs, while carefully controlling the task complexity. In the related research area that studied the general phenomenon compositional generalization, for example, the existence of simple synthetic tasks such as SCAN (Lake and Baroni, 2017) and later CFQ (Keysers, et al., 2020) that succeeded in capturing the challenge of compositional generalization in very simple and well-controlled environments, provided an important test bed for research on improving solutions for compositional generalization. Like these, we hope that cSCAN can serve as a valuable test bed for solutions to the challenges of conceptual learning tasks and a stepping stone toward solutions to the motivating real world tasks.
>
> We appreciate, however, that we could have made the connection between cSCAN and our motivating use case clearer, and we have updated the wording of our introduction and the beginning of Section 3 to improve this.
>
> Brenden M. Lake and Marco Baroni. Generalization without systematicity: On the compositional skills of sequence-to-sequence recurrent networks. 2017.
>
> Daniel Keysers, et al. Measuring Compositional Generalization: A Comprehensive Method on Realistic Data. 2020.

---

### Official Review · Reviewer_VbTf · 2022-11-05

**Confidence:** 4
**Correctness:** 3
**Technical Novelty And Significance:** 3
**Empirical Novelty And Significance:** 3
**Recommendation:** 6

**Clarity, Quality, Novelty And Reproducibility:**

**Clarity**: The writing quality is good but I think there could be significant improvements to the presentations of the ideas in the paper, as discussed above. Also a small comment beyond those above: would be good that figure 2 would be much closer to where it’s mentioned (page 1) compared to where it’s at right now (page 4).

**Novelty**: The explicit discussion and exploration of evaluation of systematic learning and the approach introduced is both novel and pursuing an important and underexplored direction in the field.

**Results**: A theoretical idea is introduced (conceptual learning, a method for creating benchmarks) and then a concrete application of it is explored (a dataset) and analyzed (experiments over language models). So the paper has the necessary components to convey a full story. I do recommend having more experiments and more baselines though (see comments above).

**Reproducibility**: The paper and especially appendixes provide a lot of detail that could allow reproducing the paper.


**Strength And Weaknesses:**

**Strengths**: The idea is novel and it explores an important direction. See more details in the question below. The paper presents an idea, concrete usage of it, and empirical analysis on metrics of accuracy and consistency. The dataset section presents the data in sufficient detail, and the baselines are also described with a good level of detail. Exploring more baselines beyond T5 though would strengthen the paper.

**Weaknesses**:
* **Presentation of the motivation, real-world case**: The motivating example about the movie recommendations used to open the paper is not convincing in my opinion.  First, in general movie recommendation is more of a typical case of example-based learning, where people too give recommendations not by strict rules but by an approximate sense of what prior movies a person like, and what are his general preferences. More specifically, all except one of the rules presented in the example to illustrate rule-based inference are actually not rules but either facts or evidence. This is problematic as an introduction since evaluating rule application in language models is the main theme of the paper, where the example doesn’t show actual rules. Other more natural examples could be used to illustrate where people use rules, beyond the most obvious examples of math or physics problems etc, one could use examples of day-to-day such as admission criteria to a school or university, rules about payments or finance, examples of rules from the law system, etc etc.

* **Presentation of the motivation, SCAN case**: The second example about SCAN is closer to exemplify about rules but the claim about people presented with rules rather than examples is not compelling either. E.g. if I showed a person example that f(A B and then C B)=aacc , f(D earlier than C earlier than A)=acd, we would quickly look for identifying rules in the example, learning that “and then” means putting after, “earlier than” means before and B means to do the action twice, even without being presented with the rules. People have a tendency to look for simple rules in examples, and they identify general rules (sometimes to general or strong) very commonly. People also commonly do induction and not only deduction, e.g. if f(abc)=d, and f(klm)=n then they will figure out that f(wxy) is likely to be z. Overall, I totally agree that rule-based learning is important in real-world and that AI models aren’t strong in this silk, but there could be the presentation of the motivation for that could be improved a lot.

* **Conceptual learning task definition**: The concept seeks to achieve  5 properties that the learner should possess. No justification is given about why these 5, how they were selected, how they complement each other into a common goal etc. I believe the definition should be reworked to be more cohesive. The definitions of some of the properties also weren't clear to me.

* **Experiments**: I would suggest to extend the experimental section, e.g. see the impact of the complexity of the rules and number of rules needed on the performance along different metrics, see generalization across multiple dimensions beyond only compositional generalization, do error analysis of what mistakes and trends are common in terms of accuracy and consistency and if any general behaviors and trends of models (vs. even human study) could be identified, etc.


**Summary Of The Paper:**

A new approach is introduced for constructing benchmarks for the application of consistent rules in language models. This approach is used to create a new dataset that is then used to evaluate T5 models on 3 dimensions: applying learned rules, scaling to larger sets of rules and compositional generalization.


**Summary Of The Review:**

I think overall the paper explores and important direction but could be improved in the presentation and extended with more experiments to make it much better and therefore at this time I recommend rejection but also wish to emphasize that I encourage the authors to keep working on the paper to make it better and then resubmit to a conference!

---

> ### Author Response · Authors · 2022-11-19
> **Author Response 1: Presentation of the motivation, real-world case**
>
> Thank you for your thoughtful feedback.
>
> **Presentation of the motivation, real-world case:**
>
> > The motivating example about the movie recommendations used to open the paper is not convincing in my opinion. First, in general movie recommendation is more of a typical case of example-based learning, where people too give recommendations not by strict rules but by an approximate sense of what prior movies a person like, and what are his general preferences.
>
> One point that we did not make clear enough in our presentation of the introductory example is that the motivating use case that we are interested in is a personal assistant that **among other things** is expected to make movie recommendations, as opposed to a system that gives movie recommendations alone. We have updated the text in the introduction to clarify this.
>
> It is also true, as you note, that there are use cases (in a personal assistant or otherwise) in which the need to perform rule-based reasoning are even more obvious. That said, one reason that we chose a motivating example involving movie recommendations is precisely to illustrate how widespread the use of knowledge in the form of rules and facts is in personal assistant scenarios -- even in cases like movie recommendations that one would traditionally associate with example-based recommender systems --and to illustrate how to achieve human-like capabilities in many tasks we need to look at the combination of both deduction and induction (in learning from both rules and examples), rather than bucketizing tasks into ones that are solvable specifically through rule-based approaches or specifically through example-based approaches. We have updated our presentation in the introduction as a whole to better clarify this motivation.
>
> > More specifically, all except one of the rules presented in the example to illustrate rule-based inference are actually not rules but either facts or evidence.
>
> Although we are necessarily space-constrained in the small figure, our intention in Figure 1 is to illustrate the full spectrum of different types of knowledge relevant to a personal assistant task, including the variety of forms in which the different types of "rules" (and "examples") may appear in natural language. While in order to construct a coherent example it was necessary to include multiple "facts" to interact with each of the more traditional "rules", we believe we nevertheless were able to cover the relevant spectrum of different types of knowledge:
> * Item 1:  A "rule" of the most traditional type that states conditional knowledge that can apply to many different movies.
> * Item 2: A concept definition, which can be equivalently construed as a rule relating two different pieces of information about a person.
> * Items 3-6: Facts stated at varying levels of granularity.
> * Item 7: A fact directly about movie preferences, which can also be construed as an example of the underlying "movie recommendation" task.
>
> We appreciate that this may not have been clear enough to readers in our original draft and based on your feedback have added a detailed caption to the figure explaining the different types of knowledge / rules that are illustrated in the figure. We have also updated the draft of the introduction to better convey that our intention is specifically to explore the combined scenario of rule-based deduction and example-based induction.
>
> > This is problematic as an introduction since evaluating rule application in language models is the main theme of the paper, where the example doesn’t show actual rules.
>
> To be clear, as noted above, the main theme of this paper is not intended to be evaluating rule application alone, but rather the combination of rule application ("learning with rules") and rule induction ("learning about rules" -- including, especially, learning about rules from examples).
>
> > Other more natural examples could be used to illustrate where people use rules, beyond the most obvious examples of math or physics problems etc, one could use examples of day-to-day such as admission criteria to a school or university, rules about payments or finance, examples of rules from the law system, etc etc.
>
> As you noted, there are a number of application areas in which the need for rule-based reasoning may be even more obvious than in a personal assistant, and the examples you mentioned of math, physics, university admissions, finance, and law are all great ones. We would push back, however, on the suggestion that the use of rule-like knowledge is not relevant to a personal assistant, for the reasons discussed above. Our preference in this paper is to focus on the personal assistant motivating use case in the introduction, in part because this is an actual use case which has motivated our current line of research into conceptual learning. We hope that our updated presentation in the introduction is now able to more effectively convey to the reader the reasons why we find this use case compelling.

---

> ### Author Response · Authors · 2022-11-19
> **Author Response 2: Presentation of the motivation, SCAN case; CLT definition; Positioning of figures**
>
> **Presentation of the motivation, SCAN case:**
>
> > The second example about SCAN is closer to exemplify about rules but the claim about people presented with rules rather than examples is not compelling either. E.g. if I showed a person example that f(A B and then C B)=aacc , f(D earlier than C earlier than A)=acd, we would quickly look for identifying rules in the example, learning that “and then” means putting after, “earlier than” means before and B means to do the action twice, even without being presented with the rules. People have a tendency to look for simple rules in examples, and they identify general rules (sometimes to general or strong) very commonly. People also commonly do induction and not only deduction, e.g. if f(abc)=d, and f(klm)=n then they will figure out that f(wxy) is likely to be z.
>
> We definitely agree that induction from examples is as much a part of human learning as is rule-based deduction. Indeed, one of the key motivations of the definition of a CLT is to explore the interaction between these two types of learning, rather than evaluating just one of these in isolation -- i.e., to consider both "learning with rules" (deduction) and "learning about rules" (induction) in the same task. We also like your points about how humans are often able to guess SCAN-like rules in particular from a small number of examples, and we have updated our presentation in the introduction to mention this explicitly.
>
> **Conceptual learning task definition:**
>
> > The concept seeks to achieve 5 properties that the learner should possess. No justification is given about why these 5, how they were selected, how they complement each other into a common goal etc. I believe the definition should be reworked to be more cohesive. The definitions of some of the properties also weren't clear to me.
>
> The common goal that we believe connects these properties and makes them complement one another is the goal of "learning with and about rules". We agree that this could have been made clearer in the presentation, however, and we have updated the paper draft to make this more explicit, including adding more examples to the properties of what these mean in terms of the motivating example from the introduction. We have also removed the 5th property ("Consistency") from the list, as while this is an important property for a diagnostic CLT like cSCAN (as discussed in Section 2.3: Consistency Requirements), it is less directly motivated by the motivating example itself.
>
> **Clarity: Positioning of figures**
>
> > Also a small comment beyond those above: would be good that figure 2 would be much closer to where it’s mentioned (page 1) compared to where it’s at right now (page 4).
>
> Thank you for this suggestion! We have re-arranged the figures so that Figure 2 now appears much earlier on page 2, while Figure 1 now appears on page 1.

---

> ### Author Response · Authors · 2022-11-19
> **Author Response 3: Experiments, Baselines**
>
> **Experiments:**
>
> > I would suggest to extend the experimental section, e.g. see the impact of the complexity of the rules and number of rules needed on the performance along different metrics, see generalization across multiple dimensions beyond only compositional generalization, do error analysis of what mistakes and trends are common in terms of accuracy and consistency and if any general behaviors and trends of models (vs. even human study) could be identified, etc.
>
> Thank you for this suggestion. In addition to the basic experiment analyses in Section 6 of the main body, we also include in the appendices a number of finer-grained analyses using breakdown metrics (Appendix K in the original draft, now Appendix M in the latest), fine-grained evaluation metrics (Appendix L in the original draft, now Appendix N in the latest), and qualitative error analysis (Appendix N in the original draft, now Appendix P in the latest).
>
> In line with your suggestions, we have further strengthened these supplementary analyses to include the following:
>
> * Appendix M.5: Effect of Example and Context Characteristics
>
>   In this section (as well as the other existing parts of Appendix M), we illustrate a number of examples of the type of drilldowns into qualitative ML system behavior that are enabled by the unique structure of the CLT and the types of example metadata that are provided with cSCAN. One thing we can observe here, for example, is that T5 accuracy decreases as the number of unique rules or the number of rule compositions involved in a given example increases, highlighting multi-hop reasoning complexity as another challenge area for the current baseline models.
>
> * Appendix N.2: Consistency Breakdown by Implications and Contradictions
>
>   In this section, we analyze the distribution of the number of different predictions that are involved in each of the implications and contradictions that contribute to the consistency metric scores for each of the models. We can see here, for example, that most of the implications and contradictions from cSCAN-B involve pairs of predictions that directly imply or contradict one another, while in cSCAN-X there are also a significant number of implications and contradictions involving interactions between 3 or 4 predictions.
>
> * Appendix P.2: Consistency error analysis
>
>   In this section, we perform manual bucketization analysis of the specific types of inconsistencies that appear in a random sample of the various sized contradictory sets from the strongest performing model (T5-Large) on the cSCAN-X Random dataset, where it achieved a consistency of 81.6%. The analysis here combined with that in Appendix N.2 above shows, for example, that T5-Large directly contradicts itself in a surprisingly large number of cases, where, for example, it may claim multiple different rules to be true that map the same left-hand side to different right-hand sides, or where it claims a certain "fact" (i.e., variable-less rule) to be true, but then behaves in a different way when the exact same scenario is presented in non-rule form.
>
> **Baselines:**
>
> > Exploring more baselines beyond T5 though would strengthen the paper.
>
> We are indeed also interested in exploring more diverse approaches to solving cSCAN, including for example neuro-symbolic approaches or meta-learning approaches. As the primary contributions of this paper are the cSCAN dataset, the code base for generation of this family of datasets, and the novel task format, however, which already lead to a quite broad scope of content to cover in the paper in the current 9-page format, we would propose to avoid further expansion of the scope of this initial paper, and instead deal with evaluation of the additional solution approaches in future work. While there are existing meta-learning and program synthesis or grammar induction solutions, for example, that we could build on, we do expect each of these to require some degree of adaptation to the CLT format, with a corresponding amount of explanation in the paper, in order to do them justice.
>
> In the more general area of strengthening the experiment results of the paper, please see also the new additions to the paper described under "Experiments" above.

---

### Author Response · Authors · 2022-11-21
**Thank you to all the reviewers**

We thank all the reviewers for the thoughtful comments. These have been very helpful in enabling us to further improve the paper, for which we uploaded a new revision together with related source code on Friday.

Below we summarize our understanding of the main highlights of the initial round of reviews, together with the changes we made to address the areas for improvement. (Please see responses sent earlier to individual reviewers for additional details.)

**Main strengths highlighted in initial reviews:**
* Idea and approach for investigating learning with and about rules is novel and important (VbTf, qoQG).
* Provides comprehensive details about how the dataset is generated, and thorough reproducibility information in general. (VbTf, N6X6).
* Writing quality is good and easy to follow (with the exception of specific areas of improvement listed below). (VbTf, qoQG)
* Effective mix of paper contributions: theoretical idea + concrete application + experimental analysis. (VbTf)
* Running example of personal assistant helps motivate and provide context to the reader. (qoQG)
* Solid connections to related literature. (qoQG)

**Main areas for improvement highlighted in initial reviews:**
* Relationship between the motivating assistant example and cSCAN was not clear enough. (VbTf, 44yb, N6X6)
  * ⇒ Improved in the revised draft.
* Some notation was confusing, particularly the recursive definition of the context. (qoQG, N6X6)
  * ⇒ Simplified the notation in the revised draft.
* Was not clear enough how the transformers consume the context -- i.e., what exactly the task looks like from the perspective of the transformer. (44yb, qoQG)
  * ⇒ In the revised draft, introduced earlier and more prominent references to Appendix O (Input-Output Format), and added further commentary to this appendix.
* Experiment section would benefit from additional dimensions affecting ML performance, additional baselines, and/or or additional error analysis. (VbTf, qoQG)
  * ⇒ In the revised draft, included additional breakdown analyses illustrating dimensions affecting ML performance (Appendix M.5), and additional quantitative and qualitative analyses of consistency errors (Appendix N.2, Appendix P.2).
* The common goal connecting the 5 properties of the CLT definition was not clear enough. (VbTf)
  * ⇒ Improved in the revised draft.
* Made too broad a claim about providing a general-purpose dataset-generating method, when we only demonstrated it on cSCAN. (qoQG)
  * ⇒ Scaled back this claim in the revised draft.
* Source code for dataset generation was not provided (though promised to be open-sourced later). (qoQG)
  * ⇒ Uploaded a supplement containing the full dataset generation and consistency metric calculation code.

**Summary of changes:**
* Updated the introduction and early sections to clarify the connections between the assistant motivating example, the original SCAN task, cSCAN, and the goal of "learning with and about rules".
* Further clarified the paper's focus on the combination of deduction ("learning with rules") and induction ("learning about rules", including learning with examples), as opposed to purely deduction.
* Updated the abstract and introduction to avoid highlighting the dataset-generating method as an independent contribution.
* Clarified the rules and other types of knowledge illustrated in Figure 1.
* Added references to Appendix O (Input-Output Format) more prominently and earlier on in the text, and added additional commentary to clarify the exact format in which the context and examples are passed to the T5 baselines.
* Simplified the notation used in the main body of the paper and in most of the appendices to avoid mention of nested contexts, and to use just a single standard notation rather than multiple shorthand variants. Moved discussion of the more general CLT formulation to a new Appendix C (Generalization of CLTs to Support Nested Contexts).
* Added Appendix G (Consistency Metric Calculation), describing how the consistency metric can be calculated efficiently in practice.
* Added Appendix M.5 (Effect of Example and Context Characteristics), containing additional breakdown analyses illustrating dimensions affecting ML performance.
* Added Appendix N.2 (Consistency Breakdown by Implications and Contradictions), containing additional quantitative analysis of consistency errors.
* Added Appendix P.2 (Consistency Error Analysis), containing additional qualitative analysis of consistency errors.
* Uploaded a supplement containing the full dataset generation and consistency metric calculation code.

**Minor changes:**
* Repositioned figures to appear closer to their first mentions in the text.
* Moved some more of the related work to the "extended related work" appendix to make room for the additional clarifying content described above.
* Made a few other minor improvements to wording and layout.

---

### Decision · Program_Chairs · 2023-01-20

**Decision:**

Reject

**Justification For Why Not Higher Score:**

While the paper presentation improved a lot during the rebuttal, the paper is still not ready for publication and the general use of the proposed benchmark is still questionable.

**Justification For Why Not Lower Score:**

N/A

**Metareview: Summary, Strengths And Weaknesses:**

The authors tackle the important problem of quantifying learning tasks that require a combination of induction and deduction. They do so by proposing conceptual SCAN (cSCAN) as a synthetic benchmark derived from SCAN for conceptual learning tasks (CLTs), i.e., tasks where language models are provided contexts and rules and asked for conceptual explainations.

The reviewers appreciated the direction the authors are taking and praised the construction of cSCAN from first principles. At the same time, they highlighted several weaknesses. These mainly include a weak presentation (missing some details about how contexts are obtained, how the program generation operates, over-complex notation, etc) and the lack of the general applicability to real-world scenarios for the proposed CLT learning task.

During the rebuttal, the authors greatly improved the presentation of the work e.g. by addressing concerns about notation and some missing details about the construction and evaluation from the main text. Reviewers acknowledged this, but still believed that the motivation behind using cSCAN as a benchmark can be lacking. More crucially, the construction of the benchmark and the evaluation procedure can be more streamlined and documented as to be made fully explicit and reproducible.

I personally believe the paper has a strong potential, but in its current form it is not yet ready for publication. I encourage authors to take the remaining reviewers' suggestions into consideration for a future submission.

**Summary Of Ac-Reviewer Meeting:**

After rebuttal, reviewers converged to rejection.